# KNOWLEDGE-CONSISTENT DIALOGUE GENERATION WITH LANGUAGE MODELS AND KNOWLEDGE GRAPHS

## ABSTRACT

Pre-trained language models have achieved impressive performances on dialogue generation tasks. However, when generating responses for a conversation that requires factual knowledge, they are far from perfect, due to the absence of mechanisms to retrieve, encode, and reflect the knowledge in the generated responses. Some knowledge-grounded dialogue generation methods tackle this problem by leveraging the structured knowledge from Knowledge Graphs (KGs). However, existing methods do not guarantee that the model utilizes a relevant piece of knowledge from the KG before generating knowledge-consistent dialogues. To overcome this limitation, we propose **SU**bgraph **R**etrieval-augmented **GE**neration (**SURGE**), a framework for generating context-relevant and knowledge-consistent dialogues with a KG. Specifically, our method first retrieves the relevant subgraph from the KG, and then enforces consistency across facts by perturbing their word embeddings conditioned on the retrieved subgraph. Then, it learns a latent representation space using contrastive learning which ensures that the generated texts have high similarity to the retrieved subgraphs. We validate the performance of our SURGE framework on the OpendialKG and KOMODIS datasets and show that our method generates high-quality dialogues that faithfully reflect the knowledge from the KG.

## 1 INTRODUCTION

Dialogue systems aim at conversing with humans by generating human-like responses, given the dialogue context. While pre-trained language models (PLMs) (Radford et al., 2019; Raffel et al., 2020) are capable of generating fluent responses, they often generate factually incorrect responses due to a lack of explicit knowledge (Shuster et al., 2021). To overcome such limitations, recent methods access the external knowledge sources, such as Wikipedia (Dinan et al., 2019) or Web (Komeili et al., 2021), and then retrieve the relevant knowledge for ongoing conversations. In addition to such document-based retrieval approaches, there also exists a variety of works (Tuan et al., 2019; Wu et al., 2020; Zhang et al., 2020a; Cui et al., 2021; Zhou et al., 2021; Galetzka et al., 2021; Li et al., 2022), which focus on the use of the Knowledge Graphs (KGs) (Bollacker et al., 2008; Vrandecic & Krötzsch, 2014) – a different form of the knowledge source which succinctly encodes the knowledge in the most compact and effective form – in dialogue generation. Specifically, KGs consist of symbolic facts which represent entities as nodes and their relations as edges, in the triplet, e.g., *(Pride & Prejudice, written by, Jane Austen)* (See Figure 1), which can help generate a knowledge-grounded response.

Most of the dialogue generation models with KGs (Galetzka et al., 2021; Li et al., 2022) utilize all the triplets associated with the entity in the dialogue context. However, not all of the facts are relevant to the ongoing conversation (e.g., Jane Austen was born in Steventon in Figure 1), which could mislead the models from generating factually incorrect responses. We found that about 87% of facts from 1-hop KG are irrelevant to the context in the OpendialKG dataset (Moon et al., 2019). Moreover, encoding all the facts including the unnecessary ones is computationally inefficient (Galetzka et al., 2021; Rony et al., 2022). On the other hand, even after correctly retrieving the relevant facts, it is not straightforward to combine two heterogeneous modalities: the dialogue context is represented as a text, meanwhile, the knowledge is represented as a graph. In other words, since PLMs already have tons of pre-trained parameters trained on the unstructured texts, properly conditioning the structured graph to PLMs is highly important. Otherwise, PLMs may generate inconsistent responses disregarding the knowledge from the retrieved subgraph, which is a phenomenon known as hallucination (Rohrbach et al., 2018), where they generate responses with their own memorized yet incorrect knowledge.

Figure 1: **Motivation.** Existing knowledge-grounded dialogue generation models with KG utilize the multi-hop subgraph for entities in the dialogue context (**Jane Austen**). However, they suffer from the following two problems: **(1) irrelevant knowledge** where only 12.6% of facts from 1-hop KG are useful to generate the target responses given a dialogue context, and **(2) inconsistent generation** including the factually wrong statement.

In this work, we tackle such challenging and fundamental issues of knowledge-consistent dialogue generation with the KG. We propose an end-to-end dialogue generation framework that considers all aspects from knowledge retrieval, encoding, and reflection along the generation process. As a first step, we propose a context-relevant subgraph retriever that retrieves only the relevant triplets from the KG to prevent the model from generating context-irrelevant responses. Notably, our subgraph retrieval method embeds the KG considering the relational structure with the Graph Neural Network (GNN) (Kipf & Welling, 2017) instead of using PLMs as in previous work (Li et al., 2022). Furthermore, it is end-to-end trainable jointly with the generation objective by marginalizing the likelihood of the generated sentences over the latent retrieved subgraph (Guu et al., 2020; Lewis et al., 2020b). Then, to encode the retrieved subgraph along with the input text sequence, we propose a graph encoding that is permutation and relation inversion invariant yet efficient. Specifically, we devise the graph encoding method that reflects the graph structure onto the representation space of PLMs, instead of prepending them in front of the text sequence to avoid the computational burden. Furthermore, to ensure that the model does make use of the encoded knowledge when generating responses, we propose a multi-modal contrastive learning objective between two different graph-text modalities to enforce the consistency across the retrieved facts and the generated texts. We call our framework **SU**bgraph **R**etrieval-augmented **GE**neration (**SURGE**).

We validate our framework on the OpendialKG (Moon et al., 2019) and KOMODIS (Galetzka et al., 2020) datasets against relevant baselines. Note that, when evaluating the generated responses from dialogue models, conventional metrics (e.g., BLEU (Papineni et al., 2002), Rouge (Lin, 2004)) can not measure how faithfully the generated responses reflect the related knowledge in KGs. Thus, in evaluation, we further introduce an additional performance metric, referred to as Knowledge-verifying Question Answering (KQA), which evaluates whether the generated responses contain the correct knowledge with an additional extractive question answering scheme. The experimental results show that SURGE generates responses that not only agree with the gold knowledge but are also consistent with the retrieved knowledge from KGs.

Our main contributions can be summarized as follows:

- We propose a GNN-based context-relevant subgraph retrieval method for KG-augmented dialogue generation, to extract only the relevant piece of the knowledge for the dialogue context from the entire knowledge graph, for generating more appropriate responses to the ongoing conversation.

- We propose an invariant yet efficient graph encoder and a graph-text contrastive learning objective to ensure that the generated responses faithfully reflect the retrieved knowledge.

- We validate SURGE against relevant baselines, demonstrating its efficacy in generating responses that are more informative by retrieving and reflecting the relevant knowledge from the KG.

## 2 RELATED WORK

**Language Models**   Pre-trained Language Models (PLMs) (Radford et al., 2019; Lewis et al., 2020a; Raffel et al., 2020) that use a Transformers-based (Vaswani et al., 2017) encoder-decoder architecture have achieved great successes on language generation tasks. As they can accurately contextualize the given context and then generate human-like sentences, they are often used as the base architecture for neural dialogue systems (Zhang et al., 2020b; Hosseini-Asl et al., 2020). Moreover, when PLMs become larger, dialogue models have shown to generate high-quality responses (Adiwardana et al., 2020), suggesting that pre-trained parameters do contain certain knowledge (Petroni et al., 2019). Despite the fluency of such PLM-based dialogue agents, they often generate factually incorrect

responses that are unfaithful to the context but look plausible – widely known as hallucination (Maynez et al., 2020). Thus, generating responses requiring specific and valid factual knowledge is still challenging. To tackle this, recent studies propose to retrieve knowledge from external sources, and then use it to augment dialogue models (Roller et al., 2021; Shuster et al., 2021).

**Dialogue Generation with KGs**   The sources of external knowledge can be categorized into two types: documents from unstructured corpora (e.g., Wikipedia (Dinan et al., 2019), Web (Nakano et al., 2021)), and symbolic facts from Knowledge Graphs (KGs) (e.g., Freebase (Bollacker et al., 2008), Wikidata (Vrandecic & Krötzsch, 2014)). Regarding dialogue generation tasks with KGs that we target, Moon et al. (2019) introduce a knowledge-grounded dialogue dataset where each dialogue comes with facts from the large-scale KG. Several works (Tuan et al., 2019; Wu et al., 2020; Zhang et al., 2020a; Cui et al., 2021; Zhou et al., 2021) suggest sequence-to-sequence models trained from scratch, which focus on generating dialogue by conditioning the output word distribution with the entities from the KG. Further, Galetzka et al. (2021) propose an efficient way to encode all of the facts in the $k$-hop neighbors of the entities that appear in the dialogue history in the given KG, in order to reduce the number of input tokens used in PLMs. On the other hand, Rony et al. (2022) propose to mask out weights for irrelevant facts in PLMs. However, all of these methods simply match and retrieve all facts for entities that appear in the dialogue context, which either may mislead the agent to generate out-of-context responses from irrelevant facts or can increase the computational overheads for prepending tokens for all facts in PLMs. Our work differs from those existing works, since we aim at retrieving only a context-relevant subgraph among all associated facts with a novel GNN-based subgraph retriever, which is end-to-end trainable along with a dialogue generation model.

## 3   METHOD

In this section, we first discuss the basic ingredients: Transformer and Graph Neural Network. We then formalize the dialogue generation problem and describe key components for our **SU**bgraph **R**etrieval-augmented **GE**neration (**SURGE**) framework: context-relevant subgraph retrieval, invariant graph encoding, and graph-text contrastive learning. Figure 2 illustrates the overview of our framework.

### 3.1   PRELIMINARIES

As we use two different modalities, namely text and graph, we first define them, and then describe the neural networks to encode them. In particular, a text is defined as a sequence of tokens $\boldsymbol{x} = [x_1, ..., x_N], \forall x_i \in \mathcal{V}$, where $x_i$ is a token and $\mathcal{V}$ is a pre-defined vocabulary formed with specific tokenization algorithms (Sennrich et al., 2016). On the other hand, a knowledge graph (KG) is a type of multi-relational graphs $\mathcal{G} = \{(\mathsf{e}_h, \mathsf{r}, \mathsf{e}_t)\} \in \mathcal{E} \times \mathcal{R} \times \mathcal{E}$, where $\mathsf{e}_h$ and $\mathsf{e}_t$ are head and tail entities (nodes) along with their relation (edge) $\mathsf{r}$; and $\mathcal{E}$ and $\mathcal{R}$ are sets of entities and relations, respectively.

To easily access different modalities in the same framework, we define the tokenization (mapping) function that maps entities and relations to the tokens used in Pre-trained Language Models (PLMs), represented as follows: $q : \mathcal{E} \cup \mathcal{R} \rightarrow \mathcal{V}^l$ where $l$ is an arbitrary length varying across different entities and relations. In other words, any entity $\mathsf{e} \in \mathcal{E}$ and relation $\mathsf{r} \in \mathcal{R}$ consisting of $l$ tokens can be tokenized to a sequence of $l$ tokens $\boldsymbol{x} \in \mathcal{V}^l$: $q(\mathsf{e}) = \boldsymbol{x}_e$ and $q(\mathsf{r}) = \boldsymbol{x}_r$. For instance, an entity *New York* (i.e., $\mathsf{e}$), is tokenized into two tokens 'New' and 'York', i.e., $\boldsymbol{x}_e = [\text{'New', 'York'}]$.

**Transformer**   A Transformer (Vaswani et al., 2017) is a neural architecture that embeds a sequence of tokens while taking their relationships into account, which is the most basic building block of recent PLMs (Devlin et al., 2019; Radford et al., 2019). Formally, given a sequence of input tokens $\boldsymbol{x} = [x_1, ..., x_N], \forall x_i \in \mathcal{V}$, a goal of generative transformers is to generate a sequence of tokens $\boldsymbol{y}_{1:t-1} = [y_1, ..., y_{t-1}], \forall y_i \in \mathcal{V}$, with encoder $\texttt{Enc}$, decoder $\texttt{Dec}$, and tokens' embedding function $f$. Thus, a hidden state at time $t$ for generating $y_t$ is $\boldsymbol{h}_t = \texttt{Dec}(\texttt{Enc}(\boldsymbol{X}), \boldsymbol{Y}_{1:t-1})$, where $\boldsymbol{X} = f(\boldsymbol{x}) = [f(x_1), ..., f(x_N)]$ and $\boldsymbol{Y}_{1:t-1} = f(\boldsymbol{y}_{1:t-1}) = [f(y_1), ..., f(y_{t-1})]$. We note that both $\texttt{Enc}$ and $\texttt{Dec}$ functions are **permutation sensitive** with positional embedding (Vaswani et al., 2017).

**Graph Neural Network**   A Graph Neural Network (GNN) represents a node with its neighboring nodes over the graph structure (Hamilton, 2020), which is formalized as follows:

$$\texttt{GNN}(\boldsymbol{e}_t; \mathcal{G}) = \texttt{UPD}(\boldsymbol{e}_t, \texttt{AGG}(\{\boldsymbol{e}_h \mid \forall \mathsf{e}_h \in \mathcal{N}(\mathsf{e}_t; \mathcal{G})\})), \tag{1}$$

where $\mathcal{N}(\mathsf{e}_t; \mathcal{G}) = \{\mathsf{e}_h \mid (\mathsf{e}_h, \mathsf{r}, \mathsf{e}_t) \in \mathcal{G}\}$ is a set of neighboring entities of $\mathsf{e}_t$; $\boldsymbol{e}_t$ and $\boldsymbol{e}_h$ are embeddings of entities (nodes) $\mathsf{e}_t$ and $\mathsf{e}_h$; $\texttt{AGG}$ is a function that aggregates embeddings of neighboring entities; and $\texttt{UPD}$ is a function that updates $\boldsymbol{e}_t$ with the aggregated messages from $\texttt{AGG}$.

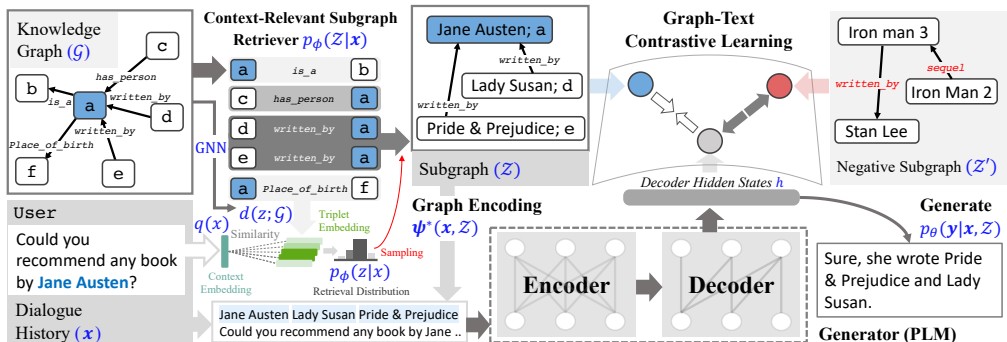

Figure 2: **Framework Overview.** Our framework, SURGE, consists of three parts. First, a context-relevant subgraph retriever $p_\phi(\mathcal{Z}|\boldsymbol{x})$ retrieves the subgraph $\mathcal{Z}$ relevant to the given dialogue history $\boldsymbol{x}$ from a knowledge graph $\mathcal{G}$ (e.g., 1-hop KG from entity *Jane Austen*; a). Specifically, we measure the similarity of a context and triplet embedding to compose the retrieval distribution $p_\phi(z|\boldsymbol{x})$ (§ 3.3). Then, we encode the retrieved subgraph $\mathcal{Z}$ into the input of the generator, using the graph encoding function $\psi(\boldsymbol{x}, \mathcal{Z})$ (§ 3.4). Finally, we use contrastive learning to enforce the model to generate a consistent response with the retrieved subgraph (§ 3.5).

## 3.2 PROBLEM STATEMENT

Here we formalize the problem of context-relevant subgraph retrieval for knowledge-grounded dialogue generation. Given a dialogue history $\boldsymbol{x} = [x_1, \ldots, x_N]$, a model with generative PLMs first encodes the input tokens, and then models a conditional distribution $p(\boldsymbol{y}|\boldsymbol{x})$ to generate an output response $\boldsymbol{y} = [y_1, \ldots, y_T]$. This problem requires a piece of specific knowledge for a conversation.

To that end, given a dialogue history $\boldsymbol{x}$, we aim at retrieving a subgraph $\mathcal{Z} \subseteq \mathcal{G}$ consisting of a set of triplets $z \in \mathcal{Z}$ where $z = (\mathrm{e}_h, \mathrm{r}, \mathrm{e}_t)$, which encodes relevant knowledge for ongoing conversation. Thus, the distribution of the context-relevant facts $\mathcal{Z}$ is $p(\mathcal{Z}|\boldsymbol{x})$, and our final likelihood of generating responses then becomes $p(\boldsymbol{y}|\boldsymbol{x}, \mathcal{Z})$. Then, to jointly optimize the objective of graph retrieval with response generation, we treat $\mathcal{Z}$ as a latent variable and then marginalize the likelihood of the generative model over all possible latent variables for retrieved subgraphs $\mathcal{Z}$, formalized as follows:

$$p(\boldsymbol{y}|\boldsymbol{x}) = \sum_{\mathcal{Z} \subseteq \mathcal{G}} p_\phi(\mathcal{Z}|\boldsymbol{x}) \, p_\theta(\boldsymbol{y}|\boldsymbol{x}, \mathcal{Z}) = \sum_{\mathcal{Z} \subseteq \mathcal{G}} p_\phi(\mathcal{Z}|\boldsymbol{x}) \prod_{t=1}^{T} p_\theta(y_t|\boldsymbol{x}, \mathcal{Z}, \boldsymbol{y}_{0:t-1}), \qquad (2)$$

where $y_0$ is the start token for the generation, $p_\phi(\mathcal{Z}|\boldsymbol{x})$ is an output distribution of the context-relevant subgraph retriever, and $p_\theta(\boldsymbol{y}|\boldsymbol{x}, \mathcal{Z})$ is the target distribution of the knowledge-augmented generator, parameterized as $\phi$ and $\theta$, respectively, which we specify in next few subsections.

## 3.3 GNN-BASED CONTEXT-RELEVANT SUBGRAPH RETRIEVER

We now provide a concrete description of our context-relevant subgraph retriever, i.e., $p(\mathcal{Z}|\boldsymbol{x})$, formalized in Equation 2. Given the dialogue history $\boldsymbol{x}$, we assume that a retrieval of each triplet in $\mathcal{Z} = \{z_1, \ldots, z_n\}$ is independent. Then, for simplicity, we decompose the retrieval of a set of triplets $p(\mathcal{Z}|\boldsymbol{x})$ into the product of individual triplet retrieval, represented as follows: $p(\mathcal{Z}|\boldsymbol{x}) = p(z_1|\boldsymbol{x})p(z_2|\boldsymbol{x})\cdots p(z_n|\boldsymbol{x})$, for $n$ retrieved triples.

From the above decomposition, it is sufficient to focus on a single triplet retrieval. We define the score for the single triplet triplet with inner product between the embedding of dialogue history $\boldsymbol{x}$ and the embedding of candidate triplet $z$ (Guu et al., 2020), as follows:

$$p_\phi(z|\boldsymbol{x}) \propto \exp(d(z)^\top q(\boldsymbol{x})), \qquad (3)$$

where $d$ is a triplet embedding function and $q$ is a dialogue context embedding function. For implementing $q$, we can use any off-the-shelf PLMs, but for $d$, we need another effective approach that captures the property of the graph. Therefore, we utilize the Graph Neural Networks (GNNs) for the triplet embedding function $d$ to consider the relational structure between entities in the KG.

More specifically, we consider a set of triplets associated to the entities that appear in the given dialogue context: $\{(\mathrm{e}, \mathrm{r}, \mathrm{e}_t) \text{ or } (\mathrm{e}_h, \mathrm{r}, \mathrm{e}) \mid q(\mathrm{e}) \subseteq \boldsymbol{x}\}$, as the retrieval candidates. Then, to effectively represent triplets consisting of entities and their relations as items, we use GNNs described in section 3.1. In our triplet retriever, utilizing both nodes and edges, which are equally essential

components for the multi-relational graph, is worthwhile to represent an entire triplet. To do so, we adopt the existing edge message passing framework (Jo et al., 2021) that transforms edges of the original graph to nodes of the dual hypergraph (Scheinerman & Ullman, 2011) (i.e., transforming $\mathcal{G}$ to $\mathcal{G}^*$), which allows us to use existing node-level GNNs for representing edges of the original graph (See Appendix D.1 for more details). Formally, our triplet embedding function is denoted as follows:

$$d(z;\mathcal{G}) = \text{MLP}([\boldsymbol{e}_h \parallel \boldsymbol{r} \parallel \boldsymbol{e}_t]), \ \boldsymbol{e}_h = \text{GNN}(\boldsymbol{e}_h^0;\mathcal{G}), \ \boldsymbol{r} = \text{GNN}(\boldsymbol{r}^0;\mathcal{G}^*), \ \boldsymbol{e}_t = \text{GNN}(\boldsymbol{e}_t^0;\mathcal{G}), \quad (4)$$

$$\boldsymbol{e}^0 = \begin{cases} \sum_{i=start}^{end} \text{Enc}(\boldsymbol{X})_i/(end-start+1), & \text{if } q(\texttt{e}) \subseteq \boldsymbol{x} \\ \boldsymbol{0}, & \text{otherwise} \end{cases}$$

where $z = (\texttt{e}_h, \texttt{r}, \texttt{e}_t)$, $q(\texttt{e}) = [\boldsymbol{x}_i, \dots, \boldsymbol{x}_j]$, $\boldsymbol{0}$ is a zero vector, and $\parallel$ is the concatenation operator. If the entity $\texttt{e}$ exists in a dialogue history $\boldsymbol{x}$, its node embedding (i.e., $\boldsymbol{e}^0$) becomes the mean of corresponding token representations on the PLM encoder $\text{Enc}$. Otherwise, a zero vector is assigned to the initial node embedding. For relation embedding $\boldsymbol{r}^0$, we use the trainable relation embedding matrix. For more justifications on this function, please refer to Appendix D. Also, we experimentally verify that the use of GNN as the triplet embedding yields the better retrieval performance compared to previous PLM-based methods (Humeau et al., 2020; Li et al., 2022), in Table 1 and Figure 4.

### 3.4 INVARIANT GRAPH ENCODING

We then now specify the remaining operation of graph encoding, which determines how to condition the structural graph $\mathcal{Z}$ along with the sequential text sequence $\boldsymbol{x}$ to generate $\boldsymbol{y}$, with regard to the Pre-trained Language Models (PLMs). Let $\boldsymbol{\psi}(\boldsymbol{x}, \mathcal{Z})$ be a graph encoding function, and there are mapping functions $q_e$ and $q_r$, which map the entity and relation in a graph $\mathcal{Z}$ into the natural languages, respectively. Then, the simplest way to encode the graph along the text is to concatenate the mapped tokens of entities and relations in front of the given text input $\boldsymbol{x}$, as in previous works for triplet-conditioned text generation with PLMs (Li et al., 2022; Ma et al., 2022). For instance, given a text $\boldsymbol{x} = [x_1, \dots, x_N]$ and a graph $\mathcal{Z} = \{(\texttt{a}, \texttt{d}, \texttt{b}), (\texttt{b}, \texttt{e}, \texttt{a}), (\texttt{a}, \texttt{d}, \texttt{c})\}$, this naïve graph encoding function is defined as follows: $\boldsymbol{\psi}(\boldsymbol{x}, \mathcal{Z}) = f([a, d, b, b, e, a, a, d, c, x_1, \dots, x_N])$ where $a = q(\texttt{a})$, $d = q(\texttt{d})$, and so on. Also, $f$ is a token embedding function of the PLM. However, this naïve encoding function violates two important invariance properties for encoding a multi-relational graph into the text sequence: permutation invariance (Zaheer et al., 2017) and relation-inversion invariance, where the order of elements matters, which are formalized in Definition 3.1 and 3.2 as follows:

**Definition 3.1.** (**Permutation Invariance**) *For any permutation $\pi \in S_n$, $\boldsymbol{\psi}(\boldsymbol{x}, \mathcal{Z}) = \boldsymbol{\psi}(\boldsymbol{x}, \pi \cdot \mathcal{Z})$, i.e., an order of elements in a subgraph does not affect a representational output.*

**Definition 3.2.** (**Relation Inversion Invariance**) *Let $\neg\texttt{d}$ be an inverse relation to $\texttt{d}$, if $(\texttt{a}, \texttt{d}, \texttt{b}) = (\texttt{b}, \neg\texttt{d}, \texttt{a}) \ \forall a, b \in \mathcal{E}$. Then, $\boldsymbol{\psi}(\boldsymbol{x}, \mathcal{Z} \cup \{(\texttt{a}, \texttt{d}, \texttt{b})\}) = \boldsymbol{\psi}(\boldsymbol{x}, \mathcal{Z} \cup \{(\texttt{b}, \neg\texttt{d}, \texttt{a})\})$ for any graph $\mathcal{Z}$.*

**Invariant Graph Encoding** To satisfy both properties, we consider two additional operations on a set of triplets up to the naïve encoding. We first define a $\text{SORT}$ operator that returns the same output regardless of the order of input set elements, as follows:

$$\text{SORT}(\pi \cdot \mathcal{Z}) = \text{SORT}(\pi' \cdot \mathcal{Z}), \ \forall \pi, \pi' \in S_n, \quad (5)$$

where $S_n$ is a set of all possible permutations for $n$ elements. Moreover, we define a $\text{INV}$ operator that adds the inverse triplet of each triplet in the graph $\mathcal{Z}$, as follows:

$$\text{INV}(\mathcal{Z}) = \mathcal{Z} \cup \{(\texttt{e}_t, \neg\texttt{r}, \texttt{e}_h) \mid (\texttt{e}_h, \texttt{r}, \texttt{e}_t) \in \mathcal{Z}\}. \quad (6)$$

Based on the above $\text{SORT}$ and $\text{INV}$ operations, we can now define a more solid graph encoding function: $\boldsymbol{\psi}(\boldsymbol{x}, \text{SORT}(\text{INV}(\mathcal{Z})))$, which satisfies both permutation and relation inversion invariance.

**Invariant and Efficient Graph Encoding** However, above encoding is not efficient since it requires $\mathcal{O}(n)$ space complexity for encoding a graph with $n$ triplets. Thus, to make it efficient, we newly define $\tilde{\boldsymbol{\psi}}$ that only encodes the unique nodes (entities) along the sequence, formalized as follows:

$$\tilde{\boldsymbol{\psi}}(\boldsymbol{x}, \text{SORT}(\text{ENT}(\mathcal{Z}))) = f([a, b, c, x_1, \dots, x_N]),$$

where $\text{ENT}(\mathcal{Z})$ returns the set of unique entities in $\mathcal{Z}$. This encoding meets both invariance properties but also efficient since it only costs $\mathcal{O}(k)$, for the $k$ entity where $k < n$. However, as it does not consider the relational information in $\mathcal{Z}$, we further perturb the token embeddings for entities in

PLMs with respect to their graph representations in $\mathcal{Z}$. Specifically, for each entity $\mathtt{a} \in \mathtt{ENT}(\mathcal{Z})$, we apply a learnable affine transformation on the token embedding of $\mathtt{a}$ as follows:

$$\boldsymbol{\beta}(f(a), \mathcal{Z}) = (1 + \boldsymbol{\gamma}) * f(a) + \boldsymbol{\delta}, \tag{7}$$
$$\boldsymbol{\gamma} = \mathtt{MLP}_1(\boldsymbol{\eta}), \quad \boldsymbol{\delta} = \mathtt{MLP}_2(\boldsymbol{\eta}), \quad \boldsymbol{\eta} = \mathtt{R\text{-}GNN}(f(a); \mathcal{Z}),$$

where $\mathtt{MLP}$ is a Multi-Layer Perceptron, $\boldsymbol{\beta} : \mathbb{R}^d \to \mathbb{R}^d$ perturbs the embedding according to $\mathcal{Z}$, $\mathtt{R\text{-}GNN}$ is the relation-aware GNN (Schlichtkrull et al., 2018; Vashishth et al., 2020), $f(a)$ is a token embedding of each entity which is used as node embedding for $\mathtt{R\text{-}GNN}$. In sum, we denote a relation-aware and invariant yet efficient encoding $\boldsymbol{\psi}^*$, defined as follows:

$$\boldsymbol{\psi}^*(\boldsymbol{x}, \mathcal{Z}) = \boldsymbol{\beta}(\tilde{\psi}(\boldsymbol{x}, \mathtt{SORT}(\mathtt{ENT}(\mathcal{Z}))), \mathtt{INV}(\mathcal{Z})),$$

where $\boldsymbol{\beta}$ can be applied to any sequence of representational inputs for texts and graphs. We conclude that our graph encoding satisfies both properties, and for proofs, please see Appendix C. For better understanding, we include comprehensive illustration of Equation 7 in Appendix Figure 8.

## 3.5 CONSISTENT GENERATION WITH GRAPH-TEXT CONTRASTIVE LEARNING

Although the previous schemes allow retrieving and encoding subgraphs that are relevant to the input dialogue history, the consistent generation with the given subgraph is further required, when generating responses with the factual knowledge. In other words, the model should be able to generate different sequences when providing different subgraphs, for the same dialogue history.

However, we only access the single ground-truth response regardless of the retrieved knowledge, while the generative model is trained with a teacher forcing. Thus, this setting can give rise to the problem of *exposure bias* (Ranzato et al., 2016): the model is never exposed to other generated tokens during training. To overcome such limitations, we introduce a novel graph-text contrastive learning method motivated by multi-modal contrastive learning (Radford et al., 2021). Formally, for a single pair of a graph and text, the contrastive learning objective is defined as follows:

$$\mathcal{L}_{cont} = \frac{1}{2} \log \frac{\exp(\mathtt{sim}(\zeta(\boldsymbol{z}), \xi(\boldsymbol{h}))/\tau)}{\sum_{\boldsymbol{h}'} \exp(\mathtt{sim}(\zeta(\boldsymbol{z}), \xi(\boldsymbol{h}'))/\tau)} + \frac{1}{2} \log \frac{\exp(\mathtt{sim}(\zeta(\boldsymbol{z}), \xi(\boldsymbol{h}))/\tau)}{\sum_{\boldsymbol{z}'} \exp(\mathtt{sim}(\zeta(\boldsymbol{z}'), \xi(\boldsymbol{h}))/\tau)}, \tag{8}$$

where $\boldsymbol{z} = \frac{1}{m} \sum_{i=1}^m \tilde{\boldsymbol{z}}_i$ is the average encoder representations of the appended knowledge from $\mathtt{Enc}(\boldsymbol{\psi}^*(\boldsymbol{x}, \mathcal{Z})) = [\tilde{\boldsymbol{z}}_1, \ldots, \tilde{\boldsymbol{z}}_m, \boldsymbol{z}_1, \ldots, \boldsymbol{z}_N]$, $\boldsymbol{h} = \frac{1}{T} \sum_{t=1}^T \boldsymbol{h}_t$ is the mean of decoder representations, $\mathtt{sim}$ is the cosine similarity, $\zeta$ and $\xi$ are learnable linear projection layers, and $\tau$ is a learnable temperature parameter. Furthermore, $\sum_{\boldsymbol{h}'}$ and $\sum_{\boldsymbol{z}'}$ indicate the summation over negative samples, which are other texts or graphs within a same mini-batch as in previous literature on contrastive learning (Chen et al., 2020; Lee et al., 2021; Radford et al., 2021). With Equation 8, the model can embed the correlated pairs closer together in order to generate a consistent response to a given graph, i.e., given a different graph, the model would generate different tokens for the same text.

## 3.6 TRAINING

We train the entire model, named as **SU**bgraph **R**etrieval-augmented **GE**neration (**SURGE**), by maximizing the log-likelihood $\log p(\boldsymbol{y}|\boldsymbol{x})$ defined in Equation 2 with respect to parameters of both the retriever $\phi$ and the generator $\theta$.

Since computing the marginal probability over entire subgraphs is infeasible, we approximate it by summing over $k$ sampled subgraphs (Guu et al., 2020; Lewis et al., 2020b). Our end-to-end training objective for retrieval-augmented generation is then defined as follows:

$$\mathcal{L}_{ret} = \log \sum_{\mathcal{Z} \subseteq \Pi} p_\phi(\mathcal{Z}|\boldsymbol{x}) p_\theta(\boldsymbol{y}|\boldsymbol{x}, \mathcal{Z}), \tag{9}$$

where $\Pi = \mathtt{samplek}(p_\phi(\cdot|\boldsymbol{x}))$ denotes sampling $k$ subgraphs over the subgraph distribution and each subgraph sampling is decomposed into sampling $n$ triplets from $p_\phi(z_i|\boldsymbol{x}) \forall i \in [1, n]$ as in subsection 3.3. We further assume that the gold subgraph is partially available in training. Thus, we can utilize the supervised retrieval loss to introduce a semi-supervised retriever learning as follows:

$$\mathcal{L}_{sup} = \log p_\phi(\mathcal{Z}^*|\boldsymbol{x}), \tag{10}$$

where $\mathcal{Z}^*$ is the available ground-truth subgraph. By combining all objectives in Equation 8, 9, and 10, our final training objective is then defined as follows: $\mathcal{L} = \mathcal{L}_{ret} + \mathcal{L}_{sup} + \mathcal{L}_{cont}$.

Figure 3: **KQA.** (Left) An example where multiple responses are acceptable but the gold response cannot reflect all of them. (Middle) We first find the fact from the KG that reflects the relation between entities within the user input and gold response (b), and then search candidate facts from the KG (c). (Right) Corresponding KQA example. If a generated response contains the one of answer candidates, the KQA can predict it (success).

## 4  A NOVEL METRIC: KNOWLEDGE-VERIFYING QA

Existing automatic evaluation metrics, namely BLEU and ROUGE (Papineni et al., 2002; Lin, 2004), are limited in that they only consider the lexical overlaps of words without measuring the factual correctness of the generated responses. As shown in Figure 3 (a), there could be multiple correct responses, but existing metrics score them lower due to the lexical mismatch. To solve this issue, we propose **K**nowledge-verifying **Q**uestion **A**nswering (**KQA**) which measures whether generated responses contain factually correct knowledge given the dialogue history. We formulate extractive QA task (Rajpurkar et al., 2016) by automatically derive QA pairs from the dialogue and the large-scale KG in each dataset (See Figure 3). Then, we fine-tune BERT (Devlin et al., 2019) on synthetic KQA pairs to build QA model. To evaluate generated responses from dialogue generation model, we concatenate the dialogue history and the generated response then forward it into the trained QA model. If the QA model yields the correct answer, we regard this case as the generated response contains accurate knowledge. For more details, see Appendix D.

## 5  EXPERIMENT

### 5.1  EXPERIMENTAL SETUP

We conduct experiments on the **OpendialKG** dataset (Moon et al., 2019), which contains 15K dialogues with 91K utterances associated with a large-scale Knowledge Graph (KG), namely Freebase (Bollacker et al., 2008) with 100k entities and 1M facts. Among them, $49\%$ of the utterances come with the gold knowledge, whereas others are not. We randomly split the dataset into train (70%), validation (15%), and test sets (15%). We also examine **KOMODIS** dataset (Galetzka et al., 2020), which contains 7.5K dialogues associated with much smaller KG with 88k facts. As retrieval candidates, we use 1-hop KG for OpendialKG and 2-hop KG for KOMODIS. Except Figure 2, most of the experiments are on OpendialKG dataset. We use **T5-small** (Raffel et al., 2020) for all experiments for the fair comparison. For more details, see Appendix D.

### 5.2  BASELINES AND OUR MODELS

We compare different variants of our SURGE framework against various KG-augmented dialogue generation models. **No Knowledge.** This model is only provided with the dialog history, thus no external knowledge is used. **All Knowledge.** This model is provided with entire facts within a $k$-hop subgraph of entities associated with the dialog history. **Gold Knowledge.** This model is provided with the exact gold knowledge, even in the test time if the gold knowledge exists. **Space Efficient Encoding.** This model takes all facts from the $k$-hop subgraph of the entities as input. We use two different encoding methods introduced in (Galetzka et al., 2021), namely Space Efficient (series) and Space Efficient (parallel). **EARL.** The latest RNN-based model, where the entities are conditioned in response generation (Zhou et al., 2021). **DiffKG.** Dialogue generative model with differentiable path traversal (Tuan et al., 2022). **Random/Sparse Retrieval.** These models are provided with selected facts from a 1-hop subgraph, via the random sampling or the sparse retrieval – BM25 (Robertson & Zaragoza, 2009). **Dense Retrieval.** This model is a variant of our framework where T5 encoder (Raffel et al., 2020) is used for $d$ in Eq. 4 instead of GNNs similar to Bi-encoder and Poly-encoder (Humeau et al., 2020). **SURGE (unsupervised).** Ours with retrieved context-relevant facts from $k$-hop subgraph, where the retrieval is trained without any supervision. **SURGE (semi-supervised).** Ours but the retriever is trained with supervision if it exists. **SURGE (contrastive).** Ours with both semi-supervised retriever learning and contrastive learning term.

Table 1: Experimental results on OpendialKG dataset with T5-small model. † indicates the model under the incomparable oracle setting, which uses the gold facts even in the test time.

| | Method | KQA | | BLEU | | | | ROUGE | | | Unigram |
|---|---|---|---|---|---|---|---|---|---|---|---|
| | | EM | F1 | B-1 | B-2 | B-3 | B-4 | R-1 | R-2 | R-L | F1 |
| *Baselines* | **No Knowledge** | 12.25 | 20.69 | 15.79 | 9.19 | 5.61 | 3.43 | 19.67 | 7.13 | 19.02 | 22.21 |
| | **All Knowledge** | 43.58 | 50.60 | 15.95 | 9.98 | 6.72 | 4.65 | 20.96 | 8.50 | 20.21 | 24.34 |
| | **Space Efficient** *(series)* | 36.60 | 42.64 | 16.15 | 10.03 | 6.66 | 4.50 | 21.15 | 8.56 | 20.44 | 24.55 |
| | **Space Efficient** *(parallel)* | 38.54 | 44.34 | 16.33 | 10.22 | 6.81 | 4.64 | 21.42 | 8.85 | 20.68 | 24.87 |
| | **EARL** | 32.47 | 35.88 | 11.49 | 6.34 | 4.06 | 2.75 | 15.36 | 4.37 | 14.61 | 16.88 |
| | **DiffKG** | 12.25 | 20.99 | 15.68 | 9.13 | 5.60 | 3.46 | 19.50 | 7.07 | 18.84 | 22.26 |
| *Retrieval variants* | **Random Retrieval** | 31.72 | 38.95 | 15.70 | 9.52 | 6.12 | 3.99 | 20.21 | 7.88 | 19.55 | 23.28 |
| | **Sparse Retrieval** (BM25) | 29.50 | 36.96 | 15.63 | 9.44 | 6.05 | 3.96 | 20.05 | 7.67 | 19.37 | 23.10 |
| | **Dense Retrieval** (Bi-encoder) | 46.17 | 52.52 | 16.67 | 10.44 | 7.05 | 4.91 | 20.41 | 8.38 | 19.66 | 23.85 |
| | **Dense Retrieval** (Poly-encoder) | 46.05 | 52.57 | 17.56 | 11.01 | 7.45 | 5.18 | 20.66 | 8.46 | 19.87 | 24.24 |
| *Ours* | **SURGE** *(unsupervised)* | 48.49 | 55.77 | **17.77** | **11.30** | **7.69** | **5.36** | **21.64** | **9.14** | **20.75** | **25.24** |
| | **SURGE** *(semi-supervised)* | **51.00** | 57.63 | 17.70 | 11.21 | 7.61 | 5.28 | 21.43 | 8.85 | 20.57 | 25.07 |
| | **SURGE** *(contrastive)* | 50.45 | **57.70** | 17.29 | 11.04 | 7.54 | 5.28 | 21.35 | 8.98 | 20.48 | 25.10 |
| *Oracle* | **Gold Knowledge**† | 63.32 | 67.90 | 18.47 | 12.79 | 9.32 | 6.92 | 24.93 | 11.97 | 24.03 | 28.82 |
| | **Gold Response** | 93.30 | 95.21 | 100.0 | 100.0 | 100.0 | 100.0 | 100.0 | 100.0 | 100.0 | 100.0 |

Table 2: Experimental results on KOMODIS dataset with T5-small model. For full experimental results, see Appendix E.

| | KQA | | BLEU | | |
|---|---|---|---|---|---|
| | EM | F1 | B-1 | B-2 | F1 |
| **Random** | 12.41 | 14.17 | 7.74 | 4.02 | 16.29 |
| **SE** *(series)* | 12.41 | 14.70 | 8.34 | 5.13 | 17.37 |
| **SE** *(parallel)* | 16.46 | 18.70 | 9.33 | 5.66 | 17.72 |
| **SURGE** (Ours) | **17.30** | **19.50** | **11.51** | **7.18** | **19.51** |

Table 3: Knowledge-consistent response generation results under the condition where we use the modified gold subgraph instead of retrieved one, to solely evaluate the efficacy of contrastive learning.

| Method | F1 | KF1 |
|---|---|---|
| **SURGE** (unsupervised) | 27.78 | 24.09 |
| **SURGE** (semi-supervised) | **28.30** | 26.38 |
| **SURGE** (contrastive) | 28.17 | **27.58** |

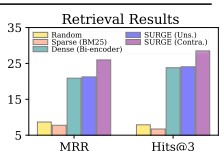

Figure 4: Knowledge retrieval results on the OpendialKG dataset, with metrics of MRR and Hits@3.

## 5.3 EVALUATION METRICS

We evaluate the generated responses using BLEU (Papineni et al., 2002), ROUGE (Lin, 2004) and F1 score with the gold response. Along with these conventional text evaluation metrics, we also evaluate the results with our new metric, KQA (§ 4), which measures whether the generated responses contain proper knowledge. Lastly, we compute the Knowledge F1 (KF1) (Shuster et al., 2021) to measure the unigram overlap between the retrieved knowledge and generated response.

## 5.4 EXPERIMENTAL RESULTS AND ANALYSIS

In Table 1, we report the knowledge-grounded response generation performances of baselines and our SURGE on OpendialKG dataset. As shown in Table 1, our models significantly outperform all the baseline models, excluding oracles, in all evaluation metrics. The high BLEU, ROUGE, and F1 refer that ours sufficiently learns the syntactic and semantic structure of the responses. Our models also achieve high F1 and EM scores in KQA. The high KQA scores indicate that the generated responses are formed with the correct facts, which are relevant to the dialog context. Even the baseline models such as *All Knowledge, Space Efficient Encoding* (Galetzka et al., 2021), *EARL* (Zhou et al., 2021), and *DiffKG* (Tuan et al., 2022), which are provided with all of $k$-hop facts, underperform than ours. The result demonstrates that selecting relevant knowledge is critical in knowledge-augmented response generation. In Figure 2, we additionally report the experimental results on KOMODIS dataset to show applicability of our method to other dataset. Our SURGE (*contrastive*) also outperforms other baselines in KOMODIS dataset. For results with all metrics, please see Table 8 in Appendix E.

**Knowledge Retrieval** Figure 4 shows performances of retrievers, for which we measure the performance on 45% of test dialogues containing the gold knowledge, with Mean Reciprocal Rank (MRR) and Hits@k as metrics. Our models outperform all baselines by large margins since ours has a learnable retriever unlike others. Further, a contrastive learning version of ours, including semi-supervised retriever training, outperforms an unsupervised version. See Appendix G for examples.

**Knowledge-Consistent Generation** We conduct an ablation study on our models to validate the knowledge consistency performance of the response generation by computing the Knowledge F1 (KF1) score (Shuster et al., 2021). To focus solely on the case where a given knowledge is consistently reflected in the generated responses, we use the gold knowledge rather than the retrieved one. We randomly modify the tail entity of each gold knowledge to ensure that responses are generated

| | Context | Gold response | Baseline response | SURGE response |
|---|---|---|---|---|
| (a) | I loved Moby Dick. Can you recommend something similar? | It was written by Herman Melville in 1851. It's sometimes called The Whale. | Moby Dick is a sailor. Do you like her work? | Moby Dick was written by Herman Melville. He also wrote The Whale. |
| (b) | Do you know anything the actor Adam Brown? | Yes, he was in the movie The Hobbit: An Unexpected Journey. | Adam Brown starred in King Kong. Have you seen it? | Adam Brown starred in The Hobbit: The Desolation of Smaug and The Hobbit: The Battle of the Five Armies. |

| (a) Retrieved Subgraph from SURGE | (b) Retrieved Subgraph from SURGE |
|---|---|
| (Moby Dick; or, The Whale, written_by, Herman Melville) | (The Hobbit: The Battle of the Five Armies, starred_actors, Adam Brown) |
| (Moby Dick, written_by, Norman Corwin) | (The Hobbit: An Unexpected Journey, starred_actors, Adam Brown) |
| (Moby Dick, written_by, Ray Bradbury) | (The Hobbit: The Desolation of Smaug, starred_actors, Adam Brown) |

Figure 5: Examples of responses from the baseline (Space Efficient, parallel) and responses from SURGE.

Table 4: Performance comparisons of variants of graph encodings, described in Section 3.4.

| | KQA | | Knowledge |
|---|---|---|---|
| Method | EM | F1 | Length |
| Naïve | 49.27 | 55.23 | 68.21 |
| Invariant | **51.55** | 56.78 | 121.82 |
| Efficient (entity only) | 40.99 | 49.99 | 15.42 |
| Invariant & Efficient | 51.00 | **57.63** | **15.42** |

Table 5: Human evaluation on **Consis**tency, **Info**rmativeness, and **Fluency**. ($p < 0.05$)

| Method | Consis. | Info. | Fluency |
|---|---|---|---|
| **All Knowledge** | **2.52** | 1.99 | 2.62 |
| **Space Efficient** | 2.47 | 1.75 | 2.46 |
| **SURGE** (ours) | **2.71** | **2.39** | **2.92** |

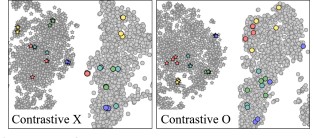
Contrastive X    Contrastive O

Figure 6: Visualization of the embedding space from our graph(star)-text(circle) contrastive learning.

from the given knowledge rather than the trained knowledge. Figure 3 shows that our model with a contrastive learning term outperforms all others in the KF1, implying that the generated responses accurately reflect the encoded knowledge.

**Sensitive Analysis on Graph Encoding**    We further conduct an analysis on graph encoding variants introduced in subsection 3.4. The knowledge length in Figure 4 indicates the average token length used for graph encoding. Our *Invariant & Efficient $\psi^*$* performs the best against other variants, while using the lesser space at the graph encoding phase. Notably, simple *Invariant* achieves a comparable performance against *Invariant & Efficient*, but yields a longer sequence.

**Retrieval and Generation Examples**    Figure 5 shows the examples of generated responses along with the retrieved knowledge. We compare our SURGE against *Space Efficient (parallel)* baseline. In example (a), the baseline response contains an incorrect fact distracted by the contextually irrelevant entity 'sailor'. Contrarily, SURGE successfully retrieves relevant facts from the KG then generates the factually correct response. In example (b), similarly, the baseline generates the response with a wrong fact, meanwhile SURGE retrieves context-relevant facts and generates a informative response.

**Human Evaluation**    We sample 30 responses of SURGE, *All Knowledge*, and *Space Efficient* on the OpendialKG test dataset, then conduct a human study of them. We recruit 46 annotators, and ask them to evaluate the quality of the generated responses by the 3 models given in a random order, with 3 criteria – consistency, informativeness, and fluency – using a 3 point Likert-like scale. As shown in Figure 5, ours obtains significantly (p-value $< 0.05$) higher scores than others in all criteria, which is another evidence that our framework generates consistent, informative, and fluent responses. We also note that the informativeness score and KQA F1 score have a 0.42 Pearson correlation coefficient. This allows us to confirm that our KQA metric positively correlates with the human evaluation results.

**Embedding Space Visualization**    We further visualize the multi-modal graph-text latent space in Figure 6. The visualization shows that, for the same dialogue with different subgraphs, our SURGE with graph-text contrastive learning (right) generates distinct response embeddings pertraining to different subgraphs, unlike the one without graph-text contrastive learning which shows less variety over responses for the same dialogue (left). We include zoomed Figure 6 in the **Appendix**.

## 6    CONCLUSION

We proposed a novel end-to-end framework for knowledge-consistent dialogue generation which retrieves context-relevant subgraph, encodes a subgraph with the text, and generates knowledge-consistent responses, called as **SU**bgraph **R**etrieval-augmented **GE**neration (**SURGE**). Our results demonstrate the effectiveness of our framework in both quantitative and qualitative experiments in knowledge retrieval and response generation tasks. The analysis shows the contribution of each proposed component: retrieval, encoding, and graph-text representation learning. Our work suggests a new direction to generate informative responses for knowledge graph-based dialogue task by empirically showing the importance of retrieving the more relevant subgraph knowledge rather than using all the relevant knowledge graphs when generating knowledge-grounded responses.

## REPRODUCIBILITY STATEMENT

We attach the source code of our SURGE framework in the supplementary file to facilitate the reproducibility of our work. For experimental setups, we provide every details in Appendix D.

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

## A    Discussion on Limitation and Potential Impact

**Limitation**    As briefly discussed in Appendix G, our work is limited in multiple dimensions primarily in terms of dataset, retrieval, and generation. First, the benchmark dataset is limited. Despite the fact that there are several public Knowledge Graph (KG) available (Vrandecic & Krötzsch, 2014; Bollacker et al., 2008), only one dataset (Moon et al., 2019) provides both the diverse set of dialogue and the corresponding large-scale KG. This circumstance may limit the rigorous evaluation of our framework's adaptability in various settings. Future work may study applying our approach for a wider range of dialogue datasets based on Wikipedia (Dinan et al., 2019) by leveraging existing public large-scale KG such as Wikidata (Vrandecic & Krötzsch, 2014). Second, the search space for retrieving context-relevant subgraphs can be expanded. Our SURGE framework now runs on a 1-hop KG that is rooted to entities in the given dialogue history. Finding the entity within the text, on the other hand, necessitates precise named entity extraction and entity linking. Therefore, future work may investigate extending our approach to a framework that can retrieve the context-relevant subgraph among entire KG instead of 1-hop KG. Third, there is still room for improvement in generation quality since we generate knowledge-enhanced responses with a small-scale Pre-trained Language Model (PLM) for efficiency. Such PLMs occasionally fail to generate natural sentences with a high quality (Raffel et al., 2020). Future work could aim to improve generation quality using a small-scale PLM.

**Broader Impact**    Our proposed knowledge-grounded dialogue generation model is essential for designing user-friendly real-world AI systems. Among various types of dialogue generation models, knowledge-grounded dialogue models are trained to interact with users and convey factual information to users in natural languages. Their conversational features can be adapted to any user interfaces that connect the bilateral interaction between human and computer. We believe that the conversational interfaces can enhance the users' experiences and reduce the users' efforts in learning how to use the systems. However, knowledge-grounded dialogue models can become vulnerable to generating offensive, harmful contents or responses with misinformation depending on the users or data. When deploying the models in the real world, in addition to generating realistic responses, they also need to be robust to adversarial feedback from malicious users and biases inherited in pre-training or training corpus, or else they could malfunction. Along with the quantitative and qualitative evaluations on generated responses, it is worthwhile to examine robustness of the dialogue models.

# B NOTATIONS

We organize the notations we used for formally describing our method in Table 6.

Table 6: A list of notations that we used for defining our method.

| | |
|---|---|
| $\mathcal{V}$ | pre-defined vocabulary of tokens for pre-trained language models (text) |
| $\mathcal{E}$ | pre-defined vocabulary of entities (symbol) |
| $\mathcal{R}$ | pre-defined vocabulary of relations (symbol) |
| $\mathtt{a}, \ldots \mathtt{z}$ | knowledge graph symbols written in typewrite font |
| $\boldsymbol{x}$ | input sequence (vector) |
| $x_1, \ldots, x_N$ | input tokens (scalar) |
| $\boldsymbol{y} = [y_1, \ldots, y_T]$ | output sequence and tokens |
| $\mathcal{G}$ | multi-relational graph, such as knowledge graph |
| $\mathcal{Z}$ | retrieved subgraph: $\mathcal{Z} \subset \mathcal{G}$ |
| $z$ | triplet (edge): $z \in \mathcal{Z}$ |
| $q$ | tokenization (mapping) function of KG symbol to sequence of tokens |
| $q(\cdot)$ | text representation function for retrieval |
| $d(\cdot)$ | triplet representation function for retrieval |
| $\mathtt{Enc}$ | Transformer Encoder |
| $\mathtt{Dec}$ | Transformer Decoder |
| $f$ | token (word) embedding function |
| $\theta$ | generator parameter |
| $\phi$ | retriever parameter |
| $\boldsymbol{\psi}$ | set encoding function |
| $\boldsymbol{\beta}$ | perturbation function |
| $\pi$ | set permutation |
| $n$ | the number of triplets in a retrieved subgraph $\mathcal{Z}$ |
| $k$ | the number of samples in a marginalization term |
| $\boldsymbol{z}$ | encoder hidden state (single token) |
| $\boldsymbol{Z}$ | encoder hidden states (sequence of tokens) |
| $\boldsymbol{h}$ | decoder hidden state (single token) |
| $\boldsymbol{H}$ | decoder hidden states (sequence of tokens) |
| $\boldsymbol{X}$ | input embeddings after token embedding function (sequence) |
| $\boldsymbol{Y}$ | output embeddings after token embedding function (sequence) |

# C PROOFS

In this section, we first show that a naïve encoding function $\psi$ in Section 3.4 is neither permutation invariant nor relation inversion invariant, formalized in Proposition C.1. After that, we prove that our invariant and efficient encoding function $\psi^*$ with graph-conditioned token embedding perturbation is both permutation invariant and relation inversion invariant, formalized in Proposition C.2.

**Proposition C.1.** *A naïve encoding function $\psi$ is neither permutation invariant nor relation inversion invariant.*

*Proof.* We prove this by contradiction.

Suppose $\boldsymbol{x} = [x_1, \ldots, x_n]$ and $\mathcal{Z} = \{(\mathtt{a}, \mathtt{d}, \mathtt{b}), (\mathtt{b}, \mathtt{e}, \mathtt{a}), (\mathtt{a}, \mathtt{d}, \mathtt{c})\}$. Moreover, let $\mathcal{Z}' = \{(\mathtt{b}, \mathtt{e}, \mathtt{a}), (\mathtt{a}, \mathtt{d}, \mathtt{b}), (\mathtt{a}, \mathtt{d}, \mathtt{c})\}$ be one of permutations of $\mathcal{Z}$ with the permutation order $\pi = (2, 1, 3)$.

From the definition of naïve encoding, $\boldsymbol{\psi}(\boldsymbol{x}, \mathcal{Z}) = [\boldsymbol{a}, \boldsymbol{d}, \boldsymbol{b}, \boldsymbol{b}, \boldsymbol{e}, \boldsymbol{a}, \boldsymbol{a}, \boldsymbol{d}, \boldsymbol{c}, \boldsymbol{x}_1, \ldots, \boldsymbol{x}_n]$ and $\boldsymbol{\psi}(\boldsymbol{x}, \mathcal{Z}') = [\boldsymbol{b}, \boldsymbol{e}, \boldsymbol{a}, \boldsymbol{a}, \boldsymbol{d}, \boldsymbol{b}, \boldsymbol{a}, \boldsymbol{d}, \boldsymbol{c}, \boldsymbol{x}_1, ..., \boldsymbol{x}_n]$. Therefore, it is easy to notice that $\boldsymbol{\psi}(\boldsymbol{x}, \mathcal{Z}) \neq \boldsymbol{\psi}(\boldsymbol{x}, \mathcal{Z}')$, thus the naïve encoding is not permutation invariant.

We then show naïve encoding is not relation inversion invariant. Suppose $\mathcal{Z}'' = \{(\mathtt{a}, \mathtt{d}, \mathtt{b}), (\mathtt{b}, \mathtt{e}, \mathtt{a}), (\mathtt{c}, \neg\mathtt{d}, \mathtt{a})\}$, where $(\mathtt{a}, \mathtt{d}, \mathtt{c}) \in \mathcal{Z}$ is changed to its inverse relation $(\mathtt{c}, \neg\mathtt{d}, \mathtt{a})$. Then, $\boldsymbol{\psi}(\boldsymbol{x}, \mathcal{Z}'') = [\boldsymbol{a}, \boldsymbol{d}, \boldsymbol{b}, \boldsymbol{b}, \boldsymbol{e}, \boldsymbol{a}, \boldsymbol{c}, \neg\boldsymbol{d}, \boldsymbol{a}, \boldsymbol{x}_1, \ldots, \boldsymbol{x}_n]$ that is different against $\boldsymbol{\psi}(\boldsymbol{x}, \mathcal{Z})$: $\boldsymbol{\psi}(\boldsymbol{x}, \mathcal{Z}) \neq \boldsymbol{\psi}(\boldsymbol{x}, \mathcal{Z}'')$. Therefore, the naïve encoding function is not relation inversion invariant.

In conclusion, from the above two counterexamples, we prove that a naïve encoding function $\psi$ is neither permutation invariant nor relation inversion invariant. □

We now provide proof of the permutation invariance and the relation inversion invariance of our invariant and effective graph encoding $\psi^*$, described in Section 3.4. Before starting the proof, we first revisit the permutation invariant property of graph neural networks that sum, mean and max operators are permutation invariant for the input set of AGGR. Thus, if we use sum, mean, or max for AGGR, then the token embedding perturbation function $\beta$ naturally satisfies the permutation invariance property. In other words, $\beta(\boldsymbol{X}, \mathcal{Z}) = \beta(\boldsymbol{X}, \pi \cdot \mathcal{Z})$, where $\boldsymbol{X} = \tilde{\psi}(\boldsymbol{x}, \text{SORT}(\text{ENT}(\mathcal{Z})))$ for any permutation $\pi$.

**Proposition C.2.** *Invariant and efficient encoding $\psi^*$ is both permutation invariant and relation inversion invariant.*

*Proof.* Suppose $\boldsymbol{x} = [x_1, \ldots, x_n]$ and $\mathcal{Z} = \{(\text{a}, \text{d}, \text{b}), (\text{b}, \text{e}, \text{a}), (\text{a}, \text{d}, \text{c})\}$. We first consider the permutation invariance for any permuted set $\mathcal{Z}' = \pi \cdot \mathcal{Z}$. While $\mathcal{Z}$ and $\mathcal{Z}'$ can have different orders of elements thus the outputs of $\text{ENT}(\mathcal{Z})$ and $\text{ENT}(\mathcal{Z}')$ could be different, we always obtain the same output with the usage of the SORT operator for encoding. In other words, $\text{SORT}(\text{ENT}(\mathcal{Z})) = \text{SORT}(\text{ENT}(\mathcal{Z}'))$ holds due to the definition of the SORT operation in Eq. 5 of the main paper. Therefore, $\tilde{\psi}(\boldsymbol{x}, \text{SORT}(\text{ENT}(\mathcal{Z}))) = \tilde{\psi}(\boldsymbol{x}, \text{SORT}(\text{ENT}(\mathcal{Z}')))$ holds.

Further, since the token embedding perturbation function $\beta(\cdot, \mathcal{Z})$ along with sum, max, or mean in AGGR is also permutation invariant with regards to any permutation on $\mathcal{Z}$, we conclude our invariant and efficient encoding $\psi^*$ is permutation invariant.

We finally prove the relation inversion invariance property of $\psi^*$. Suppose $\mathcal{Z}'' = (\mathcal{Z} \cup t') \setminus t$ where $t \in \mathcal{Z}$ is any triplet in a set and $t'$ is inverse of $t$. Then, $\text{ENT}(\mathcal{Z}) = \text{ENT}(\mathcal{Z}'')$ that is trivial as $\text{ENT}(\mathcal{Z})$ returns the set of only unique nodes in $\mathcal{Z}$. Therefore, $\tilde{\psi}(\boldsymbol{x}, \text{SORT}(\text{ENT}(\mathcal{Z}))) = \tilde{\psi}(\boldsymbol{x}, \text{SORT}(\text{ENT}(\mathcal{Z}'')))$ correspondingly holds.

The remaining step to conclude the proof is to show the following equality: $\beta(\cdot, \text{INV}(\mathcal{Z})) = \beta(\cdot, \text{INV}(\mathcal{Z}''))$, to conclude that $\psi^*(\boldsymbol{x}, \mathcal{Z}) = \psi^*(\boldsymbol{x}, \mathcal{Z}'')$ from $\beta(\tilde{\psi}(\boldsymbol{x}, \text{SORT}(\text{ENT}(\mathcal{Z}))), \text{INV}(\mathcal{Z})) = \beta(\tilde{\psi}(\boldsymbol{x}, \text{SORT}(\text{ENT}(\mathcal{Z}''))), \text{INV}(\mathcal{Z}''))$. We note that $\text{INV}(\mathcal{Z}) = \text{INV}(\mathcal{Z}'')$, as INV makes any graph as bidirectional one by the definition in Eq. 6 of the main paper. Therefore, $\beta(\cdot, \text{INV}(\mathcal{Z})) = \beta(\cdot, \text{INV}(\mathcal{Z}''))$ holds, and the relation inversion invariance property of $\psi^*$ holds.

□

## D    EXPERIMENTAL SETUP

In this section, we introduce the detailed experimental setups for our models and baselines. Specifically, we describe the details on implementation, dataset, training and model in the following subsections of D.1, D.2, D.3 and D.4, one by one.

### D.1    IMPLEMENTATION DETAILS

We use the T5-small (Raffel et al., 2020) as the base Pre-trained Language Model (PLM) for all experiments. For the pre-trained checkpoint, we use the version that the authors released. For all implementations, we use Pytorch (Paszke et al., 2019). To easily implement the language model, we use the huggingface transformers library (Wolf et al., 2020).

**Retriever Details**    In this paragraph, we describe the implementation details of our context-relevant subgraph retriever, including the triplet embedding and dialogue context embedding for the retriever.

For the dialogue history embedding function $q$, we use the existing pre-trained language model (PLM). Specifically, we use the encoder part of the T5-small model (Raffel et al., 2020) and freeze the parameters of it not to be trained. We then instead add a Multi-Layer Perceptron (MLP) on top of it, to give a point-wise attention (Bahdanau et al., 2015) to each token, whereby all tokens are not

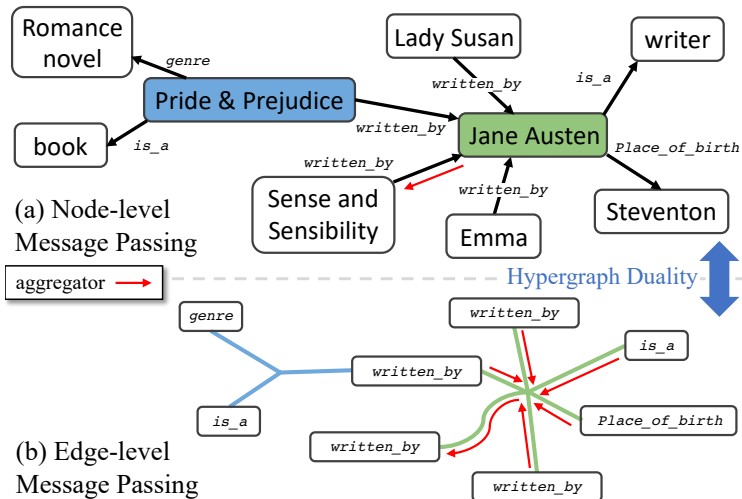

Figure 7: **GNN-based Triplet Representation for Retrieval.** To represent each triplet with regards to its graph structure, we use the message passing on both nodes and edges. (a) Node-level Message Passing. To represent the entity *Sense and Sensibility*, the message from its neighbors – the entity *Jane Austen* – is aggregated. (b) Edge-level Message Passing. To represent the relation written_by, the messages from relations associated to a green hyperedge are aggregated. We do not draw self-loops and inverse edges for simplicity.

equally considered in the sentence encoding. Formally,

$$q(\boldsymbol{x}) = \sum_{i=1}^{n} \alpha_i * \boldsymbol{z}_i, \qquad \boldsymbol{Z} = [\boldsymbol{z}_1, \ldots, \boldsymbol{z}_n] = \texttt{Enc}(\boldsymbol{X}), \qquad \alpha_i = \frac{\exp(\texttt{MLP}(\boldsymbol{z}_i))}{\sum_{j=1}^{n} \exp(\texttt{MLP}(\boldsymbol{z}_j))} \, \forall i$$

where $\alpha_i$ is a scalar, and MLP is a Multi-Layer Perceptron consisting of two linear layers and ReLU nonlinearity.

For obtaining triplet representations, we need to embed the entity (node) and relation (edge) into the latent space. Similar to the token embedding matrix used in PLMs, we can introduce the entity and relation embedding matrices. However, since the number of entities used in Freebase of OpendialKG (Moon et al., 2019) is too large compared to the number of tokens in T5 (100,814 vs 32,000) (Raffel et al., 2020), it is inefficient to introduce the trainable entity embedding matrix for the retriever. Furthermore, the use of standalone entity embedding matrix might be sub-optimal in terms of generalization since there is no evidence that all entities in a large-scale KG emerge in training dataset.

Thus, we instead reuse the contextualized representation from the PLM encoder, to embed each node if the corresponding entity exists in the dialogue context. Formally, suppose that there is a triplet $\{(\mathsf{e}_h, \mathsf{r}, \mathsf{e}_t)\}$ in the 1-hop subgraph $\mathcal{G}$, which satisfies the following condition: $q(\mathsf{e}_h) \subseteq \boldsymbol{x}$ or $q(\mathsf{e}_t) \subseteq \boldsymbol{x}$. If so, we can know the position of the mapped entity within the dialogue history: $[x_{start}, ..., x_{end}] = q(\mathsf{e}_h)$ from $q(\mathsf{e}_h) \subseteq \boldsymbol{x}$. Therefore, the node embedding for the entity $\mathsf{e}_h$ is obtained by $\texttt{EntEmb}(\mathsf{e}_h) = \frac{1}{|q(\mathsf{e}_h)|} \sum_{i=start}^{end} \texttt{Enc}(\boldsymbol{X})_i$ iff $q(\mathsf{e}_h) \subseteq \boldsymbol{x}$. If the entity mention does not exist in the dialogue history, we use the zero vector as the node embedding. For edge embedding, we use the trainable relation embedding matrix $\boldsymbol{R} \in \mathbb{R}^{|\mathcal{R}| \times 128}$ to represent the edge, since the number of relations is relatively small (1,357).

With our node and edge representations, we now focus on representing the triplet in Eq. 4 of the main paper for its retrieval. In particular, we use the Graph Neural Networks (GNNs) for encoding triplets, where we obtain the node representations from the Graph Convolutional Network (GCN) (Kipf & Welling, 2017) that is a widely used architecture for representing the nodes with respect to their graph structures. However, for representing the edges, we use the Edge Hypergraph Graph Neural Network (EHGNN) used in Jo et al. (2021), due to its simplicity but effectiveness for edge representations. We summarize our triplet representation in Figure 7.

**Graph Encoding Details** In this paragraph, we describe the implementation details of the token embedding perturbation function $\boldsymbol{\beta}$ used in our *Invariant and Efficient* graph encoding introduced

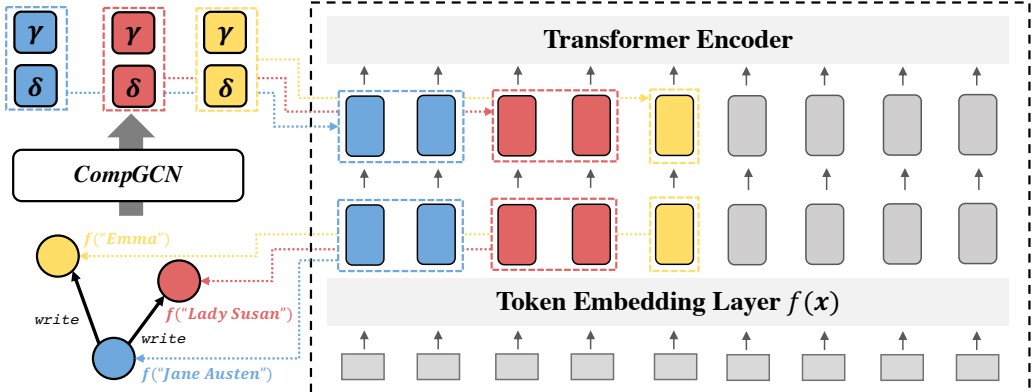

**Figure 8: Comprehensive diagram for Invariant and Efficient graph encoding.** Our proposed graph encoding first concats the sorted list of entities in front of the dialogue history. Then, we form the learnable affine transformation $\gamma, \delta$ for each entity using relation-aware GNN such as CompGCN (Vashishth et al., 2020).

in Section 3.4. To be aware of the relation of the graph over GNNs, we use the simplified version of CompGCN (Vashishth et al., 2020). For architectural details, instead of using the different linear layers to distinguish the inverse relation from its opposite relation, we use the same linear layer. Also, we use subtraction as the specific composition operator for reflecting relations in CompGCN.

Then, we form the learnable affine transformation based on the aggregated representation from GNN layers, to perturb the token embeddings with respect to their graph information as in Equation 7 of the main paper. In particular,

$$\boldsymbol{\eta} = \texttt{R-GNN}(f(a); \mathcal{Z}) = \texttt{UPD}(f(a), \texttt{AGGR}(\{f(b), \texttt{r} \mid \forall \texttt{b} \in \mathcal{N}(\texttt{a}; \mathcal{Z})\})),$$
$$\boldsymbol{\gamma} = \texttt{MLP}_1(\boldsymbol{\eta}), \quad \boldsymbol{\delta} = \texttt{MLP}_2(\boldsymbol{\eta}), \quad \boldsymbol{\beta}(f(a), \mathcal{Z}) = (\mathbf{1} + \boldsymbol{\gamma}) * f(a) + \boldsymbol{\delta},$$

where $\texttt{MLP}_1$ and $\texttt{MLP}_2$ are learnable MLPs consisting of two linear layers with ReLU nonlinearity. In Figure 8, we illustrate comprehensive diagram of Equation 7, which enables our *Invariant and Efficient* graph encoding to understand the structure of the retrieved subgraph $\mathcal{Z}$.

**Contrastive Learning Details**  For contrastive learning, we initialize $\tau$ in Equation 8 as 0.01.

**KQA Details**  In this paragraph, we describe the implementation details for our Knowledge-verifying Question Answering (KQA) introduced in section 4. For building the QA dataset, we first gather the dialogue sessions where the gold response contains the entity from the whole OpendialKG dataset. Then, we extract the triplet from the given whole KG where the head entity is placed within the dialogue history and the tail entity is placed within the gold response. We build a QA training dataset based on the extracted triplets and a corresponding dialogue session. To diversify the training data, we replace the tail entity of each triplet with plausible candidate entities within KG and change the entity in the response following the changed entity on the triplet. As a result, we obtain the QA dataset size of 200k. We train the BERT-base (Devlin et al., 2019) with the constructed QA dataset. We hold out 10% of data for validation and obtain the fine-tuned BERT model with 88.89 F1 score on the hold-out validation set. When we apply the fine-tuned QA model on the evaluation of the generated responses, we rebuild the QA evaluation set with the generated response instead of a gold response as illustrated in Figure 3 of the main paper.

## D.2 DATASET DETAILS

We mainly conduct experiments on **OpendialKG** (Moon et al., 2019), which provides the parallel dialogue corpus corresponding to the existing large-scale Knowledge Graph (KG) named Freebase (Bollacker et al., 2008). The provided large-scale KG consists of total 1,190,658 fact triplets over 100,813 entities and 1,358 relations. This dataset is collected from 15K human-to-human role-playing dialogues, having multi-turns, from which we pre-process that each assistance response is the label and its corresponding dialogue history is the input. Although some of the data contain

the gold knowledge that is useful for generating the response on the ongoing conversation, we found that $51\%$ of data has no gold knowledge. To overcome this limitation, we additionally find entities from the dialogue history using the Named Entity Recognition module in spaCy[1], and then include the extracted entities' corresponding triplets in the KG to the dataset. For entity linking, we use the exact match. Since the dataset does not provide the pre-defined data split, we randomly split sessions into train (70%), validation (15%), and test sets (15%). We also conduct experiments on **KOMODIS** (Galetzka et al., 2020) dataset and follows the same preprocessing as in OpendialKG dataset.

### D.3 TRAINING DETAILS

All experiments are constrained to be done with a single 48GB Quadro 8000 GPU. SURGE training needs 12 GPU hours. For all experiments, we select the best checkpoint on the validation set. We fine-tune the SURGE for 30 epochs on the training set, where we set the learning rate as 1e-4, weight decay as 0.01, learning rate decay warmup rate as 0.06, maximum sequence length for dialogue history as 256, maximum sequence length for knowledge as 128, and batch size as 24. For retrieval, we use the subgraph size $n$ as 3, and sample size $k$ for marginalization as 4. We use the AdamW (Loshchilov & Hutter, 2019) optimizer for training. For fair evaluation, we apply the same training setting to all baselines if applicable.

### D.4 MODEL DETAILS

In this subsection, we describe the details of baselines and our models used in our experiments, as follows:

1. **No Knowledge**: This model is provided with only the dialog history. No knowledge is used to generate responses.
2. **Gold Knowledge**: This model is provided with the dialogue history along with its exact gold knowledge for the gold response. Thus, since this model uses such gold knowledge, we expect the results of it as the upper bound of the task.
3. **Space Efficient (series)**: This model is provided with all the knowledge which are related to the entities that appeared in the dialogue history (Galetzka et al., 2021), by matching the entities in the dialogue history and the entities in the KG. In particular, this model encodes the entities and their relations explicitly in the words in the encoder part.
4. **Space Efficient (parallel)**: This model is mostly the same as the above model – space Efficient (series) – except the knowledge encoding part. Specifically, it encodes the entities in the words like the above, whereas, encoding the relation between entities in the segmentation block of the entities Galetzka et al. (2021).
5. **EARL**: This model uses the RNN-based encoder-decoder architecture with the entity-agnostic representation learning (Zhou et al., 2021), with all the provided knowledge associated with the entities in the dialogue history. Specifically, this model first calculates the probability of words obtained by encoding the entities in the KG, and then uses such probabilities to generate a word in the decoding phase.
6. **DiffKG**: This model (Tuan et al., 2022) uses a differentiable path reasoning, which is jointly trainable along with the dialogue generation. After the path reasoning, the entities in the reasoning path are naively appended in front of the dialogue history, then concatenated input is forwarded to the pre-trained language model.
7. **Random Retrieval**: This model is provided with entire facts from k-hop subgraphs of entities that appeared in the dialogue history. However, instead of encoding all the knowledge in one-hop subgraph as in Space Efficient, this model randomly samples them, which are then used for generating responses.
8. **Sparse Retrieval** (BM25): This model is also provided with entire facts from k-hop subgraphs of entities. To sample relevant facts to the dialogue history among the entire facts, this model uses BM25 (Robertson & Zaragoza, 2009) that is a sparse retrieval model. To be specific, let assume we have a dialogue history and its corresponding facts from k-hop subgraphs of matched entities. Then, to run BM25, we first concatenate components of each fact consisting of two entities and

---

[1]https://spacy.io/

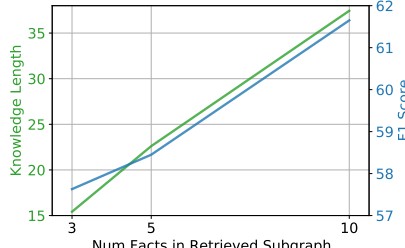

| Method | MRR | Hits@1 | Hits@3 | Hits@5 | Hits@10 | Hits@100 |
|---|---|---|---|---|---|---|
| **Random Retrieval** | 8.67 | 3.11 | 7.89 | 10.86 | 17.84 | 66.22 |
| **Sparse Retrieval (BM25)** | 7.76 | 2.61 | 6.72 | 9.89 | 16.67 | 61.21 |
| **Dense Retrieval (Bi-encoder)** | 20.89 | 11.99 | 23.79 | 30.34 | 39.26 | 67.26 |
| **Dense Retrieval (Poly-encoder)** | 21.47 | 12.13 | 24.12 | 31.51 | 41.43 | 68.03 |
| **SURGE** (unsupervised) | 21.24 | 10.46 | 24.06 | 31.97 | 44.17 | 74.74 |
| **SURGE** (semi-supervised) | 22.53 | 13.40 | 24.79 | 31.84 | 42.37 | 69.33 |
| **SURGE** (contrastive) | **25.98** | **16.67** | **28.50** | **35.72** | **46.11** | **74.31** |

Figure 9: (Left:) Performances of our SURGE by varying the number of facts for retrieving the subgraph (i.e., varying the number of triplets in the subgraph) from three, to five, to ten, with the length of sequence for knowledge (knowledge length) and F1 scores of KQA as evaluation metrics. (Right:) We additionally report the knowledge retrieval performances, with MRR and Hits@K as evaluation metrics.

Table 7: Experimental results on OpendialKG dataset with **BART-base**.

| Method | KQA | | BLEU | | | | ROUGE | | | Unigram |
|---|---|---|---|---|---|---|---|---|---|---|
| | EM | F1 | B-1 | B-2 | B-3 | B-4 | R-1 | R-2 | R-L | F1 |
| **No Knowledge** *(BART-base)* | 31.17 | 37.54 | 17.38 | 10.79 | 7.16 | 4.81 | 20.64 | 8.22 | 19.92 | 24.36 |
| **Space Efficient** *(BART-base, Series)* | 48.49 | 53.83 | 18.56 | 11.85 | 8.01 | 5.56 | 22.36 | 9.43 | 21.48 | 26.38 |
| **Space Efficient** *(BART-base, Parallel)* | 49.80 | 55.06 | **18.90** | **12.19** | **8.35** | **5.81** | 22.63 | 9.79 | 21.76 | **26.79** |
| **SURGE** *(BART-base, semi-supervised, $n = 10$)* | 50.84 | 57.35 | 17.80 | 11.12 | 7.48 | 5.18 | 18.64 | 7.27 | 17.77 | 22.07 |
| **SURGE** *(T5-small, semi-supervised, $n = 3$)* | 51.32 | 58.45 | 17.63 | 11.28 | 7.41 | 5.39 | 21.74 | 9.18 | 20.85 | 25.57 |
| **SURGE** *(T5-small, semi-supervised, $n = 10$)* | **54.50** | **61.65** | 17.70 | 11.37 | 7.81 | 5.50 | 21.55 | 9.09 | 20.65 | 25.44 |

one relation, and tokenize the dialogue history and the facts for obtaining corpus and queries, respectively, for BM25. After that, BM25 calculates the lexical overlapping score between the dialogue context (corpus) and the one-hop fact (query), from which we use the relevant facts having top-$k$ scores by BM25.

9. **Dense Retrieval** (Bi-encoder, Poly-encoder): This model uses a pre-trained language model for the triplet embedding of the retriever instead of using GNN. Specifically, we consider each triplet as a single sentence (e.g, (Jane Austen, write, Susan) → "Jane Austen write Susan") and embed them with the pre-trained language model. For scoring, we use both bi-encoder and poly-encoder architectures (Humeau et al., 2020).

10. **SURGE (unsupervised)**: Our basic subgraph retrieval-augmented generation framework that is provided with entire facts from k-hop subgraphs of entities. In particular, this model trains the structure-aware subgraph retriever without any guidance of the gold knowledge (i.e., ground truth knowledge for the dialogue history is not given). In other words, for the given dialogue context, this model implicitly learns to retrieve the context-relevant knowledge, and then generates the response with the retrieved knowledge.

11. **SURGE (semi-supervised)**: Our subgraph retrieval-augmented generation framework with semi-supervised learning of graph retrieval, with provided entire facts from k-hop subgraphs of entities. Unlike the unsupervised version of SURGE, this model trains the retriever to select the gold knowledge if the dialogue context has such knowledge during training.

12. **SURGE (contrastive)**: Our full subgraph retrieval-augmented generation framework with the contrastive learning of graph-text modalities as well as the semi-supervised learning of graph retrieval, with provided entire facts from k-hop subgraphs of entities. Unlike aforementioned frameworks of ours, this additionally enforces the model to faithfully reflect the retrieved knowledge in the input, to the generated response with contrastive learning.

# E  ADDITIONAL EXPERIMENTS

## E.1  VARYING THE NUMBER OF FACTS IN SUBGRAPHS

We experiment our SURGE framework with varying the number of facts in retrieval, which are then used in our graph encoding function to condition the encoded graph information for response generation. Specifically, in Figure 9, we report the length of sequence for knowledge (knowledge length) and F1 scores measured by our KQA for our SURGE framework, with different numbers

Table 8: Experimental results on KOMODIS dataset with T5-small.

| Method | KQA | | BLEU | | | | ROUGE | | | Unigram |
|---|---|---|---|---|---|---|---|---|---|---|
| | EM | F1 | B-1 | B-2 | B-3 | B-4 | R-1 | R-2 | R-L | F1 |
| **Random** | 12.41 | 14.17 | 7.74 | 4.02 | 2.46 | 1.68 | 21.79 | 4.00 | 21.44 | 16.29 |
| **Space Efficient** *(Series)* | 12.41 | 14.70 | 8.34 | 5.13 | 3.77 | 3.05 | 22.36 | 4.85 | 22.06 | 17.37 |
| **Space Efficient** *(Parallel)* | 16.46 | 18.70 | 9.33 | 5.66 | 4.06 | 3.20 | 22.80 | 4.12 | 22.47 | 17.72 |
| **SURGE** *(unsupervised)* | 16.18 | 18.51 | 11.46 | 7.10 | 5.15 | 4.07 | 23.49 | 5.77 | 23.09 | 18.70 |
| **SURGE** *(semi-supervised)* | 16.62 | 19.48 | 11.28 | 6.98 | 5.05 | 3.98 | 23.58 | 5.79 | 23.21 | 18.68 |
| **SURGE** *(contrastive)* | **17.30** | **19.50** | **11.51** | **7.18** | **5.20** | **4.10** | **24.13** | **6.17** | **23.74** | **19.51** |

of facts within a retrieved subgraph: $n = [3, 5, 10]$. Note that, in this experiment, we only use the semi-supervised model without the contrastive loss. We expect that the performance of our SURGE will increase as we increase the number of facts within the retrieved subgraph, since the model can leverage more numbers of knowledge for response generation. As shown in Figure 9, we observe the significant performance improvements on using ten facts against using three and five facts, while the performance difference between the three and five is marginal. We suggest that this result should be interpreted with the retrieval results on the right side of Figure 9, where about 40% of retrieved subgraphs including the ten different facts contain at least one necessary knowledge, thus the generation performance is boosted according to the improvement in retrieval.

### E.2 DISCUSSIONS ON USING LARGER PLMS

Notably, we observe that the use of larger Pre-trained Language Models (PLMs) – three times more number of parameters compared to T5-small that we use – does not result in better performance for the knowledge-grounded dialogue task. Specifically, in Table 7, we report the experimental results of selected baselines and our SURGE semi-supervised model with BART-base (Lewis et al., 2020a) as the base PLM. We want to clarify that the BART-base model has 220M parameters, which is about **three times larger** than the number of parameters of the T5-small model (60M).

We first observe that BART-base shows decent performance without any knowledge (No Knowledge) compared to the no-knowledge case of T5-small, verifying that the larger PLM generally contains more factual knowledge within its pre-trained parameters. Moreover, BART-base obtains higher scores in the simple word overlap metrics such as BLEU (Papineni et al., 2002) and ROUGE (Lin, 2004), whose results further confirm that a larger PLM can generate more natural or syntactically better sentences than the smaller one, thanks to its parameter size.

On the other hand, we find that BART-base is less suffered from the irrelevant knowledge issue (i.e., conditioning irrelevant knowledge for the given context when generating responses) than T5-small, therefore, the performance of *Space Efficient Encoding* on KQA is quite high. However, the use of BART-base does not result in significant improvement on the KQA metric for our SURGE framework. Moreover, ours with T5-small shows better performance than ours with BART-base in terms of KQA scores, when the number of facts within the retrieved subgraph is 10: $n = 10$. This result suggests that the quality of the generated response – having relevant knowledge to the given context – might depend on the performance of the subgraph retriever whose goal is to retrieve the context-relevant knowledge, rather than the inherent performance of PLMs.

### E.3 FULL EXPERIMENTAL RESULT ON KOMODIS

In the main paper, we mostly focus on OpendialKG dataset (Moon et al., 2019), since it is the largest and most realistic public datasets that provides both dialogues across diverse domains and corresponding large-scale Knowledge Graph (KG) (Bollacker et al., 2008). To verify the effectiveness of our SURGE framework, the existence of the large-scale KG and the importance of relevant fact searching is important since we focus on the real-world scenario where the response generation requires the relevant fact acquirement from the large-scale KG.

However, one can raise the question regarding the versatility of our method on other datasets. To alleviate the issue, we conduct additional experiments on another dataset named KOMODIS (Galetzka et al., 2020), which is also KG-based dialogue dataset. Compared to OpendialKG, KOMODIS does

Table 9: (Left:) Performance evaluation with the diversity metric named Distinct. (Right:) Ablation study results on GNN variants in our modules.

| Method | Dist-1 | Dist-2 |
|---|---|---|
| No Knowledge | 6.06 | 15.73 |
| All Knowledge | 9.67 | 24.45 |
| SEE (Series) | 8.49 | 21.77 |
| SEE (Parallel) | 8.78 | 22.70 |
| EARL | 5.15 | 16.46 |
| Sparse Retrieval (BM25) | 7.65 | 19.63 |
| SURGE (semi-supervised) | 10.18 | 27.85 |

| Method | KQA | | BLEU | | | | ROUGE | | | Unigram |
|---|---|---|---|---|---|---|---|---|---|---|
| | EM | F1 | B-1 | B-2 | B-3 | B-4 | R-1 | R-2 | R-L | F1 |
| Eq 4. GCN $\rightarrow$ GAT | 49.16 | 56.10 | 17.42 | 10.96 | 7.39 | 5.17 | 21.10 | 8.65 | 20.25 | 24.79 |
| Eq 7. CompGCN $\rightarrow$ GCN | 48.61 | 55.53 | 17.48 | 10.97 | 7.34 | 5.05 | 21.23 | 8.73 | 20.37 | 24.77 |
| **SURGE** *(semi-supervised)* | **51.00** | **57.63** | **17.70** | **11.21** | **7.61** | **5.28** | **21.43** | **8.85** | **20.57** | **25.07** |

not provide the corresponding large-scale KG and most of responses do not require the knowledge. Therefore, we only measure the automatic evaluation to evaluate the performance of each method on KOMODIS dataset. In Table 8, we present the experimental results on the KOMODIS dataset. Results obviously show that our SURGE framework shows superior performance against baselines on the additional dataset. Therefore, we can conclude that our method can generalize to other datasets beyond the opendialKG dataset.

### E.4 DIVERSITY EVALUATION

In the main paper, we evaluate model generation performance primarily on its quality. We measure the distinct metric (Li et al., 2016), which is one of the most popular metrics for evaluating the diversity of the generative model, to evaluate the performance of each model in more diverse aspects. In Figure 9 left, we report the performance of baselines and our models in distinct metric. Our SURGE framework generates more diverse responses than all other baselines, according to the results.

### E.5 ABLATIONS STUDIES ON GNN DESIGN CHOICES

We use two different types of Graph Neural Networks (GNN) in our SURGE framework. One is the Graph Convolutional Network (GCN) (Kipf & Welling, 2017), which is used to embed each node entity on the entire 1-hop subgraph in the triplet embedding function $d$ of the main paper Equation 4. Another is Composition-Based Multi-Relational Graph Convolutional Networks (CompGCN) (Vashishth et al., 2020), which is used to embed each entity by considering the relations between entities in the token embedding perturbation function $\beta$ of the main paper Equation 7. In this subsection, we conduct ablation studies on both GNN design choices. First of all, we replace the GCN in Equation 4 with Graph Attention Network (GAT) (Velickovic et al., 2018) to validate the effect of the GNN design choices on the node embedding in the triplet embedding function. Then, we run experiments by changing CompGCN in Equation 7 to GCN to see how important the relationships are in the graph encoding. We present the results on Figure 9 right. Results indicate that the use of GAT in Equation 4 does not have any impact on the performance a lot. However, the use of relation-aware GNN is highly important in effective and efficient graph encoding, since removing the relation awareness of GNN reduces the performance of our model a lot.

### F HUMAN EVALUATION

In this section, we describe the details of human evaluation used in section 5 of the main paper. We request the annotators to evaluate the responses generated from two baselines (i.e., ALL Knowledge and Space Efficient) and our SURGE framework in response to the given dialogue context, according to three criteria – consistency, informativeness, and fluency. Figure 10 is the instructions provided to each annotator. Specifically, regarding the consistency metric, we ask annotators to check whether the generated response makes sense in the context of the conversation. For informativeness, we ask annotators to check whether the response contains correct and enough information, whereby experiment participants are recommended to use the internet search, to check whether the response contains correct facts. In addition to this, we also provide the dialogue-related facts from Freebase as a reference for fact checking for annotators. For fluency, we ask annotators to check whether the response is grammatically correct and naturally sound.

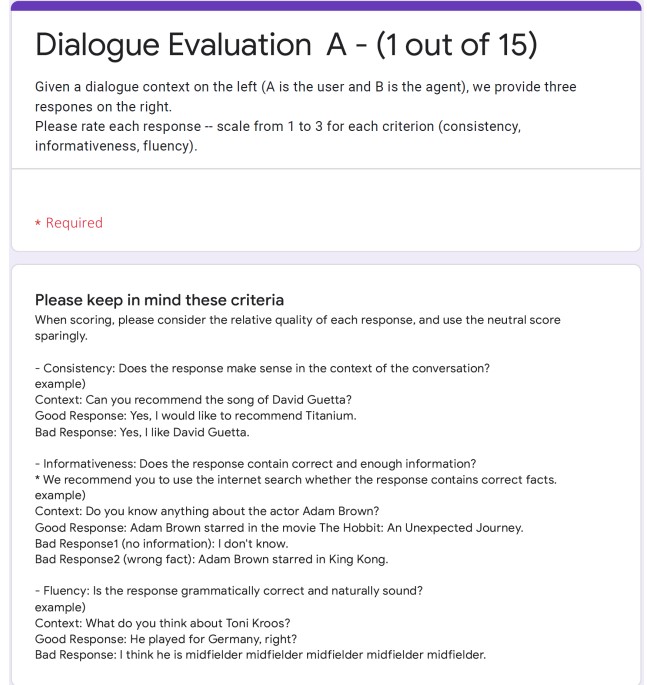

Figure 10: **Human Evaluation Instructions.** To measure the qualitative performances of the generated responses, annotators are provided with the following instruction on three criteria – consistency, informativeness, and fluency.

## G  RETRIEVAL AND GENERATION EXAMPLES

In this section, we provide the examples for knowledge retrieval and response generation, for the given dialogue history.

**Embeeding Space Visualization**   In Figure 11, we present a larger version of Figure 6 in the main paper. Specifically, we embed the hidden representations before the projection layer for each graph (star) and the embedding of the generated text (circle) through the dimensionality reduction using t-SNE (van der Maaten & Hinton, 2008). As mentioned in the main paper, the visualization highlights that our SURGE framework with graph-text contrastive learning generates more distinct responses to different subgraphs, unlike the one without graph-text contrastive learning which shows less variety over responses even with different graphs.

**Retrieval Examples**   We provide the retrieval examples of various models, such as random retrieval, sparse retrieval and our SURGE models. In particular, in the first (top) example of Figure 12, we are given a dialogue context in regard to books for Richard Maxwell, and baselines including random and BM25 retrievers select the facts associated to the entity Richard Maxwell, which are but irrelevant to the ongoing conversion, for example, (Richard maxwell, is-a Theatre director). Also, as shown in the second (bottom) example of Figure 12, we observe that the simple term-based matching model (i.e., BM25) cannot contextualize the current and previous dialogues, but retrieves the facts associated to frequent words, for example, song, which are less meaningful for the user's question. In contrast to baselines, as our SURGE framework trains a retriever in an end-to-end fashion, it first contextualizes the given dialogue context, and then accurately retrieves relevant knowledge.

**Generation Examples**   We provide the generation examples from our model. To be specific, we provide the dialogue context along with its corresponding retrieved subgraph and generated response obtained from our SURGE framework. In Figure 13 and Figure 14, we provide the correct examples: our model retrieves a context-relevant subgraph, but also generates a factual response from retrieved knowledge. On the other hand, in Figure 15, we provide the failure cases. In particular, as shown in the first row of Figure 15, the fact in the knowledge graph could be ambiguous or inaccurate, as it defines the release year of the book – Wicked – as both 2008 and 2014. Moreover, we further provide

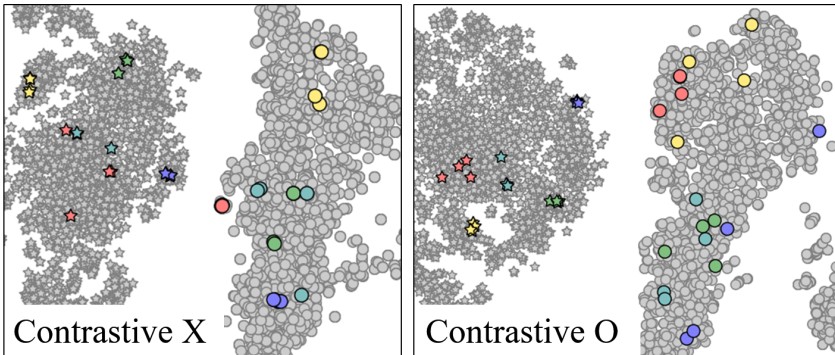

Figure 11: Large version of Figure 6 in the main paper. Stars indicate the embedding of graph and circles indicate the embedding of decoder hidden states (text), respectively.

the failure example on retrieval in the second row of Figure 15, where the user asks about the Bourne Legacy, while the dialogue agents retrieve the irrelevant knowledge to the question. Finally, we show the common problem in PLMs in the last row of Figure 15, where the generative model repeats the meaningless words at the end, while the retriever correctly selects the relevant knowledge.

**Dialogue Context**

A: Could you recommend any books written by Richard Maxwell?

**Gold Knowledge**

Richard maxwell, ~written_by, a tale of two cities

**Random Knowledge**

Richard maxwell, sibling, jan maxwell

Screenwriter, ~is-a, Richard maxwell

Theatre director, ~is-a, Richard maxwell

**BM25 Knowledge**

Richard maxwell, is-a, Theatre director

Screenwriter, ~is-a, Richard maxwell

Richard maxwell, organization founded, new york city players

**Our Knowledge**

Richard maxwell, ~written_by, a tale of two cities

Richard maxwell, sibling, Jan maxwell

**Dialogue Context**

A: I like Adam Levine.
B: OMG me too! I love that song Moves Like Jagger.
A: Yes, Love that too. It is really fun. Can you tell me more.
B: Did you know it's considered a power pop song?
A: No, I did'n. Do you know Love the way you Lie?

**Gold Knowledge**

Song, ~kind of composition, Love the way you lie

Love the way you lie, composer, Eminem

**Random Knowledge**

Blue monday, kind of composition, Song

The look of love, kind of composition, Song

Bad romance, kind of composition, Song

**BM25 Knowledge**

Song, ~kind of composition, This charming man

Behati prinsloo, ~spouse (or domestic partner), Adam levine

Song, ~kind of composition, safe & sound

**Our Knowledge**

g minor, ~key, Love the way you lie

Eminem, ~composer, Love the way you lie

Skylar grey, ~composer, Love the way you lie

Figure 12: Examples of the dialogue history with its corresponding gold knowledge as well as the retrieved knowledge from random retrieval and sparse retrieval baselines and from our SURGE framework. The retrieved fact is represented as the format of (head, relation, tail), where ∼symbol in the front of relation (i.e., ∼relation) in the retrieved knowledge denotes the inverse relation.

---

**Dialogue Context**

A: Do you like Rob Marshall? My friends have recommend his films.

---

**Retrieved Knowledge**

Nine, has_genre, Drama

Pirates of the Caribbean: On Stranger Tides, directed_by, Rob Marshall

Memoirs of a Geisha, directed_by, Rob Marshall

---

**Generated Response**

B: Yes, he wrote Pirates of the Caribbean: On Stranger Tides, Memoirs of a Geisha Nine and many others. Have you seen either of those?

---

**Dialogue Context**

A: Are there any works by Chris Cleave, which you can recommend me?
B: Certainly, wrote The Other Hand, Incendiary, and Old. Have you read them?
A: No, I haven't read The Other Hand out of the ones you mentioned. What genre is it?

---

**Retrieved Knowledge**

The Other Hand, release_year, 2008

2008, ~release_year, The Other Hand

The Other Hand, has_genre, Literary fiction

---

**Generated Response**

B: It's Literary fiction, and was released in 2008.

---

**Dialogue Context**

A: Who wrote Of Mice and Men?
B: It was written by John Steinbeck
A: Didn't he also write The Red Pony?
B: Yes He also wrote The Red Pony
A: Can you suggest other titles by him to read?

---

**Retrieved Knowledge**

Cannery Row, written_by, John Steinbeck

The Grapes of Wrath, written_by, John Steinbeck

Tortilla Flat, written_by, John Steinbeck

---

**Generated Response**

B: Yes he also wrote Tortilla Flat

---

Figure 13: Examples of the dialogue history with its corresponding retrieved knowledge and generated response from our SURGE framework. The fact is represented as the format of (head, relation, tail), where ~symbol in the front of relation (i.e., ~relation) in the retrieved knowledge denotes the inverse relation. In this example, we only provide the correct cases of both retrieval and generation.

---

**Dialogue Context**

A: Could you recommend some movies by director Simon Wood?

---

**Retrieved Knowledge**

The One That Got Away, written_by, Simon Wood

Simon Wood, ~written_by, The One That Got Away

Author, ~is-a, Simon Wood

---

**Generated Response**

B: Simon Wood directed The One That Got Away. Have you seen that?

---

---

**Dialogue Context**

A: I like David McCullough. Could you recommend any books of him?

B: Sure. He wrote The Greater Journey: Americans In Paris. Also, he wrote some documentary and Indie films.

A: Thank you for the information. When was The Greater Journey: Americans In Paris released?

---

**Retrieved Knowledge**

The Greater Journey: Americans In Paris, release_year, 2011

2011, ~release_year, The Greater Journey: Americans In Paris

Paris, written_by, Edward Rutherfurd

---

**Generated Response**

B: It was released in 2011. It's a great book.

---

---

**Dialogue Context**

A: I like Tony Tanner, what do you think?

---

**Retrieved Knowledge**

Sense and Sensibility, written_by, Tony Tanner

Tony Tanner, ~written_by, Mansfield Park

Mansfield Park, written_by, Tony Tanner

---

**Generated Response**

B: I think he is a great actor. He starred in Sense and Sensibility and Mansfield Park.

---

Figure 14: Examples of the dialogue history with its corresponding retrieved knowledge and generated response from our SURGE framework. The fact is represented as the format of (head, relation, tail), where ~symbol in the front of relation (i.e., ~relation) in the retrieved knowledge denotes the inverse relation. In this example, we only provide the correct cases of both retrieval and generation.

---

**Dialogue Context**

A: Do you know what books Jennifer L. Armentrout has written?

B: She wrote Wicked, White House Kiss and Wait For You. Are you interested in her other books as well?

A: No thanks. What genre is Wicked?

B: Wicked is a Play that won the Pulitzer Prize for Drama. Would you like more information about it?

A: Sure, sounds interesting

---

**Retrieved Knowledge**

Wicked, release_year, 2008

2008, ~release_year, Wicked

Wicked, release_year, 2014

---

**Generated Response**

B: It is a play written by Jennifer L. Armentrout and released in 2008.

---

---

**Dialogue Context**

A: I like the movie Armageddon.

B: Did you know Armageddon is written by Tony Gilroy?

A: Oh, didn't he also write The Bourne Legacy?

---

**Retrieved Knowledge**

Michael Clayton, starred_actors, Tony Gilroy

Michael Clayton, written_by, Tony Gilroy

Proof of Life, written_by, Tony Gilroy

---

**Generated Response**

B: Yes, he did. He also wrote Michael Clayton.

---

---

**Dialogue Context**

A: I like Deadpool or any movies like Deadpool.

---

**Retrieved Knowledge**

Deadpool, starred_actors, Ryan Reynolds

Deadpool, starred_actors, Stefan Kapicic

Deadpool, starred_actors, Ed Skrein

---

**Generated Response**

B: Deadpool is a great movie. Stefan Kapicic starred in it. Stefan Kapicic also starred in The Last Man and The Last Man.

---

Figure 15: Examples of the dialogue history with its corresponding retrieved knowledge and generated response from our SURGE framework. The fact is represented as the format of (head, relation, tail), where ∼symbol in the front of relation (i.e., ∼relation) in the retrieved knowledge denotes the inverse relation. In this example, we only provide the failure cases due to the problem on data (first row), retrieval (second row), and generation (third row).

