# OpenReview forum: "Knowledge-Consistent Dialogue Generation with Language Models and Knowledge Graphs"
_ICLR.cc/2023/Conference — Submitted to ICLR 2023_

### Official Review · Reviewer_6h9N · 2022-10-25

**Confidence:** 4
**Correctness:** 3
**Technical Novelty And Significance:** 2
**Empirical Novelty And Significance:** 3
**Recommendation:** 6

**Clarity, Quality, Novelty And Reproducibility:**

-

**Strength And Weaknesses:**

-

**Summary Of The Paper:**

This paper proposes subgraph retrieval-augmented generation (SURGE), a framework for knowledge-grounded dialogue. It can be divided into 3 major components:
(i) a subgraph-retriever, which retrieves only  context-relevant triples from a KG,
(ii) a permutation and inversion invariant graph encoding scheme, and
(iii) a contrastive learning objective to enforce consistency between the retrieved triples and generated text

The authors also propose a new metric called knowledge-verified question-answering (KQA) to evaluate the generated responses. They evaluate the method on OpendialKG and KOMODIS. Both see fair improvements over the baselines on KQA and minimal improvements on lexical overlap metrics like bleu and rouge.

The paper is well-written and understandable. However,
(i) some space could be saved in section 3.4 and used for ablations and analysis.
(ii) I would have liked to see more information about sensitivity analysis on graph encodings and a discussion on why the authors think retrieving proper subgraphs is easier/more important than say, teaching the decoder to look for/ignore relevant/irrelevant knowledge conditioned on the history. Some toy experiments would have cemented the premise of the paper.
(iii) contrastive training does not give any significant benefit. Why is that? One would intuitively think otherwise. Some analysis there would be appreciated.

There are minor writing errors. E.g.: in section 5.2, the last line should be SURGE (contrastive)

**Summary Of The Review:**

-

---

> ### Author Response · Authors · 2022-11-15
> **Response to Reviewer 6h9N**
>
> We sincerely thank you for your constructive and helpful comments. We initially address all your concerns below:
>
> ---
>
> > **[Q1]** Use more space on ablation studies and analyses.
>
> Thank you for your suggestion. We have tried our best to include diverse ablation studies and analyses in the main paper while making the paper as compact as possible, however, due to page limits, we have unfortunately moved some less important analyses in the Appendix. If one additional page is allowed after the acceptance, we will move more analyses from the Appendix to the main paper. Below, we summarize ablation studies and analyses we have conducted.
>
> ### Analyses
> * Table 3: Efficacy of contrastive learning for knowledge-consistent response generation
> * Figure 4, 8 (right): Performances of various knowledge retrieval schemes
> * Table 5: Human evaluation results
> * Figure 6: Embedding space visualization of the contrastive learning
> * Table 7: Experimental results with another pre-trained language model, namely BART-base
> * Figure 8 (left): Sensitive analyses on the number of facts for retrieval
> * Table 9 (right): Sensitive analyses on GNN variants in our graph encoding
> * Figure 5, 11, 12, 13, 14: Examples of dialogue generation and subgraph retrieval, including failure cases for future study
>
> ### Ablation studies
> * Table 1: Ablation studies on retrieval variants & contrastive learning
> * Table 4: Ablation studies on graph encoding schemes
>
> ---
>
> > **[Q2]** I would like to see sensitivity analyses on graph encodings.
>
> For sensitive analyses on graph encodings, please refer to **Table 4 and the “Sensitive Analysis on Graph Encoding” paragraph in Section 5.4**, where we use different variants for graph encoding functions that we described in Section 3.4. In addition to this, we also vary GNNs for graph encoding, and show results in Table 9 of the Appendix.
>
> ---
>
> > **[Q3]** I would like to see a discussion on why retrieving proper subgraphs is more important, than training the decoder to look for the relevant knowledge.
>
> Thank you for pointing out the important point, and we already discussed such the point in our experiments, especially in comparisons to recent baselines EARL [1] and Space Efficient Encoding [2].
>
> Note that EARL [1] trains the RNN decoder to look for the relevant knowledge conditioned on the dialogue history, and Space Efficient Encoding [2] compresses the k-hop subgraph knowledge in the input of PLMs to generate the response conditioned on the compressed knowledge and the dialogue history. First of all, Space Efficient Encoding [2] outperforms EARL [1], which suggests that encoding the subgraph information in the input of PLMs might be more beneficial than teaching the decoder to look for the relevant knowledge. This might be because, we can contextualize the knowledge in the input more, compared to using the knowledge selection in the decoder part.
>
> Moreover, our SURGE outperforms the Space Efficient Encoding [2], which further suggests that additionally conducting subgraph-relevant context retrieval is more beneficial than directly compressing all k-hop subgraphs into the compact form, for knowledge encoding. This might be because, as shown in Figure 1, k-hop subgraphs might contain irrelevant knowledge. Therefore, we conclude that retrieving proper subgraphs is more effective than existing schemes in [1, 2].
>
> ---
>
> > **[Q4]** Contrastive learning does not provide significant benefits.
>
> We first would like to clarify the goal of contrastive learning, that is **we aim to ensure the model to reflect the retrieved subgraphs in the generated responses**, by regularizing the latent space of graphs and texts formalized in Equation 8. And, the results in Table 1 might not be optimal to measure the benefit of contrastive learning, since it does not measure how faithful the generated response is by the retrieved knowledge. Also, since the generation performance is affected by the retrieval performance, due to similar retrieval results across our SURGEs, the performances of our variants might be similar.
>
> Therefore, to exactly measure the effectiveness of contrastive learning, we have to use the same retrieval results across different SURGEs, and measure how faithful the generated response is by the augmented knowledge. As shown in Table 3, under such the controlled experimental setting with the KF1 metric, we demonstrated that SURGE (contrastive) faithfully reflects the augmented facts in the responses (See the Knowledge-Consistent Generation paragraph of Section 5.4 for details).
>
> ---
>
> > **[Q5]** Minor writing errors.
>
> Thank you for pointing it out, and we have fixed it in the revision.
>
> ---
>
> ### References
>
> [1] EARL: Informative Knowledge-Grounded Conversation Generation with Entity-Agnostic Representation Learning, EMNLP 2021.
>
> [2] Space Efficient Context Encoding for Non-Task-Oriented Dialogue Generation with Graph Attention Transformer, ACL 2021.

---

> ### Author Response · Authors · 2022-12-03
> **The end of the discussion phase is approaching**
>
> Dear Reviewer 6h9N,
>
> We appreciate your positive comments that our work is well-written and understandable. We have done our best to respond to your comments in the discussion phase, which we believe have made our work more solid. Please let us know if you have any further questions.
>
> Best regards, Authors

---

### Official Review · Reviewer_YCpx · 2022-10-26

**Confidence:** 4
**Correctness:** 3
**Technical Novelty And Significance:** 3
**Empirical Novelty And Significance:** 2
**Recommendation:** 6

**Clarity, Quality, Novelty And Reproducibility:**

The description is mostly clear and easy to follow.
The experiments are done correctly. The use of knowledge verifying measure is very interesting.
The implementation details are described in detail in annex. The source code is provided. The method can be reasonably reproduced.

**Strength And Weaknesses:**

The idea of selecting relevant subgraphs from a knowledge graph is well motivated. This is a known problem. There are approaches proposed to do it. This paper also relies on an existing tool to identify the entities in the knowledge graph and to select subgraphs.
The two conditions for subgraph encoding are important in some cases. This paper proposes interesting ideas to satisfy them.
However, the tricks used to satisfy these conditions may raise some questions:
1. The permutation invariance condition is satisfied by sorting the triples. This is a kind of normalization, that transforms a subgraph to a normalized form. Even though the output will be permutation invariant, it does not require that the encoding process after the transformation be permutation invariant. The latter is what one intend to obtain by imposing the condition. So, it is not sure that making this transformation would make the encoding better.
2. The encoding process then considers a set of triples as a sequence of tokens. that is submitted to a PLM for encoding. In this step, one may lose the specific triple structure for relations. Interpreting a set of triples as a sentence makes the encoding easier, but may alter the meaning.
3. In the approach to increase efficiency, it seems that the entities from different triples are mixed up, then sorted. The purpose is to encode an entity only once. However, this process would alter the meaning of the initial subgraph. For example, the initial set of triples [(a,r1,b), (c,r2,d)] would become the same as [(a,r2,c),(d,r1,b)] after this operation. I understand that this operation would transform both into a set of entities like [a,b,c,d,r1,r2] (this part is not very clear). However, the two different sets of triples may mean very different things and we may not want to unify them.
4. The relation inversion invariance is imposed to all the repations. If the relation is kept the same after inversion, this does not make sense. It seems that the inversed relation is different:  ~relation. In this case, I do not see what new information this inversion could add. For example, saying that (a, born-in, b) would be the same as saying (b,  ~born-in, a). What's the benefit of adding the latter?

In the experiments, it seems that the way subgraphs are retrieved is quite simple. There could be more sophisticated methods in the literature. For example [1] proposes a knowledge selection network to make the selection. Given this fact, it is unsure that the method is compared to the state-of-the-art knowledge selection approaches.

[1] Yutao Zhu, Jian-Yun Nie, Kun Zhou, Pan Du, Hao Jiang, and Zhicheng Dou, Proactive Retrieval-based Chatbots based on Relevant Knowledge and Goals, SIGIR 2021.

**Summary Of The Paper:**

This paper proposes an approach to knowledge-grounded dialogue. The key problems the paper intends to tackle is the selection and encoding of relevant knowledge according to the dialogue context. To this end, subgraphs are selected using the context. Subgraph encoding is asked to meet some conditions: permutation invariance and relation inversion invariance. Two tricks are used to satisfy these conditions: by sorting the triples, and by adding the inverse relation.
The proposed method is compared to a set of baselines, in which either random knowledge selection, or a selection using T5 is used. The experiments show that the proposed method outperforms the baselines.
In addition, a knowledge verifying metric - KF1 - is used to measure if the generated response is consistent with the known knowledge.
The overall ideas of locating relevant subgraphs and doing subgraph encoding are interesting.

**Summary Of The Review:**

The idea is very well motivated. The paper proposes some way to encode the retrieved subgraphs so that the pieces of knowledge can be better encoded.
However, there are questions about the way the subgraphs are encoded. It may lead to undesirable cases.

---

> ### Author Response · Authors · 2022-11-11
> **Response to Reviewer YCpx (2/2)**
>
> > **[Q5]** The proposed subgraph retrieval is quite simple, and there could be more sophisticated methods, for example [1].
>
> We respectfully agree with your comment that, the problem of subgraph retrieval is not entirely novel and has been actively tackled in previous works on different domains, such as question answering [2] and dialogue generation [3].
>
> However, compared to [1] which requires explicit supervised signal for knowledge selection, **our retrieval method is jointly trainable with the dialogue generation objective without any supervision for the retrieval**. Also, compared to [1] that performs classification to select the knowledge and the response, our framework tackles the challenging dialogue generation problem with graph neural networks for subgraph retrieval, and with pre-trained language models where retrieved triplets are directly augmented as the input of PLMs.
>
> Note that dialogue generation with PLMs are very recently studied [4, 5], and we believe that our paper handles a very-hot and important topic: knowledge-consistent dialogue generation with language models over the subgraph-retrieval-augmented framework, compared against previous dialogue literature with RNNs and classification settings for subgraph retrieval.
>
> ---
>
> ### References
>
> [1] Proactive Retrieval-based Chatbots based on Relevant Knowledge and Goals, SIGIR 2021.
>
> [2] Generating Natural Answers by Incorporating Copying and Retrieving Mechanisms in Sequence-to-Sequence Learning, ACL 2017.
>
> [3] A Copy-Augmented Sequence-to-Sequence Architecture Gives Good Performance on Task-Oriented Dialogue, EACL 2017.
>
> [4] BlenderBot 3: a deployed conversational agent that continually learns to responsibly engage, preprint 2022.
>
> [5] LaMDA: Language Models for Dialog Applications, preprint 2022.

---

> ### Author Response · Authors · 2022-11-15
> **Response to Reviewer YCpx (1/2)**
>
> We sincerely thank you for your constructive and helpful comments. We initially address all your concerns on Section 3.4 (Invariant graph encoding), and on related work below. Please note that we also have reflected your comments on Section 3.4 in our updated paper.
>
> ---
>
> > **[Q1-1]** Permutation invariance is satisfied by sorting the triples; however, it may not be preserved after the encoding process.
>
> Note that, as long as the input of pre-trained language models (PLMs) is the same across different triplet permutations, the output of the PLMs is the same. Therefore, since we allow the PLMs to receive **the same input for different orders of triplets along with sorting operation** in Equation 5, the permutation invariance property holds after the encoding process. Please see Appendix C for proofs.
>
> ---
>
> > **[Q1-2]** Explain why this graph transformation would make the encoding better.
>
> This is because, for two semantically same graphs in terms of permutation- and relation-invariance properties, **our graph encoding function yields the same representation**, which is not possible without our graph encoding. See Table 4 for experimental validation, which shows that our graph encoding function brings performance improvements.
>
> ---
>
> > **[Q2]** The proposed encoding process treats a set of triplets as a sequence of tokens, which may lose the structural information of triplets and alter the intended meaning.
>
> This may be the misunderstanding, since **we use the graph-structural information in Equation 7 during the encoding process**. Specifically, we perturb each token embedding of triplets with respect to its embedding from relation-aware GNNs, which allows the model to capture graph-structural information. We have further emphasized such point in Section 3.4, during the revision.
>
> ---
>
> > **[Q3]** In Graph Encoding of Section 3.4, when merging multiple retrieved triplets into one set, the merged set might have different meanings from original triplets, e.g., two triplets {(a,r1,b), (c,r2,d)} into [a,b,c,d,r1,r2].
>
> Thank you for your comment. However, we are afraid that you might misunderstand the important idea of our graph encoding method.
>
> First of all, in the case of naive encoding, your understanding is right. Given a set of triplets {(a,r1,b), (c,r2,d)}, the naive graph encoding function transforms it into a sequence of tokens: [a, r1, b, c, r2, d], and this might yield the undesired information in a graph-structural point of views.
>
> To remedy this issue, we propose the invariant and efficient graph encoding function that can reflect the graph structure into PLMs. Specifically, we first transform the retrieved triplets into a sequence of entity tokens [a, b, c, d], after sorting entities. Then, as described in the response of **[Q2]**, we perturb entity tokens with respect to their graph embeddings from relation-aware GNNs formalized in Equation 7.
>
> ---
>
> > **[Q4]** The inversion invariance seems to add the same information twice, since (a, born-in, b) might be the same as (b, ~born-in, a). What is the benefit of adding inversions?
>
> As described in the response of **[Q1-1]**, PLMs embed (a, born-in, b) and (b, ~born-in, a) differently, since PLMs are permutation sensitive. Therefore, to make sure that the representation of the meaning (a, born-in, b) is the same as the representation of (b, ~born-in, a), we provide the same input sequence [a, born-in, b, b, ~born-in, a] to the PLMs for the input (a, born-in, b) and (b, ~born-in, a), with the SORT and INV operations.

---

> > ### Comment · Reviewer_YCpx · 2022-12-02
> > **reactions to your answers**
> >
> > I'm not convinced by the answers. Rather, I have more doubt now.
> > Q1-2: I agree that your sorting will make the representation permutation-invariant. The question is about whether such a representation is better (reasonable), and is what we want.
> > Let me show an example: suppose you have a set of neighbors to your house. You sort them according to their names, and use that sequence to create a representation about your neighbors. This representation is permutation-invariant: the same set of names is transformed to the same sequence.The question is: Is it natural to assume and impose an order among your neighbors based on their names? Would such a permutation-invariant representation a good one, i.e. capturing what we want to encode? The order information IS used by PLM later, but you do not have the control on how this information is used. It might be possible (and very likely due to the position encoding in PLM) that the two persons listed side-by-side have more influence mutually. In reality, they may be neighbors that do not have any relationship and do not know each other. Imposing an order among the entities to create a permutation-invariant representation is artificial.
> >
> > Q2-3: You argue that the graph structure is considered by Equation (7), that is by using the existing approach R-GNN, which tries to create node embeddings using the neighbors and the relations. This can indeed incorporate information using the triples.
> > My initial question is about the loss of information when the entities connected by different relations are simply merged, and this for different entities. Taking the example again: given two triples (a,r1,b), (c,r2,d), if we mix up the entities into (a,b,c,d), how can one differentiate it from the triples (c,r1,b), (a,r2,d) ? Assuming that you have two mariage triples. Once the 4 people are mixed up together, you will be unable to figure out who form couples.
> > You also argue that this is the naive approach that you do not use, and you add sort and inverse relations in your approach. However, these operations you added do not solve the above problem.

---

> > > ### Author Response · Authors · 2022-12-02
> > > **Follow-up Response to Reviewer YCpx**
> > >
> > > We sincerely appreciate your response and the chance to address your remaining concerns. In this response, we address your remaining concerns/comments in your response below:
> > >
> > > ---
> > >
> > > > **[Q1-2]** Is it natural to assume and impose an order of neighboring nodes based on their names?
> > >
> > > First of all, we want to recapitulate why the “permutation invariance” is important in encoding multi-relational graphs into the text sequence. As you mentioned, PLM is “permutation sensitive” since the meaning of the sentence can vary when we change the order of words (e.g., “A born in C” != “C born in A”).
> > >
> > > However, the multi-relational graphs are “permutation invariant” since they are represented as a set of triplets. For instance, given the multi-relational graphs with two triplets, {(a, born-in, b), (c, born-in, d)}, **the order of elements (triplets) does not affect the entire representation of the graph.**
> > > (e.g., {(A, born-in, C), (B, born-in, D)} == {(B, born-in, D), (A, born-in, C)})
> > >
> > > We want to emphasize that **elements** in Definition 3.1. indicate **triplets** in a subgraph, not the entities or relations in each triplet.
> > > Therefore, the SORT operation (Equation 5) **sorts the triplets in the subgraph, neither the entities nor the relations**. For instance, the desired output of the sort operation is as follows: SORT({(B, born-in, D), (A, born-in, C)}) = {(A, born-in, C), (B, born-in, D)}.
> > >
> > > **NOT THIS**: SORT({(B, born-in, D), (A, born-in, C)}) = [A, B, C, D, born-in, born-in]
> > >
> > > With a naive encoding (simple concatenation of every triplet in a graph), **PLM yields different representations for different orders of triplets in the subgraph**. Therefore, if the PLM is only fine-tuned with the input [A, born-in, C, B, born-in, D, where was A born?], there is no guarantee that the PLM will output the exact same response to the trained one given the input with a permuted graph [B, born-in, D, A, born-in, C, where was A born?] in the inference stage, since the PLM is order-sensitive due to its positional encoding. In order to prevent the aforementioned scenarios, we decide to design the permutation-invariant graph encoding which yields stable results regardless of the order of triplets in the graph.
> > >
> > > ---
> > >
> > > > **[Q2-3]** Eq.(7) can indeed incorporate information using the triples. My initial question is about the loss of information when the entities connected by different relations are simply merged, and this for different entities.
> > >
> > > There is no loss of information; as you mentioned, Equation 7 can indeed incorporate information using the triplets, which also include relational information.
> > > In particular, for efficiency, we only use the set of unique entities in the graph encoding. In this case, the SORT operation becomes problematic **if we do not consider the graph structure**. From the example of [Q1-2], after the graph encoding $\tilde{\psi}$ in the last equation on page 5, the input sequence becomes [A, B, C, D, where was A born?]. As you mentioned, PLM cannot figure out where A was born since we lost the information about the relations, only for this naive case.
> > >
> > > However, we clarify our scheme again that, we add **perturbation of the token embeddings for each entity in PLMs with respect to their graph representation in $\mathcal{Z}$** ($\beta$ function in equation 7) to solve the given problem, along the SORT and INV operation. In conclusion, our proposed graph encoding operation is $\psi^*$ **in the last equation of Section 3.4**. The graph encoding using only entities is **NOT** the method we proposed, rather, it is one of the baselines we used to validate the significance of our approach.
> > >
> > > In other words, we can preserve the graph structure in the token embedding space, rather than the input sequence. For example, given the above example [A, B, C, D, where was A born?], the token embedding function maps A to $f$(A). In a simple form, the token embedding of A becomes $\beta$($f$(A), $\mathcal{Z}$) $= f$(A) + $h$(A, born-in, C) after the perturbation function $\beta$, where $h$ is the conceptual function of the relation-aware message passing. Therefore, with our method, A can still retain the relational information that A was born in C even after the SORT operation in the text sequence.
> > >
> > > We also empirically show that the graph-conditioned perturbation can provide the relational information into the sequence only with entities. In Table 4, we clearly show that the entity-only graph encoding (third row) leads to a significant performance drop in the knowledge-consistent response generation. On the other hand, our method (last row) that utilizes the graph-conditioned perturbation (Invariant & Efficient) shows a decent performance even with a shorter sequence length.
> > >
> > > ---
> > >
> > > We hope that our answer clarifies your concerns about permutation invariance and graph encoding. We will add more details with illustrations in the future revision. Let us know if you have any further questions or anything else that we should address.

---

> ### Author Response · Authors · 2022-12-03
> **The end of the discussion phase is approaching**
>
> Dear Reviewer YCpx,
>
> We appreciate your positive comments that our idea is very well-motivated and interesting. Moreover, We sincerely appreciate the opportunity to address your additional concerns and your follow-up comments. We have made every effort to faithfully address all your comments in the responses. Please let us know if you still have remaining concerns we should address.
>
> We thank you again for your time and efforts in reviewing our paper, and thank you for engaging with us.
>
> Best regards, Authors

---

### Official Review · Reviewer_CSao · 2022-10-27

**Confidence:** 4
**Clarity, Quality, Novelty And Reproducibility:** see above.
**Correctness:** 3
**Technical Novelty And Significance:** 2
**Empirical Novelty And Significance:** 2
**Recommendation:** 3

**Strength And Weaknesses:**

**Strengths**

The paper is generally well-written.

The study of generating context-relevant and knowledge-consistent dialogues with a KG is important and interesting. I appreciate the authors' investigation of this important problem.

The three components (the GNN-based context-relevant subgraph retrieval method, the graph encoder, and a graph-text contrastive learning objective) seem to be new. Experimental results show the empirical advantage of the new methods.

**Weaknesses**

The overall framework is not new, very similar to (Lewis et al., 2020b). In fact, the authors should make this point more explicit.

Although the three components are somewhat new, neither is of high originality, and neither is throughly investigated (see more comments below). This paper seems to belong to a kind of assembly innovation: putting several pieces together, while each piece brings a small adjust. I suggest the authors to focus on major novelty and fully validate the novelty.

More ablation studies are needed to show some significant benefit of each component, by fixing other components while modifying one component. Now the significance of each component is not clear. It is not good to simply say that "We empirically verify that the use of GNN as the triplet embedding yields the better retrieval performance compared to previous PLM-based methods."

In Table 1 and Table 3, the advantages of SURGE (contrastive) are not significant, compared with SURGE (unsupervised) and SURGE (semi-supervised), especially considering SURGE (contrastive) also consists of semi-supervised learning.

In Table 1, SURGE (unsupervised) outperforms the other two SURGE performance on BLEU, ROUGE and Unigram, is there any explanation for that?

Using only unique entities does not guarantee permutation invariant.

Some problems with writing:

Section 3.4 (INVARIANT GRAPH ENCODING) is hard to be understood, even after reading the Appendix.

The two examples in Figure 1 illustrate the two problems - irrelevant knowledge and inconsistent generation. Somewhat confusing. "Irrelevant facts" and "factually wrong" actually may express almost the same meanings.

maaping -> mapping

duplicate definition of SURGE (semi-supervised) on the bottom of page 7

Some places should be re-written to be clearer in both grammar and meaning:

we utilize the Graph Neural Networks (GNNs) in the triplet embedding function
-> we utilize the Graph Neural Networks (GNNs) for the triplet embedding function ?

The 2nd paragraph, Section 3.1: Any entity and relation are mapped to a sequence of l tokens. Is the length l fixed for different entities and relations?

--- **To reply to Follow-up Response from the authors (Provided here, since the comment button is disabled 2022/12/12)** ---

Using $\eta$ alone should be an important baseline to be compared.

If you think that Eq.(7) is the first in developing GNN-based entity embedding in the literature (not necessarily for dialog generation), then you can clearly say so in the paper. Suppose this is yes, then the contribution should be stated around this, rather than putting more emphasis on "Invariant and Efficient Graph Encoding". To my understanding, "Invariant and Efficient" are actually down to only encoding the unique entities, which essentially is entity embedding. Clarity issues remain.

I can see the ablation result in Table 3. My concern is that the overall performance of SURGE (contrastive) is limited. In real applications, the model needs to work with retrieved knowledge.

**Summary Of The Paper:**

Existing knowledge-grounded dialogue generation models with KG suffer from irrelevant knowledge and inconsistent generation. To attack the above two problems, this paper proposes a GNN-based context-relevant model to retrieve relevant subgraphs from KG, an invariant yet efficient graph encoder, and a graph-text contrastive learning objective which overcome the exposure bias in teacher forcing to guarantee knowledge-consistent generation. Extensive experiments have been conducted on OpendialKG and KOMODIS and the results show the superiority of SURGE over baselines.

**Summary Of The Review:**

see above.

---

> ### Author Response · Authors · 2022-11-11
> **Response to Reviewer CSao (3/3)**
>
> > **[Q7]** In Figure 1, two problems: “irrlevant facts” and “inconsistent generation”, are confusing, since they may be the same problem.
>
> Figure 1 illustrates two different problems, and **they are not the same problem**, as we clarify below.
>
> First of all, the first problem (i.e., irrelevant facts) in Figure 1 indicates that the extracted subgraphs from the dialogue history contain **irrelevant facts, not helpful to generate the correct response**. For example, as described in the second paragraph of Section 1, one fact, (Jane Austen, place_of_birth, Steventon), is not relevant to generate the response about recommending books written by Jane Austen. Also, we additionally note that, we tackle this problem with context-relevant subgraph retrieval, which aims to retrieve and augment only the related knowledge.
>
> Meanwhile, the second problem (i.e., inconsistent generation) indicates that the **generated response contains factually wrong knowledge**, even though we retrieve only the relevant facts. For instance, as shown in the right example (i.e., Issue 2. within the blue box) of Figure 1, the generated response (i.e., “The Secret Life of a Man Who Was Born”) is not consistent with the retrieved knowledge, which is also factually wrong. To tackle this problem, we propose graph encoding and contrastive learning to effectively reflect the knowledge from graphs.
>
> Therefore, both “irrelevant facts” and “inconsistent generation” are two individual problems.
>
> ---
>
> > **[Q8]** Is the token length $l$ in Section 3.1 fixed for different entities and relations?
>
> We apologize for the confusion, and the token length $l$ can differ for different entities and relations (i.e., $l$ is decided by the number of tokens for entities and relations after tokenizing them), which we have revised in the revision in Section 3.1.
>
> ---
>
> > **[Q9]** Other minor typos and grammar corrections.
>
> Thank you for pointing them out, and we have revised them accordingly.
>
> ---
>
> ### References
>
> [1] Retrieval-Augmented Generation for Knowledge-Intensive NLP Tasks, NeurIPS 2020.

---

> ### Author Response · Authors · 2022-11-11
> **Response to Reviewer CSao (2/3)**
>
> > **[Q5]** Invariant Graph Encoding described in Section 3.4 is hard to understand, even in the Appendix.
>
> We apologize for the confusion, and thank you for pointing it out. **We have included an illustration of our graph encoding mechanism in Appendix (Figure 8)**, for additionally helping the understanding of it. Please let us know if there remains anything else in Section 3.4 that we can further improve.
>
> ---
>
> > **[Q6-1]** More ablation studies are required, since the significance of each component is unclear.
>
> It seems that you have missed the experimental results from the ablation studies. To deeply examine each component of our framework, **we have exhaustively and comprehensively conducted various ablation experiments in Tables 1, 3, and 4** of the main paper, and also additionally in **Figure 8 and Table 9 of Appendix**. We summarize them as follows:
>
> * In Table 1, we vary the training objectives of our proposed SURGE framework, while using the same graph encoding method (i.e., Invariant & Efficient encoding), to see the effect of each training objective. As shown in Table 1, we observe that all of our variants significantly outperform baselines, and also we observe that the models trained with semi-supervised objective (semi-supervised and contrastive) bring more performance gains in KQA metrics. These results confirm that the explicit signal for knowledge retrieval contributes to the factual response generation with augmented knowledge.
>
> * Note that, the performances of our SURGEs in Table 1 are affected by the retrieval results. Therefore, it is difficult to analyze the direct impact of graph-text contrastive learning from the results in Table 1. To solely evaluate the effect of contrastive learning, in Table 3, we report the performance of our SURGEs with the same retrieval results by using the same gold knowledge (See the Knowledge-Consistent Generation paragraph of Section 5.4 for details). As shown in Table 3, we observe that our graph-text contrastive learning scheme improves the KF1 score, demonstrating that it improves the knowledge-consistent generation performance.
>
> * Lastly, in Table 4, we conduct the ablation study on the different graph encoding methods, such as Naive, Invariant, Efficient, and Invariant & Efficient, discussed in Section 3.4, where we train them with the semi-supervised learning objective. As shown in Table 4, we observe that the model with invariant & effective graph encoding shows the best performance.
>
> * We have more ablation studies and sensitive analyses in Appendix. In particular, in Figure 8, we vary the number of augmented facts, and also more exhaustively compare the retrieval performances of different models. Also, in Table 9, we perform a sensitive analysis by varying the graph neural networks used in our proposed framework.
>
> Based on all experimental results that we have reported, we believe that **all architectural design choices that we propose are verified with experimental results**, i.e., each proposed component is necessary for our target problem: knowledge-consistent dialogue generation.
>
> ---
>
> > **[Q6-2]** It is not good to simply say that "We empirically verify that the use of GNN as the triplet embedding yields the better retrieval performance compared to previous PLM-based methods".
>
> We apologize for the confusion; however, as shown in Table 1 and Figure 4, we already **experimentally verified that the use of GNNs is better than PLM-based methods for triplet retrieval**. In particular, our SURGE with contrastive learning outperforms other retrieval variants (sparse and dense retrievers) on both generation and retrieval performances. To further clarify, we have included the experimental references for our claim, during the revision.

---

> ### Author Response · Authors · 2022-11-15
> **Response to Reviewer CSao (1/3)**
>
> We sincerely thank you for your constructive and helpful comments. We initially address all your concerns below:
>
> ---
>
> > **[Q1]** The overall framework is not new, similar to the existing RAG [1].
>
> This is a critical misunderstanding of our work's contribution. We first would like to clarify that **document retrieval tackled in RAG [1] is a completely different problem from our subgraph retrieval**. We already acknowledged in Section 3.6 that, to make our subgraph retrieval framework trainable from the dialogue generation loss, we factorize and sample the latent variable, as formalized in Equation 10; however, this is only the similarity of our framework to RAG [1], and the novelty of our work is not solely limited to the retrieval-augmented generation.
>
> Besides the above one similar point, our work tackles the completely different challenges arising from **knowledge-consistent dialogue generation with knowledge graphs**. In particular, we aim to retrieve the knowledge over the graph-structured data, and we aim to consistently generate the textual response from the retrieved subgraphs; And, to this end, we propose the novel **GNN-based subgraph retrieval**, and also propose novel **invariant graph encoding** and **graph-text contrastive learning**, which are not considered in the existing RAG [1], and also not similar to the RAG [1].
>
> ---
>
> > **[Q2]** The proposed three components are somewhat new, yet they are not of high originality; this paper has the assembly innovation.
>
> As you acknowledged, our knowledge-consistent dialogue generation framework itself is novel. However, we respectfully disagree that the proposed components lack high originality, and confidently believe that **each individual component is also novel**, that is, to the best of our knowledge, not explored in the existing knowledge-consistent dialogue generation literature. We recapitulate the originality of three proposed components, as follows:
> * **(GNN-based subgraph retrieval for KG-augmented dialogue generation)** We propose to retrieve the facts in the knowledge graph and then augment the retrieved ones directly in the language model input, where we use GNNs for effective and efficient retrieval of graph-structured data with leveraging both node and edge representations, which is so far not explored in the existing work.
> * **(Permutation- and inversion-invariant yet efficient graph encoding)** We propose to effectively encode graph-structural information into language models by considering permutation invariance and relation-inversion invariance properties of graph-structured data, which are novel schemes for augmenting knowledge into LMs.
> * **(Graph-text contrastive learning for knowledge-consistent generation)** To ensure that the augmented knowledge from the retrieved subgraphs is faithfully reflected in the generated text response, we propose graph-text contrastive learning, which is, to the best of our knowledge, not explored for knowledge-consistent dialogue generation.
>
> ---
>
> > **[Q3-1]** Also, SURGE (unsupervised) outperforms the other two SURGE (semi-supervised; contrastive) on BLEU, ROUGE, and Unigram.
>
> As discussed in Section 4, the **metrics (e.g., BLEU, ROUGE, and Unigram) that use lexical overlaps of words are largely suboptimal** for generation tasks, since, the labeled gold response covers only one case, while there can be many different answer responses (i.e., the generated answer containing the correct knowledge but not labeled in the answer is evaluated as incorrect). Therefore, the differences among our variants might be small in such suboptimal metrics, and sometimes the inferior model (unsupervised) can outperform others (semi-supervised; contrastive). On the other hand, the proposed Knowledge-verifying QA (KQA) can capture multiple possible answers validated with knowledge graphs, therefore, we argue that **we should evaluate models with KQA metrics**, where two SURGEs (semi-supervised; contrastive) are superior than the SURGE (unsupervised).
>
> > **[Q3-2]** The advantages of SURGE (contrastive) are not significant, compared with SURGE (unsupervised; semi-supervised).
>
> Similar to the above issues, the common metrics might not be optimal to validate whether the generated sentence contains the retrieved knowledge, which is the aim of contrastive learning. However, as shown in **KF1 score of Table 3**, SURGE (contrastive) outperforms other variants with large margins, therefore, we believe that contrastive scheme is sufficiently helpful to generate the knowledge-consistent response.
>
>
> ---
>
> > **[Q4]**  Using only unique entities does not guarantee permutation invariant
>
> We appreciate your comment but we are afraid that you have missed some important points.
>
> We do not claim that using only unique entities does guarantee permutation invariant. In Section 3.4, **we use ‘SORT’ operation (Equation 5) to satisfy the permutation invariance property**, and it is theoretically justified in Proposition C.2. of Appendix C.

---

> > ### Comment · Reviewer_CSao · 2022-12-09
> > **About SURGE (contrastive) performance**
> >
> > > However, as shown in **KF1 score of Table 3**, SURGE (contrastive) outperforms other variants with large margins, therefore, we believe that contrastive scheme is sufficiently helpful to generate the knowledge-consistent response.
> >
> > However, by looking F1 (not clear it is KQA F1 or Unigram F1?) in Table 3, SURGE (contrastive) is inferior. Also  by looking the overall performance as reported in Table 1, SURGE (contrastive) is inferior in KQA EM and very close in KQA F1, as compared to SURGE (semi-supervised).

---

> ### Author Response · Authors · 2022-11-29
> **A Gentle Reminder**
>
> Dear Reviewer CSao,
>
> We sincerely appreciate your time and effort in reviewing our paper. We believe that your comments will further strengthen our paper. During the discussion period, we have made every effort to faithfully address all your comments in the responses and revision. Therefore, we kindly request you read our responses and reconsider your rating on our work. We thank you again for your time and efforts in reviewing our paper. Please let us know if you have any further questions.
>
> Best regards, Authors

---

> ### Author Response · Authors · 2022-12-03
> **The end of the discussion phase is approaching**
>
> Dear Reviewer CSao,
>
> We appreciate your positive comments that our paper is well-written; our study is important and interesting; and our method has novelty and empirical advantage. We have made every effort to faithfully address all your comments in the response. Here, we briefly summarize the main points of our responses below:
>
> * We have recapitulated the novelty of our work, which tackles completely different problems compared to RAG [1], where the **challenges arise from a knowledge-consistent dialogue generation with knowledge graphs**. Moreover, we further have recapitulated the originality of our components in the answer for Q2.
> * We have clearly pointed out several important points that you may miss. Specifically, we have clarified that **all ablation studies and experimental results you suggested already exist in the paper**, and referred to them in the response.
>
> We have made every effort to faithfully answer all your comments in the responses. We strongly believe that, after reading our response, **you might no longer be inclined to reject our paper**. If you have any further questions or concerns, please let us know so that we can discuss and resolve your remaining concerns about our work.
>
> Best regards, Authors

---

> ### Comment · Reviewer_CSao · 2022-12-09
> **After reading response from authors**
>
> Thank the authors for detailed response. I have read the response from the authors. Some issues have been clarified. However, the response does not fully address my concerns. There remain two major concerns. I tend to keep my score.
>
> **1. This paper seems to belong to a kind of assembly innovation**
>
> I can understand the three points, stated by the authors for the contributions.
>
> 1. GNN based triplet embedding function for retrieval model
> 2. Invariant and Efficient Graph Encoding for generation model
> 3. Contrastive learning for overcoming exposure bias
>
> Suppose that we remove Point 3 and its related result, the main experimental results (such as in Table 1) will not be much affected. This is my judge whether a paper belongs to a kind of assembly innovation. Also, the three pieces are not strongly connected. You can remove one, and the overall performance  is not affected much (e.g., only degrading from 100% performance down to 95%, rather than from 100% performance down to 80%, which would indicate strong connection). Please see more in my separate comment below, responding to the authors' response.
>
> **2. More solid ablation studies**
>
> For example, the proposed Graph Encoding involves both PLM based entity embeddings $f(a)$ and GNN-based representation $\eta$ through an ad-hoc affine transformation as shown in Eq (7). Then, separately using $f(a)$ and $\eta$ alone should be important baselines to be compared. Eq (7) also seems to be very ad-hoc, and the paper does not provide any insight and ablation study. Also, there are some prior methods mentioned in Related Work Section. It is not clear from the paper whether this paper is the first in proposing GNN based triplet embedding function or not. Any prior methods on GNN-based entity embedding (not necessarily for dialog generation) should be important baselines.

---

> > ### Author Response · Authors · 2022-12-11
> > **Follow-up Response to Reviewer CSao**
> >
> > Thank you for replying back to us. We are glad that some of your concerns are resolved in the initial response. We hope your remaining concerns can be resolved in the following response.
> >
> > ---
> >
> > > **[Q1]**  This paper seems to belong to a kind of assembly innovation.
> >
> > We start this paper with a research question on how to design an **end-to-end knowledge-consistent dialogue generation framework**.  As you understand, the major components of our framework can be explained three-fold. We value your opinion, but we cannot fully agree with you that our components are not strongly connected. We believe that each component is necessary for building toward the knowledge-consistent dialogue generation framework. Table 1, 3, and 4 strengthen our belief that each of them takes part in the framework and becomes meaningful when they are all used together.
> >
> > Table 1 shows the overall performance of our proposed framework, Table 3 shows the impact of contrastive learning, and Table 4 shows the ablation study on various graph encoding methods. In this Q1, we will explain the experimental results in Table 1. In the following Q2 and Q3, we respectively have answered your concerns on graph encoding and contrastive learning.
> >
> > In Table 1, our proposed models have outperformed other baselines. However, as you mentioned, the performance gain of SURGE (contrastive) seems marginal compared to SURGE (semi-supervised). It can be misunderstood that contrastive learning is an unnecessary component in our framework. The purpose of SURGE (contrastive) is to overcome exposure bias by enforcing the model to learn the similarity between retrieved knowledge and the generated response. The generation performance is affected by the retrieval performance of the model. Since Table 1 reports the generation performance based on learned and retrieved knowledge of our model, which makes it hard to truly observe the effect of contrastive learning, Table 3 must be examined with it.
> >
> > ---
> >
> > > **[Q2]** More solid ablation studies on graph encodings.
> >
> > **We actually have shown exhaustive ablation studies on graph encoding in Table 4**. In Table 4, we demonstrated the case of separately using $f(a)$, which is referred to as *Efficient (entity only)*. The *Invariant & Efficient* is the case where we apply Equation (7) on *Efficient (entity only)*, which encodes GNN-based representation $\eta$ to PLM-based entity embeddings $f(a)$. The KQA scores show a prominent difference between using and not using Equation (7), which are *Invariant & Efficient* and *Efficient (entity only)*, respectively. The method integrating graph representation based on Equation (7) surpasses *Efficient (entity only)* graph encoding in KQA scores.
> > The results conform to the conclusion that our proposed permutation- and inversion-invariant yet efficient graph encoding is an effective method for augmenting knowledge into LMs.
> > To the best of our knowledge, we proposed a novel knowledge-augmenting method for pre-trained LMs. If you know of any existing paper related to ours that we need to compare, please suggest it to us.
> >
> > ---
> >
> > > **[Q3]** About SURGE (contrastive) performance.
> >
> > In Table 3, F1 is the unigram F1. We measured it along with KF1 to compare the word match between generated responses with the gold responses. We believe 0.13 is a comparable difference between SURGE (semi-supervised) and SURGE (contrastive). A similar F1 score indicates that SURGE (contrastive) generates responses with a similar word distribution as SURGE (semi-supervised). Both F1 and KF1 scores indicate that SURGE (contrastive)  faithfully reflects the augmented facts in the valid responses.
> >
> > ---
> >
> > We hope this response helps you to reconsider our work.

---

> ### Author Response · Authors · 2022-12-12
> **Follow-up Response to Reviewer CSao (to the response updated in the main review)**
>
> Thank you for your follow-up response. In this response, we faithfully address your remaining concerns below:
>
> ---
>
> > **[Q1]** Using $\eta$ alone should be an important baseline to be compared.
>
> Thank you for your suggestion. We believe that using $\eta$ alone can be one of the design choices to implement our “Invariant and Efficient Graph Encoding”.  However, we respectfully disagree with your opinion that our work should not be accepted because we did not include an ablation study on only using $\eta$ for the graph encoding.
>
> Nevertheless, **we are now conducting the ablation study you required**, but we are not sure if the experiment will be done before the end of the discussion stage. We hope the reviewer generously understands this timely situation.
>
> **--- Ablation study results (added 12/13) ---**
>
> |   | KQA EM  | KQA F1  | B-1  | B-2  | B-3  | B-4  | R-1  | R-2  | R-L  | F1 |
> |---|:---:|:---:|:---:|:---:|:---:|:---:|:---:|:---:|:---:|:---:|
> | $\eta$ only |  19.79  | 28.46  | 16.91  | 10.10  | 6.48  | 4.22  | 19.86  | 7.39 | 19.21  | 22.93  |
> | Ours (Equation (7))  | 51.00  | 57.63  | 17.70  | 11.21  | 7.61  | 5.28  | 21.43  | 8.85  | 20.57  | 25.07  |
>
> We provide the ablation study results the reviewer requested. In the Table above, we observe that only using $\eta$ as the token embedding for entities appended before the text decreases the performance. This result is obvious given that the whole replacement of the pre-trained token embedding adversely affects the performance of the pre-trained language model. This result also demonstrates why we modulate the token embedding of entities conditioned on $\eta$.
>
> ---
>
> > **[Q2]** If you think that Eq.(7) is the first in developing GNN-based entity embedding in the literature (not necessarily for dialog generation), then you can clearly say so in the paper. Suppose this is yes, then the contribution should be stated around this, rather than putting more emphasis on "Invariant and Efficient Graph Encoding". To my understanding, "Invariant and Efficient" are actually down to only encoding the unique entities, which essentially is entity embedding. Clarity issues remain.
>
> Since the goal of our work is not just to improve entity embedding methods, we respectfully disagree with your opinion.
>
> In Section 3.4, we target to effectively encode the graph into the generative Pre-trained Language Model (PLM), so that the PLM can generate the text conditioned on the given graph.
> In this circumstance, we assert that our work focuses on developing “to effectively encode graph-structural information into the PLM by considering permutation invariance and relation-inversion invariance properties of graph-structured data”.
> To accomplish this, we propose a novel graph encoding that prepends unique entities in front of the text and then modulates entities’ PLM token representations conditional on the graph-structural information from GNN.
>
> Therefore, we consider “developing better entity embedding methods” is a **highly orthogonal direction** to our work.
>
> ---
>
> > **[Q3]** About SURGE (contrastive) performance.
>
> Experimental results in Table 3 support that SURGE (contrastive) contributes to generating responses that faithfully reflect the given knowledge, which also means that contrastive learning prevents the dialogue model from generating hallucinated texts. We believe that this result is meaningful. On the other hand, SURGE (contrastive) performs the best in the KOMODIS dataset (see **Table 8**), which raises the possibility that contrastive learning could be beneficial in generating responses with more accurate knowledge.
>
> Moreover, we expect improving the retrieval performance will further maximize the effect of contrastive learning; therefore, we consider this as future work.
>
> ---
>
> We sincerely appreciate your comments as well as your responsiveness during the discussion period. We hope the above responses satisfactorily address your points, and please let us know if you have anything else we should address.

---

> > ### Comment · Reviewer_CSao · 2022-12-12
> > **Feedback to authors**
> >
> > [Q1&Q2] You may misunderstand my comment.
> >
> > Section 3.4 mainly consists of two parts. First, from the beginning to the last equation in page 5, the discussion about Invariance and Efficient boils down to prepending unique entities in front of the text. Second, developing Eq (7) - "modulates entities’ PLM token representations conditional on the graph-structural information from GNN". This is what I mean by "developing GNN-based entity embedding".
> >
> > The first part is phrased using an unnecessarily higher tone. Invariance and Efficient simply boils down to SORT(ENT(Z)). The development of the second part is more essential than the first part and needs more clarification and ablation. There is room to improving the presentation and phrasing in this paper.
> >
> > [Q3] The performance comparisons of contrastive vs semi-supervised/unsupervised in Table 3 (on OpendialKG) and Table 8 (on KOMODIS) are rather mixed. In Table 3, unsupervised performed best for many metrics, contrastive outperformed semi-supervised very marginally in KQA F1 (57.63 vs 57.70). In Table 8, contrastive indeed performed better, but still very marginally in KQA F1 (19.48 vs 19.50). From these numbers, it is hard to convince the readers for the efficacy of contrastive. It would be better to conduct significance test.
> >
> > Overall, this paper indeed did interesting study, but the above concern still remains. I would recommend 4.

---

> > > ### Author Response · Authors · 2022-12-13
> > > **Follow-up Response to Reviewer CSao (to Feedback)**
> > >
> > > Thank you for actively engaging in the discussion with us and acknowledging our work as an interesting study. We also appreciate your quick response, and are happy to respond to your comments.
> > >
> > > ---
> > >
> > > > **[Q1,2]** You may have misunderstood my comment. The second part (i.e., $\boldsymbol{\psi}^*$) is more essential than the first part (i.e., $\tilde{\boldsymbol{\psi}}$), and there is room to improve the presentation and phrasing in this paper.
> > >
> > > Thank you for your detailed feedback and suggestions. While the second part that leverages the relational information looks more important, we believe that the first part is also highly necessary since it makes graph encoding invariant and efficient, which is the fundamental concept that we study and focus in this work. However, we apologize if you feel that our language in the first part of Section 3.4 is unnecessarily high-toned, and we will make sure to revise the presentation and phrasing, which we believe requires minor revision.
> > >
> > > Also, thank you for acknowledging that the second part of Section 3.4 is essential. We already provided detailed explanations on it in Appendix D.1 with Figure 8, and also provided the comprehensive analyses in Table 4. We will more clearly describe and explain them in future revisions to ensure that the essential points are clear to readers.
> > >
> > > ---
> > >
> > > > **[Q3]** It is hard to convince the readers for the efficacy of contrastive learning with current experimental results. It would be better to conduct a significance test.
> > >
> > > We first would like to emphasize that contrastive learning is one variant of our model, which ensures faithful reflection of graph knowledge, and our SURGE with contrastive learning significantly outperforms all baselines, which is a significant contribution. Then, in comparisons between our models (e.g., semi-supervised and contrastive), we would like to clarify that KQA F1 (19.48 vs 19.50) might not be a convincing result to argue its efficacy. However, we believe that all the automatic evaluation metrics must be contemplated together. In Table 8, SURGE (contrastive) performs better in KQA EM (16.62 vs 17.30) by 0.68, and, in Table 3, SURGE (contrastive) performs better in KF1 (26.38 vs 27.58) by 1.20; therefore, we believe that there are huge differences between SURGE (semi-supervised) and SURGE (contrastive). Also, considering all the results (Table 1, Table 3, Table 8, and Figure 6), contrastive learning inclines to a meaningful component that assists the model in generating knowledge-consistent responses. We will emphasize the strong results on contrastive learning; furthermore, based on your advice, we will conduct a significance test.
> > >
> > > ---
> > >
> > > Thank you again for your valuable feedback.

---

### Official Review · Reviewer_KmrR · 2022-10-27

**Confidence:** 4
**Correctness:** 3
**Technical Novelty And Significance:** 4
**Empirical Novelty And Significance:** 4
**Recommendation:** 6

**Clarity, Quality, Novelty And Reproducibility:**

Paper was not an easy read.

Work looks novel and reproducible due to availability of source code.

Improvements in writing needed.

**Strength And Weaknesses:**


Strengths

* Empirically showed the importance of retrieving the more relevant subgraph knowledge rather than using all the relevant knowledge graphs when generating knowledge-grounded response

* Proposed multiple components for retrieval, encoding, and graph-text representation learning

* Better results than baselines on OpendialKG and KOMODIS datasets

* Source code is available for reproducibility


Weaknesses

* KQA correlation with human ratings is small

* Is human evaluation performed on only 30 examples?



**Summary Of The Paper:**

What is the goal of the paper?

* Knowledge-consistent dialogue generation with the KG

What has been done before?

* Many existing methods do not guarantee that the model utilizes a relevant piece of knowledge from the KG before generating knowledge-consistent dialogues. The ones which do simply match and retrieve all facts for entities that appear in the dialogue context, which either may mislead the agent to generate out-of-context responses from irrelevant facts or can increase the computational overheads for prepending tokens for all facts in PLMs. This work differs from those existing works, since it aims at retrieving only a context-relevant subgraph among all associated facts with a novel GNN-based subgraph retriever, which is end-to-end trainable along with a dialogue generation model.

What are the contributions of the paper?

* Propose SUbgraph Retrieval-augmented GEneration (SURGE), a end-to-end dialogue generation framework for generating context-relevant and knowledge-consistent dialogues with a KG that considers all aspects from knowledge retrieval, encoding, and reflection along the generation process.

* Introduced an additional performance metric, referred to as Knowledge-verifying Question Answering (KQA), which evaluates whether the generated responses contain the correct knowledge with an additional extractive question answering scheme.

* Proposed a GNN-based context-relevant subgraph retrieval method for KG-augmented dialogue generation, to extract only the relevant piece of the knowledge for the dialogue context from the entire knowledge graph.

* Proposed an invariant yet efficient graph encoder and a graph-text contrastive learning objective to ensure that the generated responses faithfully reflect the retrieved knowledge.

* Evaluated SURGE against relevant baselines on the OpendialKG and KOMODIS datasets , demonstrating its efficacy in generating responses that are more informative by retrieving and reflecting the relevant knowledge from the KG.






**Summary Of The Review:**

Paper has many strengths with scope of improvements in writing.


Strengths

* Empirically showed the importance of retrieving the more relevant subgraph knowledge rather than using all the relevant knowledge graphs when generating knowledge-grounded response

* Proposed multiple components for retrieval, encoding, and graph-text representation learning

* Better results than baselines on OpendialKG and KOMODIS datasets

* Source code is available for reproducibility


Weaknesses

* KQA correlation with human ratings is small

* Is human evaluation performed on only 30 examples?

---

> ### Author Response · Authors · 2022-11-15
> **Response to Reviewer KmrR**
>
> We sincerely appreciate your acknowledgment of our work and thank you for your feedback. We initially address all your few concerns below:
>
> ---
>
> > **[Q1-1]** The correlation between KQA and human rating is small.
>
> Thank you for your feedback. The small correlation between KQA results and human evaluation results might be because, the number of samples that we used for human evaluation is 30. However, considering the range of the Pearson correlation coefficient which is [-1, 1], we believe our correlation score 0.42 is sufficiently large, and enough for confirming that there exists a correlation between KQA and human rating.
>
> ---
>
> > **[Q1-2]** More samples are required for human evaluation.
>
> Due to the limited resource, we performed human evaluations with 30 samples for our initial submission. However, we will collect more human evaluation samples, and then re-report the human evaluation results, but also re-evaluate the correlation between KQA and human rating.
>
> ---
>
> > **[Q2]** Paper was not an easy read; requiring improvements in writing.
>
> **We have clarified some vague points, especially in Section 3 and 4, pointed out by other reviewers in the updated version**. If you come across any writing that needs to be clarified further, please let us know. We are happy to address it.

---

> > ### Author Response · Authors · 2022-11-19
> > **UPDATE in [Q1-2] More samples are required for human evaluation.**
> >
> > From your advice, we have conducted 100 samples for human evaluation. We recruited four annotators to evaluate the quality of 100 generated responses for each model. As the original result, SURGE significantly outperforms all other baselines (p-value < 0.05). We computed the agreement scores using Fleiss’ kappa coefficient, supporting the reliability of judgments made between the annotators. Also, we re-evaluated the correlation score between the KQA and human evaluation results. The correlation coefficient is 0.55 (previous:0.42), showing positive relationships between them.
> >
> > | Method | Consistency | Informativeness | Fluency |
> > |--------|--------|--------|--------|
> > | Agreement|  0.4364 | 0.6484 | 0.4348 |
> > | All Knowledge | 2.35 | 2.21 | 2.67 |
> > | Space Efficient| 2.43 | 1.94 | 2.51 |
> > | SURGE (our) | **2.56** | **2.44** | **2.77** |

---

> ### Author Response · Authors · 2022-12-03
> **The end of the discussion phase is approaching**
>
> Dear Reviewer KmrR,
>
> We sincerely appreciate your positive comments about the novelty of the problem, the strength of our proposed methods, the empirical efficacy of our method from the experimental results, and the reproducibility of our work. We have done our best to respond to other reviewers’ comments in the discussion phase, which have made our work more solid.
> Please let us know if you have any further questions.
>
> We thank you again for your time and efforts in reviewing our paper, as well as your constructive comments.
>
> Best regards, Authors

---

> > ### Comment · Reviewer_KmrR · 2022-12-09
> > **Thanks**
> >
> > Thanks for the author response. After reading author responses for my review and others reviews. I will like to keep my score unchanged.

---

### Official Review · Reviewer_ypx1 · 2022-10-31

**Confidence:** 4
**Correctness:** 4
**Technical Novelty And Significance:** 3
**Empirical Novelty And Significance:** 3
**Recommendation:** 8

**Clarity, Quality, Novelty And Reproducibility:**

This paper is very well written with sufficient justification on the claims made in this work. The paper is of high quality with certain novel proposals that might be beneficial for the community. The paper has sufficient details to reproduce the work and after a high level look on the code added, I can say that I might be able to reproduce it satisfactorily.

**Strength And Weaknesses:**

Strengths:
* This paper is really well written with extensive experiments and analysis to support the claims in the paper.
* Their approach and proposed metric can be a valuable contribution to the community in the space of knowledge grounded generation especially since it is more efficient than previous approaches using KGs.
*  They talked about their design choices and assumptions in detail.

Weakness:
* Some of the parts in this paper could be better explained, especially the new proposed metric in section 4. It was unclear what this metric is exactly.
* This paper lacks comparison with approaches that use commonsense KGs in a similar way.
* Using subgraphs is not entirely novel. (Wenhao Yu, Chenguang Zhu, Zaitang Li, Zhiting Hu, Qingyun Wang, Heng Ji, and Meng Jiang. 2022. A Survey of Knowledge-Enhanced Text Generation. ACM Comput. Surv. Just Accepted (March 2022). https://doi.org/10.1145/3512467 talk about it in much more detail), though the way the authors find a subgraph might be novel.
* The mapping function is unclear. Why did you split New York into New and York? Why not just use the base model's tokenizer? If you did use the tokenizer, then instead of calling this a "mapping function" you can simply refer it as "the entity name was directly passed to the PLM to get its encoding". It is also unclear how you did entity linking to get the KG entities from the context. Do you do exact match?
* What is a) in Figure 2 caption?
* From reading section 3.3 It is not clear if you're using all triplets present in the KG at each step to filter and obtain a subgraph. After reading the appendix (limitations) as well as line 2 in last para of 3.3, it's clear that you're using only the k hop neighborhood of the entities present in the context. Again, how do you do the linking? Also, wont you miss out on potential entities that are not exactly mentioned in the context? Did you consider also using relations? (eg: (Barack Obama, president of, Usa) is present in your KG but the context is "Who is the top-person of the american armed forces" or something, then you'll miss out on Barack Obama).
  * In the limitations, you do discuss limitations on entity linking but have not provided how you are doing the same (I might have missed it).
* Please describe the relation embedding matrix in the main paper section 3.3.
* Human evaluation using just 30 samples might not be sufficient for statistical significance. If possible, please try with 100 samples. In any case, please report statistical significance.
* The metric is only slightly correlated with human judgement and more evaluation of the metric is needed before it can be widely used.


Questions:
* How did you arrive at the 87% figure for facts that are irrelevant to the context in OpendialKG dataset (2nd para 4th line in Intro)?
* Why did you choose zero vector for initialization of node embeddings?
* What is EM that is reported in Table 1? ( I think you can fix this by describing the proposed metric a bit more).
* What is F1 vs KF1 in Table 3?


Grammar and nitpicks:
* The related works section lacks some discussion on works which use GNNs in different ways in dialogue systems. Some quick google search examples at the end.
* Figure 2 caption last line - "finally we use ~a~ contrastive learning to enforce".

Suggestions:
* Update the hyperlinks to Figures, and tables to include link in the "Figure" or "Table" keyword too instead of just linking the number.
* Please add more details in section 4.

Some other works that use GNNs in dialogue (from a quick google search) (I'm not asking you to cite them just that it would be nice to talk about other dialogue works that use GNNs) :
1) - Chen, L., Lv, B., Wang, C., Zhu, S., Tan, B., & Yu, K. (2020). Schema-Guided Multi-Domain Dialogue State Tracking with Graph Attention Neural Networks. Proceedings of the AAAI Conference on Artificial Intelligence, 34(05), 7521-7528.
2) Mauajama Firdaus, Nidhi Thakur, and Asif Ekbal. 2020. MultiDM-GCN: Aspect-guided Response Generation in Multi-domain Multi-modal Dialogue System using Graph Convolutional Network. In Findings of the Association for Computational Linguistics: EMNLP 2020, pages 2318–2328, Online. Association for Computational Linguistics.
3) W. Nie, R. Chang, M. Ren, Y. Su and A. Liu, "I-GCN: Incremental Graph Convolution Network for Conversation Emotion Detection," in IEEE Transactions on Multimedia, doi: 10.1109/TMM.2021.3118881.
4) X. Zhao, L. Chen and H. Chen, "A Weighted Heterogeneous Graph-Based Dialog System," in IEEE Transactions on Neural Networks and Learning Systems, doi: 10.1109/TNNLS.2021.3124640.
5) L. Qin, W. Che, M. Ni, Y. Li and T. Liu, "Knowing Where to Leverage: Context-Aware Graph Convolutional Network With an Adaptive Fusion Layer for Contextual Spoken Language Understanding," in IEEE/ACM Transactions on Audio, Speech, and Language Processing, vol. 29, pp. 1280-1289, 2021, doi: 10.1109/TASLP.2021.3053400.
6) Rishabh Joshi, Vidhisha Balachandran, Shikhar Vashishth, Alan Black, and Yulia Tsvetkov. 2021. Dialograph: Incorporating interpretable strategygraph networks into negotiation dialogues. In International Conference on Learning Representations.
7) [AttnIO: Knowledge Graph Exploration with In-and-Out Attention Flow for Knowledge-Grounded Dialogue](https://aclanthology.org/2020.emnlp-main.280) (Jung et al., EMNLP 2020)
8) Zhou, Li and Kevin Small. “Multi-domain Dialogue State Tracking as Dynamic Knowledge Graph Enhanced Question Answering.” ArXiv abs/1911.06192 (2019): n. pag.
9) C. -M. Wong et al., "Improving Conversational Recommender System by Pretraining Billion-scale Knowledge Graph," 2021 IEEE 37th International Conference on Data Engineering (ICDE), 2021, pp. 2607-2612, doi: 10.1109/ICDE51399.2021.00291.
10) Liu, Z., Niu, Z., Wu, H., & Wang, H. (2019). Knowledge Aware Conversation Generation with Explainable Reasoning over Augmented Graphs. EMNLP.
11) Jing Li, Qingbao Huang, Yi Cai, Yongkang Liu, Mingyi Fu, Qing Li, Topic-level knowledge sub-graphs for multi-turn dialogue generation, Knowledge-Based Systems, Volume 234, 2021,
12) Chen, Yu, Lingfei Wu, and Mohammed J. Zaki. "Toward Subgraph Guided Knowledge Graph Question Generation with Graph Neural Networks." arXiv preprint arXiv:2004.06015 (2020).

**Summary Of The Paper:**

This paper provides an approach to perform knowledge grounded dialogue generation. Their GNN based approach retrieves a relevant subgraph from the KG to efficiently generate an appropriate response. They also propose a permutation and relation inversion invarient yet efficient way to encode the graph information along with the text using a contrastive learning objective that ensures that the generated responses reflect the retrieved knowledge. Finally, they propose a novel metric called Knowlege-verifying Question Answering to evaluate whether the generated responses contain the correct knowledge.

**Summary Of The Review:**

This paper provides a novel approach to filter knowledge from KGs efficiently and make dialogue responses follow the retrieved knowledge.  This is a well written paper with adequately supported claims using extensive experiments and analysis. The metric proposed in this work can be used by the community for evaluating knowledge grounded generation better. The strengths far outweigh the weaknesses, which are mostly to do with clarifications.

---

> ### Author Response · Authors · 2022-11-11
> **Response to Reviewer ypx1 (3/3)**
>
> > **[Q13]** Other works [1-12] that use graphs in dialogue generation would be nice to discuss.
>
> Thank you for your thoughtful suggestions. We note that, including the works you suggested, there is a vast amount of literature that tackles dialogue-related tasks with knowledge graphs; however, most of them are clearly different from ours. To be specific, we focus on knowledge-consistent dialogue generation, which we tackle with the subgraph-retrieval-augmented generation, invariant graph encoding, and contrastive learning, while others are not, as follows:
> * Some methods [1, 3, 8, 9, 12] **do not deal with dialogue generation**, which is completely different from ours that aims to generate responses for the given conversation.
> * Some methods [2, 4, 5, 6] are **not concerned with the knowledge graph**, instead 1) using particular graph data, for example, graphs of natural texts and images [2], or symptom occurrence-based heterogeneous graphs [4], or 2) using particular graph structures, such as utterance graphs [5, 6]. Therefore, our main motivation and focus on how to retrieve the external facts in the knowledge graph, and how to inject the retrieved facts into the language models are clearly different from them.
>
> While some works [7, 10, 11] consider dialogue generation with knowledge graphs, they are also clearly different from our work, in both motivational and detailed technical perspectives, which we explain one by one as follows:
>
> * First of all, [7] proposes the path traversal scheme over the knowledge graph, to reach the answer fact for the dialogue context; however, **it does not handle dialogue generation**. However, our main focus is on generating the dialogue responses faithfully with respect to the retrieved knowledge.
> * [10], similar to the aforementioned work [7] that aims at relevant path traversal over the knowledge graph, proposes to directly maximize the path that has answer entities for the dialogue history with reinforcement learning, to select the relevant paths. While this work [10] additionally generates the response for the conversation, similar to the work (Zhou et al., 2021) namely EARL that we already discussed and compared, **it does not consider language models and does not explicitly augment the retrieved knowledge in the model input**. Note that, language models often have a large number of parameters with internalized knowledge during pre-training, and such points yield entirely new challenges in effectively reflecting the knowledge from knowledge graphs, which we tackle with subgraph-retrieval-augmented generation along with graph encoding and graph-text contrastive learning.
> * [11] proposes to decompose the entire knowledge graph into particular topic-level subgraphs with manually defined rules, and then use the topic-level subgraphs for topic-aware dialogue generation. Also, similar to the aforementioned work [7] and EARL (Zhou et al., 2021), **this work [11] uses RNN-based architectures** for response generation. In contrast to them, our work retrieves the context-relevant subgraph in an end-to-end fashion, and also augments the extracted subgraph directly in the input of pre-trained language models.
>
> We will discuss them in the next revision.
>
> ---
>
> > **[Q14]** A typo in the caption of Figure 2.
>
> Thank you for pointing it out, and we have fixed it during the revision.
>
> ---
>
> > **[Q15]** For hyperlinks of Figures and Tables, please make "Figure" and "Table" also clickable, instead of only for numbers after them.
>
> Thank you for your suggestion, and we have updated them during the revision.
>
> ---
>
> ### References
>
> [A] Rajpukar et al., SQuAD: 100,000+ Questions for Machine Comprehension of Text, EMNLP 2016.
>
> [B] He et al., Generating Natural Answers by Incorporating Copying and Retrieving Mechanisms in Sequence-to-Sequence Learning, ACL 2017.
>
> [C] Eric et al., A Copy-Augmented Sequence-to-Sequence Architecture Gives Good Performance on Task-Oriented Dialogue, EACL 2017.
>
> [D] Shuster et al., BlenderBot 3: a deployed conversational agent that continually learns to responsibly engage, preprint 2022.
>
> [E] Thoppilan et al., LaMDA: Language Models for Dialog Applications, preprint 2022.
>
> [F] Yamada et al., LUKE: Deep Contextualized Entity Representations with Entity-aware Self-attention, EMNLP 2020.
>
> [G] Zhang et al., ERNIE: Enhanced Language Representation with Informative Entities, ACL 2019.

---

> ### Author Response · Authors · 2022-11-11
> **Response to Reviewer ypx1 (2/3)**
>
> > **[Q6]** What is "a)" in the caption of Figure 2.
>
> The abbreviated symbol "a" indicates the entity "Jane Austen" in the knowledge graph.
>
> ---
>
> > **[Q7-1]** Please specify the relation embedding matrix in Section 3.3.
>
> As described in the Retriever Details paragraph of the Appendix D.1, **the relation embedding matrix is the trainable embedding matrix** for every relation in the knowledge graph, i.e., each relation has its own trainable vector, which is initialized with random. We have included more details in Section 3.3 as well, during the revision.
>
> ---
>
> > **[Q7-2]** Why do you use the zero vector for initialization of node embeddings in Section 3.3?
>
> As formalized in Equation 4, if the entity exists in the dialogue history, we initialize its node embedding with its token representations in LMs. However, if the entity does not exist in the dialogue history, we use the zero vector for embedding initialization.
>
> Note that, for the initialization of entities, similar to the relation matrix and similar to the previous works [F, G], we might use the additional trainable matrix for all entities. However, the usage of the additional entity embedding matrix has the following drawbacks:
> * Efficiency issue: the number of all entities (100,814) within a large KG (e.g., Freebase) is relatively large, compared to the number of all tokens (32,000) in T5, a LM. We discuss this issue in details, in the Retriever Details paragraph of the Appendix D.1.
> * Generalization issue: All 100k entities in the KG are less likely to appear at least one time in the training dataset, and many entities appear only a few times. Therefore, entity embeddings stored in the entity embedding matrix might be suboptimal.
>
> To avoid the above two issues, we decide to use the output of the LMs as the node embedding for the entity, if the entity exists in the input text of LMs (See Equation 4). Otherwise, we use the zero vector for entity embedding initialization. Note that, based on the recursive updates of GNNs, all entities have their proper node embeddings, since the zero-initialized entity also receives the meaningful representation from its neighborhood entity representations.
>
> We have further clarified this point in Section 3.3, during the revision.
>
> ---
>
> > **[Q8-1]** Human evaluations with 30 samples might not be sufficient, and I recommend to use 100 samples.
>
> Due to the limited resource, we performed human evaluations with 30 samples for our initial submission. However, we will collect more human evaluation samples, and then re-report the human evaluation results.
>
> ---
>
> > **[Q8-2]** For human evaluation results, please report the statistical significance.
>
> In Table 5, we already marked only the statistically significant results with p-value < 0.05 as bold, described in the Human Evaluation paragraph of Section 5.4. However, this was missing in the caption, and we have included the p-value in the caption, during the revision.
>
> ---
>
> > **[Q9]** The KQA metric is slightly correlated with the human judgement, therefore, more evaluation is required for verifying this metric.
>
> The small correlation between KQA results and human evaluation results might be because, the number of samples that we used for human evaluation is 30. However, considering the range of the Pearson correlation coefficient which is [-1, 1], we believe our correlation score 0.42 is sufficiently large, and enough for confirming that there exists a correlation between KQA and human rating.
>
> As described in the response of **[Q8-1]**, after collecting more human evaluation results, we will recompute the correlation between KQA and human rating, to more concretely verify the effectiveness of the KQA metric.
>
> ---
>
> > **[Q10]** How to calculate the number of irrelevant knowledge for the dialogue context, which is 87%, in Figure 1?
>
> We calculate (the number of gold facts / the number of facts within 1-hop KG) for dialogue contexts, and then average them.
>
> ---
>
> > **[Q11]** What are the differences between F1 and KF1 in Table 3?
>
> F1 and KF1 are both unigram metrics that compute word overlapping between the generated response and the reference text, where the reference text could be **either the gold knowledge (KF1) or the correct response (F1)**, as explained in Section 5.3. In other words, the KF1 can measure whether the augmented knowledge is reflected in the generated response, meanwhile, the F1 measures word overlapping between the generated response and the labeled response without particularly validating the knowledge-consistency.

---

> > ### Author Response · Authors · 2022-11-19
> > **UPDATE in [Q8-1] Human evaluations with 30 samples might not be sufficient, and I recommend to use 100 samples.**
> >
> > We have collected 100 samples. We recruited four annotators to ask them to rate the quality of 100 generated responses for each model. We follow the original human experiment setting, where we randomly shuffle the generated responses and use a 3-point Likert-like scale (1.Disagree, 2.Neutral, and 3.Agree). SURGE achieve significantly high scores in all three criteria, consistency, informativeness, and fluency as the original result (p-value < 0.05).  Also, we measured Fleiss’ kappa coefficient to quantify consistency between the annotators’ decisions. The agreement scores support the moderate reliability of our human evaluation results. Lastly, the correlation score between the KQA and human evaluation result is 0.55 (previous:0.42), indicating a much stronger association between them.
> >
> > | Method | Consistency | Informativeness | Fluency |
> > |--------|--------|--------|--------|
> > | Agreement|  0.4364 | 0.6484 | 0.4348 |
> > | All Knowledge | 2.35 | 2.21 | 2.67 |
> > | Space Efficient| 2.43 | 1.94 | 2.51 |
> > | SURGE (our) | **2.56** | **2.44** | **2.77** |

---

> ### Author Response · Authors · 2022-11-15
> **Response to Reviewer ypx1 (1/3)**
>
> We sincerely appreciate your insightful comments. We believe that the quality of our work has improved a lot by reflecting your comments in the revision. We initially address all your concerns/comments below:
>
> ---
>
> > **[Q1-1]** The new metric (i.e., KQA) proposed in Section 4 could be further clarified.
>
> We apologize for the confusion, and thank you for pointing it out.
>
> The goal of KQA is **to measure whether the correct answer knowledge exists in the generated response**. For this goal, we first construct the extractive QA dataset, which consists of the dialogue history with the gold or acceptable response as the context, the question and target answers from the corresponding triplet, illustrated in Figure 3.
> Then, we train the extractive QA model on this synthetic QA dataset, to measure the existence of the correct knowledge in the generated response with F1 and Exact Match scores as evaluation metrics.
>
> During the revision, **we have clarified Section 4** by additionally describing how the proposed KQA metric is used to evaluate the existence of the knowledge in the generated response. Also, for more details, please refer to Appendix D.
>
>
> ---
>
> > **[Q1-2]** What is KQA - EM that is reported in Table 1?
>
> EM indicates the **exact match score**, which checks the exact string match between the labeled answer and the classified answer, and which is commonly used in the extractive question answering tasks [A].
>
> ---
>
> > **[Q2]** This paper lacks comparison with approaches that use commonsense KGs.
>
> Thank you for your suggestion. However, we already discussed some prior works that use commonsense KGs in the Related Work section. In addition, **we already compared EARL as a baseline**, which is one of the most recent methods similar to prior works using commonsense KGs.
>
> ---
>
> > **[Q3]** Subgraph retrieval is not entirely novel, while the proposed mechanism for subgraph retrieval might be novel.
>
> We respectfully agree with your comment that, the problem of subgraph retrieval is not entirely novel and has been actively studied in previous works on different domains, such as question answering [B] and dialogue generation [C].
>
> However, our main novelty is not suggesting the subgraph retrieval problem, but rather, as you acknowledged, **our novelty is on how to tackle this problem for knowledge-consistent dialogue generation**. Specifically, we tackle this problem under the context-relevant subgraph-retrieval-augmented generation setting, where we use graph neural networks with LMs, as well as graph-encoding and graph-text contrastive learning schemes. Therefore, we believe such the approach for the subgraph retrieval problem has the solid contribution for dialogue generation literature, given that recent existing dialogue application works consider how to use pre-trained LMs [D, E].
>
> ---
>
> > **[Q4]** The mapping function, which maps entities/relations to their tokens, is unclearly described. Do you use the tokenizer for mapping?
>
> We apologize for the confusion, and thank you for pointing it out. As you understood, we use the tokenizer to map the entities/relations to their tokens. We also agree with you that the mapping function can be simply considered as the tokenizer for LMs, and we have clarified it in the revision.
>
> ---
>
> > **[Q5-1]** Do you use all triplets in the KG, or use the k-hop neighborhood of the entities in the dialogue context?
>
> We consider k-hop neighborhoods of the entities in the dialogue context as the candidate triplets to retrieve, as described in Section 5.1.
>
> ---
>
> > **[Q5-2]** How do you perform entity linking to get the KG entities from the dialogue context? Do you use exact match?
>
> As described in Appendix D.2, we use the provided knowledge if they are available in the dataset; and also use the existing NER module to identify the entity in the dialogue context first and then use the **exact string matching** for entity linking. We have further clarified this point in Appendix D.2 during the revision.
>
> ---
>
> > **[Q5-3]** During the entity linking, do you miss out potential entities that are not exactly mentioned in the context? Also, do you consider relations for candidate triplet selections?
>
> As you understood, we might miss out potential entities, due to errors in named entity recognition and exact matching for entity linking. For example, our scheme cannot capture the aliases or acronyms of the entity. Also, we do not additionally consider relations for candidate triplet selections. However, all the other baselines also suffer from such the same issue in experiments, and the reason we perform entity linking is to capture more entities as possible as we can. We believe improving the entity linking for dialogue generation is out-of-scope of our research, which we leave as the future work.

---

> > ### Comment · Reviewer_ypx1 · 2022-11-23
> > **Response to Authors response to my initial review**
> >
> > I thank the authors for taking my feedback constructively and improving their work.
> >
> > Q 1/ 2 - After going over figure 3, section 4 and appendix D, I'm not even more confused what this novel metric is and what exactly is the training data for this. Why is the question in a triplet format? What are "answer candidates"? Can you clarify what exactly is the input and what is the output of this model here in a comment?
> >
> > Q 3 - I thank the authors for clarifying that the idea of subgraph retrieval is not novel. The only novelty in this case might be that this was used for KG augmented dialogue response generation.
> > From what I understand, for each dialogue example, a subgraph is identified based on the similarity between the triplets and the dialogue context. The similarity is not calculated based on the whole KG but based on a predefined filtered set of triplets (k hop distance). And that's just 1 hop or 2 hop.
> >
> > Note: I no longer feel that this is a "significant" contribution of this work and feel that the claim of identifying a subgraph in the KG is a bit of an exaggeration. It's more like filtering triplets in the 1/2 hop neighborhood.
> >
> > I am also concerned about the number of layers that the graph neural network had to embed the entities / relations? If just 1 layer, then that's just using the direct edges and not really using any "graph structure (multi hop information)" to form these encodings.
> >
> > I'm also now a bit less convinced about the novel "graph encoder" claim because there's no graph structure information in a 1 hop neighborhood.
> >
> > After reading some of the other reviews, I'm also a bit curious on what is the need for relation invariance? Can the need for this be motivated in the paper. Permutation invariance makes sense as you want the same encoding for the same input.
> >
> > Q4 - I thank the authors for clarifying that the "mapping" function is not novel and for clarifying the same in the paper
> >
> > Other questions - Thanks for clarifying and thanks for adding more human evaluation. Regarding the related works, I would like the authors to clarify with the other reviewers since I'm not aware of the related works in a lot of detail.
> >
> > Currently, I would still like to recommend this paper be accepted (if the novelty/related works aspects are properly justified to the other reviewers). I'm slightly inclined towards decreasing my score from an accept to a borderline accept for these reasons, and i'm strongly inclined towards decreasing my confidence to 3.
> >
> > TLDR of the review - I feel the authors have shown the efficacy of their method clearly and have detailed descriptions, but I do agree with some of the other reviewers that the novelty in this work seems to be in getting a combination of various prior approaches working in this setup.

---

> > > ### Author Response · Authors · 2022-11-25
> > > **Follow-up Response to Reviewer ypx1 (3/3)**
> > >
> > > > **[Q3-2]** Remaining concerns on method designs.
> > >
> > > >> **[Q3-2-1]** When considering 1 layer graph neural networks, or 1-hop neighborhood information, the model does not really capture graph-structural information, which is a bit less convincing.
> > >
> > > **A single-layer graph neural network can capture graph-structural information**. For example, in a graph with two edges: { (a, r1, b), (a, r1, c) }, a single message-passing step represents both nodes ('b' and 'c') with the information from the relational-structures, coming from relation ('r1') and entity ('a'). Similarly, in a 1-hop neighborhood, there exist graph structures to reflect. For instance, for a graph with two edges: { (a, r1, b), (a, r1, c) }, the single message passing makes both nodes ‘b’ and ‘c’ be represented with the relational structures from relation ‘r1’ and entity ‘a’.
> > >
> > > Note that multi-hop is just one type of graph structure to consider, and considering 1-hop with 1-layer GNNs can reflect a meaningful graph structure. To experimentally demonstrate the effectiveness of our graph encoder, we also conduct ablation studies in Table 4, and we kindly suggest referring to the results on it. In particular, Efficient (entity only) in Table 4 indicates the model where we only append entities to the text input, while Invariant & Efficient indicates the model with our novel graph encoding: token embedding perturbation generated by the knowledge structure as shown in Equation 7 and Figure 8. The model with our graph encoding outperforms the entity-only variant, according to the results (57.63 vs 49.99 in KQA F1). This result implies the significance of using graph structures in graph encoding, and the capability to capture the graph structure even with a single layer of GNN.
> > >
> > > >> **[Q3-2-2]** Why is relation invariance needed?
> > >
> > > Please see the answer for question 4 for the Reviewer YCpx, where we already answered there. Here we shortly summarize why it is necessary. PLMs may embed two semantically identical triplets (a, born-in, b), and (b, ~born-in, a), into the different representations since PLMs are permutation sensitive. We therefore give the same input sequence [a, born-in, b, b, ~born-in, a] to the PLMs for both triplets: (a, born-in, b), and (b, ~born-in, a), with SORT and INV operations, to ensure that the representation of (a, born-in, b) is the same as the representation of its inverse. We also will further clarify this in the next revision.
> > >
> > > ---
> > >
> > > > **[Q; ETC]** Clarification on related works, suggested from other reviewers.
> > >
> > > We already clarified that the related works, suggested by other reviewers, are significantly different from ours. We hope other reviewers would check our responses, which clarify that our ideas/components are different from existing works, and then respond to us.
> > >
> > > ---
> > >
> > > We hope that our responses address your concerns. Please let us know if there are any points that we misunderstood from your comments, or if you have any other questions or suggestions. We will be more than happy to address them.
> > > Thank you.
> > >
> > > ### References
> > >
> > > [1] Tuan et al., Towards large-scale interpretable knowledge graph reasoning for dialogue systems, Findings of ACL 2022.
> > >
> > > [2] Lewis et al., Retrieval-augmented generation for knowledge-intensive NLP tasks, NeurIPS 2020.
> > >
> > > [3] Galetzka et al., Space efficient context encoding for non-task-oriented dialogue generation with graph attention transformer., ACL 2021.
> > >
> > > [4] Humeau et al., Poly-encoders: Architectures and pre-training strategies for fast and accurate multi-sentence scoring, ICLR 2020.

---

> > > ### Author Response · Authors · 2022-11-25
> > > **Follow-up Response to Reviewer ypx1 (2/3)**
> > >
> > > *(continued)* Lastly, we clarify what exactly answer candidates are, and how multiple answers could exist. As in the example of Figure 3, there are many other books written by “Jane Austen”, including “Lady Susan” in the gold response. Then, for this particular example, we search the triplets, such as { (Lady Susan, write, Emma), (Lady Susan, write, Sense & Sensibility) }, which have the same head and the relation to the identified triplet, in order to find such acceptable candidates that are not covered by the gold response annotated by humans. In other words, for this example, the candidate answers for the KQA are the tail entities from the identified triplet. Furthermore, to additionally avoid the situation where the KQA metric is high even though the generated responses do not contain accurate knowledge, we remove answers that already exist in the user's query in the candidate answers. Finally, when building the synthetic QA dataset for training the KQA model, we append the gold response instead of using the generated response.
> > >
> > > We hope that our answer clarifies your questions on KQA. We will add more details with illustrations in a future revision. Let us know if you have any further questions or anything else that we should address.
> > >
> > > ---
> > >
> > > > **[Q3-1]** The idea of subgraph retrieval is not novel.
> > >
> > > Thank you for initiating the discussion on the novelty. However, subgraph retrieval tasks is an existing problem, like question answering and dialogue generation tasks, and we do not claim that we proposed a novel problem. Rather, our novelty is in **how to retrieve and augment facts in subgraphs, to tackle challenges on dialogue generation**.
> > >
> > > More specifically, if “the idea” refers to the subgraph retrieval problem, we agree with you since we are not the first attempt at solving this problem. However, our method to tackle the problem, **which retrieves context-relevant subgraphs for knowledge-consistent generation, is highly novel**. Specifically, we proposed a graph neural network (GNN) -based subgraph retrieval method, which is jointly trained with dialogue generation loss in an end-to-end manner with weak supervision. This is clearly different from previous works on subgraph retrieval or reasoning path prediction [1]. Further, we utilize a GNN to reflect the graph structure, when embedding triplets in the Knowledge Graph (KG), which is also clearly different from the previous work on the retrieval-augmented generation with the text [2].
> > >
> > > Also, it is worthwhile to mention that, along with the novelty of the subgraph retrieval mechanism, there are other novel contributions in our proposed framework, which have, to the best of our knowledge, not been explored so far in the literature on KG-based dialogue generation. We recapitulate the originalities of three proposed components, as follows:
> > >
> > > * **(GNN-based subgraph retrieval for KG-augmented dialogue generation)** We propose to utilize GNNs to effectively and efficiently retrieve graph-structured data in the k-hop KG, while leveraging both node and edge representations, which is novel.
> > > * **(Permutation- and inversion-invariant yet efficient graph encoding)** We suggest that by taking into account two invariance properties of graph-structured data, it is possible to effectively encode graph-structural information into Pre-trained Language Models (PLMs). This approach, which has not previously been discussed in the literature, is a novel scheme for encoding graph-structured knowledge with PLMs.
> > > * **(Graph-text contrastive learning for knowledge-consistent generation)** We propose graph-text contrastive learning, which has, to the best of our knowledge, not been explored for a knowledge-consistent generation; that is proposed to ensure that the knowledge from retrieved subgraphs is reflected in the generated text-response.
> > >
> > > Note that, in our experiments, we do successfully validate each component's effectiveness. In particular, by comparing our approach with baselines using the entire k-hop KG, we demonstrate how inefficient the baselines (All knowledge, Space Efficient [3]) are, in Table 1. We assert that these findings demonstrate the significance of retrieving context-relevant subgraphs even from 1-hop or 2-hop KG, for dialogue generation. Furthermore, Table 1 and Figure 4 clearly demonstrate that the use of GNNs in the triplet embedding is essential to handle the graph-structured knowledge by comparing our retrieval method against the already-existing dense retrieval method [4], for subgraph retrieval. Additionally, Table 2 and Table 3 demonstrate how well our suggested graph encoding performs in terms of generation performance; And how well contrastive learning performs in terms of knowledge-consistent response generation.

---

> > > ### Author Response · Authors · 2022-11-25
> > > **Follow-up Response to Reviewer ypx1 (1/3)**
> > >
> > > We sincerely appreciate you for follow-up our response, and providing additional comments. In this response, we address all your remaining concerns/comments in your response below:
> > >
> > > ---
> > >
> > > > **[Q1/2]** KQA metrics are still confusing. Why is the question in a triplet format? What are the answer candidates? Can you clarify what exactly is the input and what is the output, for the KQA model?
> > >
> > > We first want to clarify that the **KQA scores are measured by an extractive Question Answering (QA) model**, which is a BERT-based model fine-tuned on the synthetic QA data. Note that, for the given question, the extractive QA model predicts the **span of the answer** within the context during evaluation. Then, in our KQA case, given a partial triplet (e.g., head entity and relation) as a question, the goal is to predict the missing entity (e.g., tail entity) in the given context. Since, in our KQA case, the context consists of the dialogue context and the generated response, if the generated response contains the correct answer (e.g., the tail entity for the partial triplet), the KQA model can select it. In other words, if the KQA model can select one of the answer entities existing in the generated response in the context, we assume and measure the generated answer is factually correct since the KQA model finds that the answer knowledge (i.e., entity) is in the response.
> > >
> > > Therefore, the question is in a triplet form, since the objective of the KQA model is to predict the missing part (i.e., entity) of the given triplet, and answer candidates are a set of entities that should exist in the generated response, which the KQA model predicts.
> > >
> > > ---
> > >
> > > To further clarify the mechanism of the KQA, here we provide **an example**. Let's take an illustration depicted in **Figure 3** as an example. The user asks the question, “Could you recommend a book written by Jane Austen?”, and the desired (gold) answer is “Sure, Lady Susan.”. Then, in this case, we can extract entities from each question and the gold response with the help of existing entity extraction/linking tools. In particular, the user's query contains the entity "Jane Austen," and the gold response contains the entity "Lady Susan." Then, we can look up the triplet from the 1-hop KG of the entity "Jane Austen", which has "Lady Susan" as the tail entity and "Jane Austen" as the head entity. Then, the identified triplet (Jane Austen, write, Lady Susan) is identified as a “Corresponding Fact” in Figure 3. From this triplet, we define the question for the KQA: “Jane Austen write ?”.
> > > On the other hand, the context for the KQA is created by concatenating the user's query and the generated response. Then, from the given question and the context, a trained QA model can now extract the appropriate span in the input context, if the generated response contains the candidate answers (i.e., correct knowledge) consisting of possible tail entities of the triplet (Jane Austen, write, …). If the candidate answers are missing from the generated response, the QA model might not be able to find the answer, and we regard the generated response as not having the correct knowledge.
> > >
> > > Consider the scenario where our SURGE model returns the response "She wrote Harry Potter.", for instance. Then, the context, question, and answer candidates are as follows:
> > >
> > >     Context: “Could you recommend any book by Jane Austen? **She wrote Harry Potter**”
> > >     Question: “Jane Austen write ?”
> > >     Answer Candidates: [“Lady Susan”, “Sense & Sensibility”, “Emma”]
> > >
> > > In this case, the QA model attempts to predict the span position of the answer within the input context and question. However, since the answer does not exist in the context, the QA model cannot identify the correct answer, and the KQA metrics (EM/F1) become 0. On the other hand, what if the generated response is “She wrote Emma.”? Since "Emma" is one of the answer candidates and does present in the context, the QA model has a chance to identify the answer entity, and, if it identifies, KQA scores (EM/F1) become 100.

---

> > > > ### Comment · Reviewer_ypx1 · 2022-12-09
> > > > **Response to the metric clarification.**
> > > >
> > > > Hi,
> > > > I'm still not convinced with the validity of this metric. I feel a metric should not behave differently with different datasets / runs. It might be better for you to just use an off the shelf QA system rather than fine-tuning it on your synthetic dataset. One concern I have with this setup is that the finetuned model could just learn to predict the span present in the candidate answer in the input with some spurious correlations like position. Not finetuning on your setup might help alleviate this issue. A better way might even be to extract this triplet from the context+candidate answer and just check if it's present in the KG or not.
> > > > In your Harry Potter example, what happens if the model just learns that the answer to the question is present somewhere towards the end of the context, and just returns "Harry Potter"? It's not present in the answer candidates extracted from the KG. Do you give a 0? If yes, why have a QA system, why not just string match to see if any of the candidates are present in the response?
> > > > Since this is a significant part of this work (evaluation of actual usage in knowledge grounded generation), I'm not super confident that this can be accepted without a more detailed analysis of the metric. Hence, unfortunately, I'm unwilling to champion for this paper though I'd like to keep my score unchanged because of the clarifications on the other aspects.
> > > > Thanks a lot for the detailed and engaging discussion on this paper.

---

> ### Author Response · Authors · 2022-12-03
> **The end of the discussion phase is approaching**
>
> Dear Reviewer ypx1,
>
> We appreciate your positive comments that our paper is very well written with sufficient justification on the claims made in the work, and of high quality with certain novel proposals that might be beneficial for the community. Moreover, we sincerely appreciate your follow-up comments about your remaining concerns. We have made every effort to faithfully answer all your comments in the responses and hope that our responses clearly address your concerns.
>
> We are pleased to discuss with you since the discussion does help us to improve our paper. We thank you again for your time and efforts in reviewing our paper, as well as your constructive comments. Please let us know if you have any further questions.
>
> Best regards, Authors

---

> ### Author Response · Authors · 2022-12-11
> **Follow-up Response on the metric to Reviewer ypx1**
>
> Thank you for following up our response. We are more than happy to discuss with you again before the end of the discussion stage.
> In this response, we faithfully  address your remaining concerns on the metric below:
>
> ---
>
> We believe that your major concern on our KQA metric is about the false negative case, where the trained QA model fails to identify the answer candidates even if they are present in the generated response. We agree with your concern that the metric based on the trained neural network is not perfect; however, the evaluation setup is fair enough since we use the same QA model when evaluating all baselines and our methods.
>
> We also agree with you that the QA system is not the only method to evaluate the knowledge groundedness of the generated response. Since the novelty of the KQA metric stems from the use of the KG to resolve the issue from the missing knowledge by only considering the gold response for evaluation, we can also utilize rule-based metrics like **string matching** as you suggested (check whether at least one of the answer candidates in KQA presents in the generated response) or **Entity F1** score (measuring the F1 score against each entity in answer candidates instead of the gold response).
>
> As we all know, automatic evaluations can be imperfect when compared to human evaluations. However, we also believe that using a variety of credible automatic evaluation metrics will strengthen the validity of the experimental results. Therefore, we supplement the experimental results with three more evaluation metrics for measuring whether the generated responses contain appropriate knowledge.
>
> | | Knowledge F1 | Entity F1 | String Matching |
> |--------|--------|--------|--------|
> | No knowledge | 13.80 | 9.33 | 13.03 |
> | Random knowledge | 16.29 | 16.49 | 32.71 |
> | All knowledge | 18.91 | 21.10 | 44.25 |
> | Space Efficient (Parallel) | 17.43 | 18.93 | 40.56 |
> | Dense Retrieval (Poly-encoder) | 19.72 | 21.46 | 48.41 |
> | DiffKG | 14.44 | 9.37 | 13.23 |
> | SURGE (Ours, semi-supervised) | **21.87** | **23.03** | **55.79** |
> | Gold Response (oracle) | 28.62 | 29.06 | 85.75 |
>
> In Table, we measure Knowledge F1 (KF1 in Table 3), string matching, and entity F1 for representative baselines and our method in OpendialKG dataset, as an extension of Table 1. For KF1, we measure the F1 score regarding the concatenation of the question (head entity and relation) and all answer candidates (available tail entities) in KQA as the gold response. The results show that all metrics show the same tendency with KQA and our proposed method still outperforms other baselines by generating responses with more proper knowledge.
>
> Although three rule-based metrics are useful for assessing the knowledge groundedness of generated responses, they do have some drawbacks. KF1 and Entity F1 are affected by the length of the generated response and answer candidates. String matching is too strict since it may miss some responses that only contain partial words of knowledge (e.g., the response only contains the first name of the author whereas the candidate answers contain the full name of the author). As a result, the use of KQA is also beneficial since the trained QA model can compensate for the shortcomings of rule-based metrics.
>
> ---
>
> We hope that our answer clarifies your concerns about the metric. We will add the above experimental results with three more metrics for all baselines and our methods in a future revision. We hope this can consolidate the validity of our experimental results on the evaluation of actual usage in a knowledge-grounded dialogue generation. We appreciate the reviewer ypx1 for valuable comments which further strengthen our paper.
>
> Please let us know if you have any further questions, or anything else that we should address.

---

### Official Review · Reviewer_p6ze · 2022-11-04

**Confidence:** 3
**Correctness:** 2
**Technical Novelty And Significance:** 2
**Empirical Novelty And Significance:** 2
**Recommendation:** 3

**Clarity, Quality, Novelty And Reproducibility:**

#### Clarity:
The paper is overall clear to get the idea.

#### Quality:
The paper overall has a good quality for writing, method design, and experiments given its current scope.

#### Novelty:
The paper misses some highly relevant works, which did dialogue generation by attending only parts of a KG. These methods are not discussed or compared in this paper.

#### Reproducibility:
The paper could be reproducible given the provided code. For the paper side, the writing of Section 3-4 can be more clear for reproducibility. For example, the first time defining $y_{<t}$ may be changed to define $y_T$ first, which was also mentioned afterwards but was not defined. Furthermore, about the KQA evaluation, I’m not sure if you calculated the KQA metric with another trained model (BERT according to the appendix) or you used the same dialogue generation model but utilized a mismatched input format to query the output? Both seem a bit weird to me.


**Strength And Weaknesses:**

### Strengths
* The paper is overall written clearly.
* The proposed method is a thoughtful combination of important details such as the invariant graph encoding and graph-text contrastive learning, which convincingly can help the training.
* The proposed evaluation metric KQA could be useful.

### Weaknesses
* The paper misses references to prior works on dialogue generation priorly or jointly extracts subgraphs, which to my understanding are closely related to this work, such as:
  * Jung, Jaehun, Bokyung Son, and Sungwon Lyu. "Attnio: Knowledge graph exploration with in-and-out attention flow for knowledge-grounded dialogue." In EMNLP 2020.
  * Zhou, Hao, Tom Young, Minlie Huang, Haizhou Zhao, Jingfang Xu, and Xiaoyan Zhu. "Commonsense knowledge aware conversation generation with graph attention." In IJCAI 2018.
  * Tuan, Yi-Lin, Sajjad Beygi, Maryam Fazel-Zarandi, Qiaozi Gao, Alessandra Cervone, and William Yang Wang. "Towards Large-Scale Interpretable Knowledge Graph Reasoning for Dialogue Systems." In Findings of ACL 2022.
  * Yang, Shiquan, Rui Zhang, and Sarah Erfani. "GraphDialog: Integrating Graph Knowledge into End-to-End Task-Oriented Dialogue Systems." In EMNLP 2020.
* Since the proposed method is a combination of subgraph retrieval and dialogue generation, I would like to see comparisons to prior works that also do dialogue generation attending to parts of a graph, e.g., the methods in above listed references.
* About the main motivation across the paper, I have a few concerns:
  * The title is too general according to the scope of this paper. If the main novelty of the proposed method is the subgraph-retrieval augmented method for dialogue generation, I would suggest the title to be more specific, e.g., mentioning “subgraph-retrieval”.
  * About the second paragraph in the introduction, it may not always be true in real applications that only a few hops in a KG can compose a good response. I would suggest revising the paragraph to be more general instead of motivating by one specific dataset that was collected for research purpose and only contained few cases in the real world.
  * The paper claims to jointly retrieve context-relevant subgraph, however, in the experiments, it seems that the “subgraph-retrieval” is based on a manually selected subgraph candidates beforehand i.e., only considering 1-hop in OpenDialKG and 2-hop in KOMODIS. Am I misunderstanding this?
  * Following question c, if the proposed method is based on pre-selected candidates, many prior works already consider such cases (as listed in question (a)). How general is this method when applying to cases with a variant number of hops? How would you analyze its scalability and complexity?



**Summary Of The Paper:**

This paper proposes SURGE, a GNN-based context-relevant subgraph retrieval method for KG-augmented dialogue generation. The proposed method considers the permutation invariance and relation inversion invariation, and uses graph-text contrastive learning to generate knowledge-consistent response.

**Summary Of The Review:**

With my listed weakness, I would suggest considering more about the line of knowledge graph attention and reasoning works into motivation, related work, and experiments (some works are listed above), or adjusting the scope of the current claim.

---

> ### Author Response · Authors · 2022-11-11
> **Response to Reviewer p6ze (3/3)**
>
> ### References
>
> [1] AttnIO: Knowledge Graph Exploration with In-and-Out Attention Flow for Knowledge-Grounded Dialogue, EMNLP 2020.
>
> [2] Commonsense Knowledge Aware Conversation Generation with Graph Attention, IJCAI 2018.
>
> [3] Towards Large-Scale Interpretable Knowledge Graph Reasoning for Dialogue Systems, Findings of ACL 2022.
>
> [4] GraphDialog: Integrating Graph Knowledge into End-to-End Task-Oriented Dialogue Systems, EMNLP 2020.
>
> [5] Scalable Neural Methods for Reasoning With a Symbolic Knowledge Base, ICLR 2020.
>
> [6] EARL: Informative Knowledge-Grounded Conversation Generation with Entity-Agnostic Representation Learning, EMNLP 2021.
>
> [7] Neural Message Passing for Quantum Chemistry, ICML 2017.

---

> ### Author Response · Authors · 2022-11-11
> **Response to Reviewer p6ze (2/3)**
>
> > **[Q3]** The paper title should be more specific, for example, mentioning subgraph-retrieval.
>
> Thank you for your advice. We will take your suggestion into consideration if the paper gets accepted, although the system will not allow us to modify the title now.
>
> ---
>
> > **[Q4]** The motivation in the second paragraph of the Introduction section: "a few hops in a KG can compose a good response in applications", may not be always true and not general.
>
> Thank you for your suggestion. However, we already discussed the general problem, which is **dialogue generation with external knowledge sources**, in the first paragraph of Introduction, and we narrowed down the problem to knowledge-graph-based dialogue generation. Also, we agree that not all questions are answerable by a few hops in KGs, and **we do not make such a claim** in the second paragraph. In the second paragraph, we rather focus on and describe two apparent problems of knowledge-irrelevant and -inconsistent generation in KG-based dialogue models, which is also illustrated in Figure 1. We believe those problems are prevalent, and sufficiently general in KG-based dialogue generation literature.
>
> ---
>
> > **[Q5]** It is unclear that the authors already use the k-hop subgraph (e.g., 1-hop for the OpenDialKG dataset) for the given dialogue, while also claiming the retrieval of the context-relevant subgraph.
>
>  As you understood, we first extract the candidate subgraph (i.e., k-hop subgraph) based on entities in the dialogue context, and then retrieve only the “context-relevant” subgraph on it. Therefore, **the context-relevant subgraph is a subset of the candidate subgraph** extracted from the dialogue context. Please see illustrations of the initial subgraph $\mathcal{G}
> $ and its subgraph $\mathcal{Z}$ based on our retrieval in Figure 2.
>
> ---
>
> > **[Q6-1]** Prior works already consider pre-selected subgraph candidates, extracted from dialogue history. How is this work different from them?
>
> As you mentioned, some prior works use the pre-selected subgraphs extracted by entities in the dialogue history. However, in contrast to them, we further retrieve **only the “context-relevant” subgraph** from the extracted one, since the pre-selected subgraph, simply extracted by all dialogue entities, includes a lot of context-irrelevant facts, shown in Figure 1.
>
> Moreover, in addition to subgraph retrieval, we also propose an **efficient and invariant graph encoding**, as well as **graph-text contrastive learning**, for a knowledge-consistent generation. Please keep in mind that both approaches are sufficiently novel, which is not considered in prior work that focuses mostly on the subgraph extraction part.
>
> ---
>
> > **[Q6-2]** Explain the generality of the proposed model when applied to a variant number of hops, and its scalability and complexity.
>
> Our context-relevant subgraph retrieval is **generalizable to any number of hops**, since it is fully end-to-end trainable jointly with language models from the generation loss, and it does not require supervised signal on retrieval. In other words, as long as we can calculate the generation loss, we can train a retriever, regardless of the retriever’s number of subgraph hops. Please see the **third paragraph of Section 1, and Section 3.6**.
>
> Regarding scalability, our method is scalable, since it requires only **linear computational complexity proportional to the number of facts** in the pre-selected k-hop subgraph. This is because our retrieval model utilizes a message-passing-based graph neural network [7], whose computational complexity is proportional to the number of edges in the graph. This feature enables our model to be applicable to a variant number of hops with low complexity.
>
> ---
>
> > **[Q7]** The two notations $y_{<t}$ and $y_{T}$ are confusing.
>
> Thank you for pointing it out. $y_{<t}$ denotes a sequence of items in $y$ before $t$-th index, which we have changed to $y_{1:t-1}$. $y_{T}$ denotes an item of $T$-th index in $y$.
>
> ---
>
> > **[Q8]**  For KQA evaluation, it is unclear whether the authors use another QA model, or the same dialogue generation model itself.
>
> We apologize for the confusion, and **we use another extractive QA model**, which is described in Appendix D.1 (KQA details paragraph) in detail. In particular, for training another QA model, we first build an extractive QA dataset, which consists of the dialogue history with the generated response as the context, the question from the labeled triplet, and the target answers from candidate answer triplets, illustrated in Figure 3. Then, we fine-tune the pre-trained BERT model on this synthetic QA dataset, which allows the model to measure whether the correct answer knowledge exists in the generated response.
>
> We also have further clarified the KQA evaluation parts in Section 4 and Appendix D.1 during the revision.

---

> ### Author Response · Authors · 2022-11-15
> **Response to Reviewer p6ze (1/3)**
>
> We sincerely thank you for your constructive and helpful comments. We initially address all your concerns below:
>
> ---
>
> > **[Q1]** Missing references on dialogue generation with subgraph extractions [1-4].
>
> Thank you for pointing out related work, however, they are clearly different from ours in both focus and technical details, and are only distantly related to our work.
>
> We first would like to clarify that our work’s main focus is **knowledge-consistent dialogue generation**, and, to achieve this goal, we propose **context-relevant subgraph retrieval**, **graph encoding**, and **graph-text contrastive learning**.
>
> While the works you suggested use knowledge graphs for dialogue generation, their goals are clearly different from our main focus of **enforcing knowledge consistency in generated dialogues under a retrieval-augmented generation setting**.
>
> Although, while some components of our framework may be reminiscent of the existing dialogue generation methods you mentioned, their goals and components are clearly different from ours:
>
> * [1] proposes the path traversal scheme over the knowledge graph, to reach the answer fact for the dialogue context, and **it does not handle dialogue generation**. However, our main focus is generating responses for the ongoing dialogue, which are further consistent with the retrieved knowledge.
>
> * [2] uses the attention over the knowledge graph to reflect the knowledge when encoding the input conversation history, and in generating the output response; however, they are **based on GRU, and do not consider language models**. Note that, language models often have a large number of parameters with internalized knowledge during pre-training. And such points yield entirely new challenges in effectively reflecting the knowledge from knowledge graphs, which we tackle with subgraph-retrieval-augmented generation along with graph encoding and contrastive learning. Also, we **already discussed and compared against a more recent work, which uses attention over knowledge graphs** which improves upon the suggested work [2], namely EARL [6], in our main paper.
>
> * [3] mainly focuses on how to traverse knowledge graphs for the input dialogue history based on the particular differentiable knowledge graph structure [5], and it **does not much consider how to consistently generate the dialogue response for the given fact**, as it simply concatenates the embeddings of traversed entities as the input for dialogue generation. However, our work focuses on knowledge-consistent dialogue generation, and, to do so, we not only explicitly augment retrieved entities directly as the textual input to language models, but also propose effective components for knowledge consistency, such as subgraph retrieval-augmentation, graph encoding, and graph-text contrastive learning, which are highly different.
>
> * [4] uses, similar to the discussed work [2] above, the attention over the knowledge graph, whose architecture is based on GRU. Also, similar to [3] which uses implicit entity embeddings as the input for dialogue generation, this work utilizes **implicit knowledge graph embeddings**. Furthermore, this work does not retrieve only the relevant facts but does consider the **entire knowledge** graph extracted from the dialogue history. In contrast to them, our work not only retrieves only the relevant facts from extracted subgraphs but also directly forwards them to the inputs of LMs.
>
> ---
>
> > **[Q2]** I would like to see comparisons to the above works, about dialogue generation with attention over subgraphs.
>
> Thank you for your suggestion on the additional comparison. First of all, note that **we already discussed and compared EARL [6]** in our work, which is one of the recent works that generate dialogue by performing attention over the sub-knowledge graphs.
>
> However, to fully address your concern, we further compare our method against the most recent method **DiffKG [3] which you suggested**. Note that we have experimented with DiffKG in our experimental setup, for a fair comparison of different methods under the same data preprocessing and data split settings. In the table below, we show that our method significantly outperforms DiffKG, even without any supervision for knowledge retrieval. This result further confirms that the proposed components for knowledge-consistent dialogue generation are highly effective.
>
> | | KQA EM | KQA F1 | BLEU-1 | BLEU-2 | BLEU-3 | BLEU-4 | ROUGE-1 | ROUGE-2 | ROUGE-L | F1|
> |--------|--------|--------|--------|--------|--------|--------|--------|--------|--------|--------|
> | DiffKG |12.25 | 20.99 | 15.68 | 9.13 | 5.60 | 3.46 | 19.50 | 7.07 | 18.84 | 22.26|
> | SURGE (Ours, unsupervised) |48.49 | 55.77 | 17.77 | 11.30 | 7.69 | 5.36 | 21.64 | 9.14 | 20.75 | 25.24|
>
> We have included the comparison result in Table 1 during the revision.

---

> ### Author Response · Authors · 2022-11-29
> **A Gentle Reminder**
>
> Dear Reviewer p6ze,
>
> We sincerely appreciate your time and effort in reviewing our paper. We believe that your comments would further strengthen our paper.
> During the discussion period, we have made every effort to faithfully address all your comments in the responses and revision. Therefore, we kindly request you read our responses and reconsider your rating on our work. We thank you again for your time and efforts in reviewing our paper. Please let us know if you have any further questions.
>
> Best regards, Authors

---

> ### Author Response · Authors · 2022-12-03
> **The end of the discussion phase is approaching**
>
> Dear Reviewer p6ze,
>
> We appreciate your positive comments that our method is a thoughtful combination of important details which convincingly can help the training. We have made every effort to faithfully address all your comments in the response. Here, we briefly summarize the main points of our responses below:
>
> * We have clarified that our work’s main focus is **enforcing knowledge consistency in generated dialogues under a retrieval-augmented generation setting**, by explaining the major difference against each work you mentioned in detail.
> * Following your suggestion, we have conducted **a comparison against the most recent method DiffKG which you suggested**. Furthermore, we showed our method outperforms the DiffKG even without any supervision for knowledge retrieval.
>
> Please let us know if you have anything else we should address. We thank you again for your time and efforts in reviewing our paper.
>
> Best regards, Authors

---

> ### Comment · Reviewer_p6ze · 2022-12-10
> **Thanks for the response**
>
> Thanks for the authors' response. After reading the authors' response, some of my original concerns are solved. However, some of my concerns still remain, I tend to keep the score unchanged.
>
> 1. It is not clear to me the reasons not comparing other subgraph-retrieval based methods. If considering the merits of this work being the proposed context-relevant subgraph-retrieval method, it seems needed to compare other methods that retrieve a subgraph related to the context even in an implicit way. The paper directly jumps into comparison to EARL and DiffKG, but does not discuss the difference among the main claims of the three works. Also, there are many other works using different methods but are missing in the paper. It is fine to not compare to all prior methods, however, I do not see discussion in the paper about prior subgraph-retrieval based works, including the details of how this work differs from them and how it gains its novelty from others.
> 2. On the other side, considering all k-hop can drop the performance on datasets that do not require much hops information, however, it can be crucial for measuring if a method can be generalized to a more complicated KG. It would be nice to see how SURGE can work on k-hop candidates rather than a pre-pruned 1- or 2-hops KG, and see the study of its computational costs. Given this reason, I am not sure if the comparisons in the experiment are fair.

---

> > ### Author Response · Authors · 2022-12-11
> > **Follow-up Response to Reviewer p6ze**
> >
> > We sincerely appreciate you for following up our response, and providing additional comments. In this response, we address all your remaining concerns in your response below:
> >
> > ---
> >
> > > **[Q1]** It is not clear to me the reasons for not comparing other subgraph-retrieval based baselines.
> >
> > First of all, we emphasize that our work primarily focuses on generating knowledge-consistent dialogue given the knowledge graph, rather than just subgraph retrieval. The context-relevant subgraph retrieval is one of our proposed components which is end-to-end trainable with the generation objective and enhances the model’s performance by retrieving only context-relevant pieces of the facts from the k-hop KG.
> > Moreover, we have already compared our approach to previous works in terms of subgraph retrieval. Specifically, we have already discussed EARL in the **Related Work** section of the main paper and DiffKG in Q1 of the **initial response Q1**.
> >
> > Regarding many other works using different methods but are missing in the paper, **it is unclear what the prior subgraph-retrieval based works you mentioned are**. In the initial response Q1, we already discussed the works you mentioned and you seem to understand the difference between our work and mentioned works. If you have a **concrete list of the works** we have overlooked but should be addressed and compared, please let us know so that we can consider them and make a fair comparison against them.
> >
> > ---
> >
> > > **[Q2]** I am not sure if the comparisons in the experiment are fair.
> >
> > We are confident that the comparisons in the experiment are fair. We use the same hop of knowledge graph across all baselines and our methods, as stated clearly in Section 5.1 of the main paper.
> >
> > ---
> >
> > We hope that our answer clarifies your concerns. We are looking forward to your follow-up response, particularly regarding **the previous subgraph retrieval-based dialogue generation works that you claimed we had missed**.

---

### Author Response · Authors · 2022-11-11
**General Response**

We sincerely thank all reviewers for their constructive comments. We are happy to see that most reviewers acknowledge the novelty of our work in the proposed components and the evaluation metric (Reviewers ypx1, CSao), found the paper well-written (Reviewers p6ze, ypx1, CSao, YCpx, 6h9N), and the experiments as extensive and showing the effectiveness of the proposed method (Reviewers ypx1, KmrR, CSao, YCpx).

During the rebuttal period, we have faithfully responded to all reviewer's comments in the individual response of each reviewer, and here we briefly summarize the major updates that we have made in the paper during the revision, which we highlight in blue.

- We have improved the notations and explanations in Section 3.
	- Modify notation $y_{<t}$ to $y_{1:t-1}$ in Section 3.1 (**Reviewer p6ze**)
	- Unify $q_e$ and $q_r$ to $q$, and revise the explanation on it accordingly in Section 3.1 (**Reviewer ypx1**).
	- Clarify the decomposition of subgraphs, and node-edge representations for triplet retrieval in Section 3.3 (**Reviewers p6ze, ypx1**).
	- Clarify the graph encoding, which uses graph-structural information, in Section 3.4, and update Equation 7; Additionally include Figure 8 in the Appendix, for illustrating graph encoding (**Reviewers CSao, YCpx**).
- We have provided more details on our novel metric, KQA, in Section 4 (**Reviewers p6ze, ypx1**).
- We have fixed the caption of Table 3 to clarify the aim of the experiment: validating contrastive learning (**Reviewers CSao, 6h9N**).
- We have fixed hyperlinks for Figure and Table, which are now clickable with their keywords as well (**Reviewer ypx1**).
- We have fixed other minor errors, and also included more details in the Appendix.

We hope our responses address your concerns. Please let us know if there is anything else we need to address.

---

### Decision · Program_Chairs · 2023-01-20

**Decision:**

Reject

**Justification For Why Not Higher Score:**

Novelty and other issues.

**Justification For Why Not Lower Score:**

N/A

**Metareview: Summary, Strengths And Weaknesses:**

In this paper, the authors propose an integrated solution to combine knowledge subgraphs and language models for dialogue systems. The authors propose SUbgraph Retrieval-augmented GEneration (SURGE), which retrieves a subgraph from the KG. Then, it learns a latent representation space using contrastive learning. The AE and the reviewers have reached a consensus about the following strengths:

+++ This is a very important direction in dialogue systems research.
+++ The authors try to explicitly model the contextual relevance, which is a good idea.

The panel has also found several limitations and concerns:
--- The novelty is incremental. All techniques are not new, and the authors assemble the three techniques, especially the idea of subgraph retrieval is not very new.
--- Several reviewers are concerned about the invariance of the triples encoding. Reviewers are not satisfied with the authors' reply. The proposed method might depend on the order. The implementation might have an issue, and imposing the order is not the best solution.
--- The related work still needs work, and some subgraph retrieval work is ignored in related studies. It is unclear if the comparison with DiffKG is conducted in a fair setting.
--- Contrastive learning results are not very impressive.
--- Ablation experiments are lacking: different graph encoding mechanisms would be experiments (no comparisons).

Based on the panel discussion and the AC-reviewer Zoom meeting, the panel has concluded that this paper is not ready to be accepted.


**Summary Of Ac-Reviewer Meeting:**

On Dec 8 Thursday, the AC and the reviewers concluded a Zoom panel to discuss this borderline paper. Four out of the six reviewers were at the meeting with the AC: one missed the meeting due to traveling at EMNLP, and another reviewer did not make it. In the meeting, there were no clear champions of this paper among the five attendees. The main concerns brought by all reviewers were the lack of technical novelty: the paper uses an assembly of existing known techniques, and therefore scientific innovation is lacking. There were other concerns raised in the meta-review above. The reviewers who gave favorable ratings have also commented that they feel this paper is not ready to be accepted due to the above limitations.